# Aging aerosol in a well-mixed continuous flow tank reactor: An introduction of the activation time distribution

Franz Friebel, Amewu A. Mensah

Institute for Atmospheric and Climate Science, ETH Zurich, Zurich, 8092, Switzerland

5   *Correspondence to*: Franz Friebel (franz.friebel@env.ethz.ch) and Amewu A. Mensah (amewu.mensah@env.ethz.ch)

**Abstract.** Two approaches are common to simulate atmospheric aging processes in the laboratory. The experiments are either performed in large aerosol chambers (several $m^3$) in order to achieve extended observation times or in small chambers (< 1 $m^3$) compensating the short observation times by elevated reactant concentrations. We present an experimental approach that enables long observation times at atmospherically relevant reactant concentrations in small chamber volumes by operating the aerosol chamber as a Continuous flow Stirred Tank Reactor (CSTR). We developed a mathematical framework that allows the retrieval of data beyond calculating mean values such as $O_3$-exposure or equivalent atmospheric aging time, using the new metric: activation time ($t_{act}$). This concept was developed and successfully tested to characterize the change in cloud condensation nuclei (CCN) activity of soot particles due to heterogeneous ozone oxidation. We found very good agreement of experimental results to the theoretical predictions. This experimental approach and data analysis concept can be applied for the investigation of any transition in aerosol particles properties that can be considered as a binary system. Furthermore, we show how $t_{act}$ can be applied for the analysis of data originating from other reactor types such as Oxidation Flow Reactors (OFRs), which are widely used in atmospheric sciences. The new $t_{act}$ concept significantly supports the understanding of data acquired in OFRs especially these of deviating experimental results in intercomparison campaigns.

## 1   Motivation

Atmospheric aerosols undergo various chemical reactions and physical modification processes once they are emitted into the atmosphere. The time scale for such reactions and processes depends on the atmospheric lifetime of the individual aerosol species. For example sea salt particles have a lifetime of approximately 0.4 days whereas soot particles can have a lifetime of more than a week (Textor et al., 2006). Simulating atmospheric aging as realistically as possible is essential to understand the real impact of different pathways of ambient particle processing. This includes the potential of aerosol particles to form clouds, an important process affecting climate and weather. Mimicking extended aging times is one of the most challenging tasks for the investigation of aerosols under laboratory conditions (Burkholder et al., 2017). There are two common approaches to solve this problem. One is to store the aerosol of interest in large chambers to achieve long observation times. Here aging durations of up to 16 hours and beyond at atmospherically relevant reactant concentrations can be achieved, which has been shown e.g. for the SAPHIR chamber of FZ Julich with a volume of 270 $m^3$ (Rohrer et al., 2005; Rollins et al., 2009). Extending the observation time by increasing the chamber volume is often technically and economically not feasible. The second option is to increase the concentration of the reactive compounds such as oxidants and aerosol particles, in order to trigger higher reaction rates and thereby reduce the reaction time (George et al., 2007; Huang et al., 2017; Ihalainen et al., 2019; Kang et al., 2007; Keller and Burtscher, 2012; Simonen et al., 2017). This allows to significantly reduce the volume of the aerosol chamber. However, in this case the reactant concentrations can be elevated by several orders of magnitude in comparison to the atmosphere. Thus, the results of such experiments can be misleading with respect to their atmospheric relevance because the aerosols aging rates are not always directly proportional to the concentration of the oxidants. Further, the aging pathways can

differ significantly besides the perturbed partitioning of reactive species and products (Donahue et al., 2006; McNeill et al., 2007; Renbaum and Smith, 2011).

Here we present an experimental approach that can be used to achieve long aerosol aging times with neither need for large chamber volumes nor high reactant concentrations by operating an aerosol chamber in the Continuous flow Stirred Tank Reactor (CSTR) mode. In a setup at ETH Zurich, aging times of more than 12 h were achieved in a stainless-steel chamber of 2.78 m$^2$ volume. This greatly exceeds the typical exposure times of several minutes that can be reached within Oxidation Flow Reactors (OFRs; Simonen et al., 2017) while compared to large environmental chambers with volumes of e.g. 270 m$^3$ the chamber used here can be considered rather small (Cocker et al., 2001; Leskinen et al., 2015; Nordin et al., 2013; Paulsen et al., 2005; Platt et al., 2013; Presto et al., 2005; Rohrer et al., 2005). Furthermore, this experimental approach requires an aerosol particle concentration that is low enough to allow for size-selection of the aerosol prior to the injection into the reaction chamber.

The CSTR-approach describes an aerosol chamber, which is continuously filled with an aerosol flow constant in composition over time. The volume of the aerosol chamber is actively stirred in order to achieve a homogenous aerosol mixture. Due to the mixing, the aerosol that is continuously extracted for analysis consists of a well-defined mixture of aerosols at different aging stages. From this perspective, the CSTR-approach is closer to atmospheric processes than other reactor types as in the real atmosphere except for individual plume emissions aerosols are rather continuously emitted, mixed, and removed. This results in a mixture of aerosols at different aging stages, but of course, the atmospheric mixture is less well defined compared to an aerosol in a CSTR.

The CSTR concept has been applied in chemical engineering for a long time e.g. (Cholette and Cloutier, 1959). One of the reasons for its limited application in atmospheric science might be the increased complexity in data analysis in comparison to batch-experiments. While changes in particle properties that can be considered continuous (e.g. particle concentration or particle growth) can be well described by different theoretical concepts e.g. (Crump and Seinfeld, 1980; Kuwata and Martin, 2012; Levenspiel, 1999), limited theoretical descriptions seem to exist for changes in particle properties that can be considered as transitions within a binary system. Such transitions in binary systems are step-wise, also referred to as non-gradual changes in a particle property, such as:

1) Freezing of a water droplet: Step-wise and therefore non-gradual change in the particle density; the water is either in liquid or solid state.

2) Deliquescence of soluble aerosol particles: The particles show a step-wise i.e. non-gradual increase in diameter.

Binary particle properties are not necessarily intrinsic particle properties, but can also be defined by the measurement protocol.

3) CCN-activity: The chemical and physical properties of an aerosol particle can vary, but the particle is either CCN-inactive or CCN-active at a defined super saturation (SS).

4) Growth beyond a threshold: Condensational growth of an aerosol particle leads to a continuous and gradual increase of the particle diameter. A binary system can be defined by introducing a threshold diameter that can be arbitrarily chosen. The aerosol particle is either smaller or larger than this defined threshold diameter. The same holds true when particles are separated e.g. in aerosol impactors.

Therefore, the concept of non-gradual transitions/transitions within binary systems can be used to describe a multitude of changes in particle properties.

In the following, we discuss a theoretical basis for the analysis of time-dependent changes in binary systems within well-mixed continuous flow aerosol aging chambers (CSTR-approach). We developed a mathematical framework which allows the retrieval of characteristic parameters from the system of interest (e.g. CCN activity) and which allows for the calculation of the parameter of interest throughout the entire duration. Key element in this framework is the activation time ($t_{act}$) which marks

the time after which the individual aerosol particle undergoes a transition within a binary system. We start by introducing an idealized system in which $t_{act}$ can be described by a single number and proceed to a more realistic setting in which we incorporate a distribution of particles with different individual $t_{act}$'s (activation time distribution, P($t_{act}$)). Further, we test the $t_{act}$-concept on real experimental data and finally apply it to other types of continuous flow aging chambers such as OFRs. We

show that application of the $t_{act}$-concept is capable of giving new insights to OFR data and further significantly improves the understanding of discrepancies in experimental results obtained in intercomparison studies Lambe et al., (2011) with different reactors such as the Potential Aerosol Mass chamber (PAM; Kang et al., 2007) and the Toronto Photo-Oxidation Tube (TPOT; George et al., 2007) . (George et al., 2007)(George et al., 2007)(George et al., 2007)

## 2    Introduction of the CSTR

The combination of results obtained from laboratory experiments, field measurements, and modelling studies promotes the understanding of atmospheric aging of aerosols. Designing experiments in the laboratory involves the aim to mimic atmospheric processes as close as possible to realistic atmospheric conditions. The benefit of laboratory experiments is that the investigation of reactions is not dependent on various uncontrolled parameters such as meteorological conditions (e.g. wind direction). Further process variables, such as reactant concentrations, can be actively and therefore systematically modified to

allow for a detailed investigation of their impacts. Such type of experiments are typically performed by creating an artificial atmosphere within reactors. From a technical perspective, generally three types of reactors are distinguished: the batch-reactor, the Plug Flow Reactor (PFR), and the Continuous flow Stirred Tank Reactor (CSTR).

In an aerosol chamber operated in batch mode, the reaction volume is first filled with the sample aerosol as fast as possible to achieve high homogeneity of the sample. After the desired start concentration is reached further addition of the sample aerosol

is stopped and the aging is initiated e.g. by addition of the oxidant. This point in time is generally defined as the start of the experiment and referred to as $t = 0$. Data acquisition of the aging sample takes place while the reaction volume is flushed with sample-free gas. The composition throughout the chamber is homogeneous but evolving in time, therefore no steady state conditions are ever achieved. This concept is used to operate many large scale environmental chambers (Cocker et al., 2001; Leskinen et al., 2015; Nordin et al., 2013; Paulsen et al., 2005; Platt et al., 2013; Presto et al., 2005; Rohrer et al., 2005).

A PFR is a steady state reactor in which no mixing along the flow path (axial mixing) but perfect mixing perpendicular to the flow (radial mixing) takes place. Further, a continuous feed-in of reactants and withdrawal of sample take place at equal flow rates simultaneously. This results in a constant composition of the output solely depending on the residence time within the reactor. This ideal system is approximated by many Oxidation Flow Reactors (OFRs) e.g. PAM chamber (Kang et al., 2007), TPOT chamber, Micro Smog Chamber (MSC; Keller and Burtscher, 2012), the TUT Secondary Aerosol Reactor (TSAR;

Simonen et al., 2017) or the Photochemical Emission Aging flow tube Reactor (PEAR; Ihalainen et al., 2019) . The main difference between an ideal PFR and real OFRs is that in OFRs significant but unintentional mixing of the aerosol along the flow path takes place (Mitroo et al., 2018). Therefore, OFRs show a significant residence distribution.

The CSTR is a steady state reactor with a constant reactant feed in and sample withdrawal as well but opposite to OFRs, the volume is actively stirred to achieve a homogeneous composition throughout the reactor volume. Due to the active mixing,

sample stream composition and conditions are the same as within the entire chamber volume. The concept of the CSTR requires perfect internal mixing, which cannot be achieved in real systems. However, due to the good miscibility and low viscosity of gases and the aerosol particles being homogenously dispersed, it is possible to achieve a degree of mixing which is very close to a perfectly mixed system. Especially in the case of mimicking atmospheric processes, residence times of several hours are achieved. Compared to that, the time needed for dissipating all gradients, which is in the order of seconds to minutes, can be

considered small. The operation procedure we introduce here starts with feeding in aerosol into an initially reactant free gas phase within the CSTR, referred to as filling regime. After a certain time of filling, the composition within the CSTR does not

change anymore and a dynamic equilibrium is reached, referred to as steady state. The time required to reach this state depends on the characteristics of the CSTR and flow rates. During a subsequent flushing regime, the aerosol is flushed out with reactant free gas. Each of these regimes can be used independently for data analysis.

The key parameter for the description of reactions within a CSTR is the hydrodynamic residence time ($\tau_{CSTR}$) which is also the mean residence time. It can be obtained from the reactor volume ($V_{CSTR}$) and the volumetric flow through the CSTR ($\dot{V}$) as shown in eq. (1; Levenspiel, 1999).

$$\tau_{CSTR} = \frac{V_{CSTR}}{\dot{V}} \tag{1}$$

## 2.1 Filling regime

As the CSTR volume is initially sample free, the aerosol particle concentration in the CSTR increases continuously during the filling regime until it reaches a stable concentration. The aerosol particle concentration ($[A_{CSTR}(t)]$) at any point in time can be calculated as a function of the experimental duration ($t$) by eq. (2), where ($[A_{feed-in}]$) is the aerosol concentration in the feed-in flow.

$$[A_{CSTR}(t)] = [A_{feed-in}] \cdot \left(1 - e^{\left(\frac{-t}{\tau_{CSTR}}\right)}\right) \tag{2}$$

We define that the CSTR reaches steady state conditions when the difference between $[A_{CSTR}(t)]$ and $[A_{feed-in}]$ is smaller than the resolution of the analytical instruments deployed. To standardize the time period, we chose the fourfold mean residence time ($4\tau$ criterion) as reference point for the start of the steady state in this publication. At this point the difference between $[A_{CSTR}(t)]$ and $[A_{feed-in}]$ is less than 2 % which is lower than the resolution of most aerosol particle counters (Mordas et al., 2008).

## 2.2 Steady state

The steady state is in fact the part of the filling regime where the CSTR is in a dynamic equilibrium. All processes and reactions continue but the concentrations of all compounds remain constant over time. In theory, this operation point can be maintained for an infinite experimental duration. Be aware that this does not mean that an infinite degree of aerosol aging can be achieved. In steady state the experimental duration is decoupled from the particle age, which is in contrast to experiments in batch-chambers but similar to OFR experiments. As a result of the continuous feed-in and flush-out flow, different aerosol fractions that enter the CSTR at different times are present simultaneously, resulting in a residence time distribution (RTD). In CSTRs the RTD can be described by eq. (3) and is plotted in Fig. 1 (solid black line – labeled with "steady state"). With an increasing individual residence time the fraction of aerosol particles declines exponentially. The individual residence time of a specific particle fraction is indicated by the color-coding in Fig. 1. The actual number of particles within an individual particle fraction at a specific residence time can be calculated by integrating RTD over time (eq. (4)). This leads to the residence time sum distribution $RTD_{sum}$ represented by the colored area under the curve. Note, while we choose $RTD(t)$ and $RTD_{sum}(t)$ for a more intuitive denotation, generally E(t) and F(t), respectively, are the official formula symbols especially in the engineering community (Levenspiel, 1999).

$$RTD(t) = e^{\frac{-t}{\tau_{CSTR}}} \tag{3}$$

$$RTD_{sum}(t) = \int_0^t RTD(t)\, dt = 1 - e^{\frac{-t}{\tau_{CSTR}}} \tag{4}$$

## 2.3 Flushing regime

From the point in time that no fresh aerosol but only particle free air is added the CSTR is operated in the flushing regime. This operation mode can be considered similar to the operation of batch-aerosol chambers as in both cases the aerosol is flushed out continuously.

The initial RTD at this switching point ($t_{switch}$) and therefore the ratio of young to old aerosol fractions is preserved throughout the entire flushing duration. Nevertheless, the individual residence time of every single aerosol fraction rises with flushing duration. In other words: All particles age simultaneously. Figure 1 illustrates how the RTD changes in the flushing regime. Note, the time on the x-axis is plotted as dimensionless time in multiples of $\tau$. Each color in the area represents an individual aerosol fraction with a corresponding residence time. Blue stands for the lowest and red for the highest residence times. The

solid black curve labeled "steady-state" represents the RTD in steady state while the other curves show the RTDs for additional time increments after the flushing regime has been initiated ($t_{switch}$). For example, the area und the grey curve labeled "+1 $\tau$" represents the RTD 1 $\tau$ after initiation of the flushing regime. The grey dashed line stands for the activation time $t_{act}$, a threshold time that will be introduced later. Here it marks a threshold time. With increasing flushing time, the fraction of aerosol particles that have an individual residence time higher than this threshold time increases. From some point in time on all particles have

crossed this threshold time as is the case for the particles under the light grey curve at "+2 $\tau$" after $t_{switch}$.

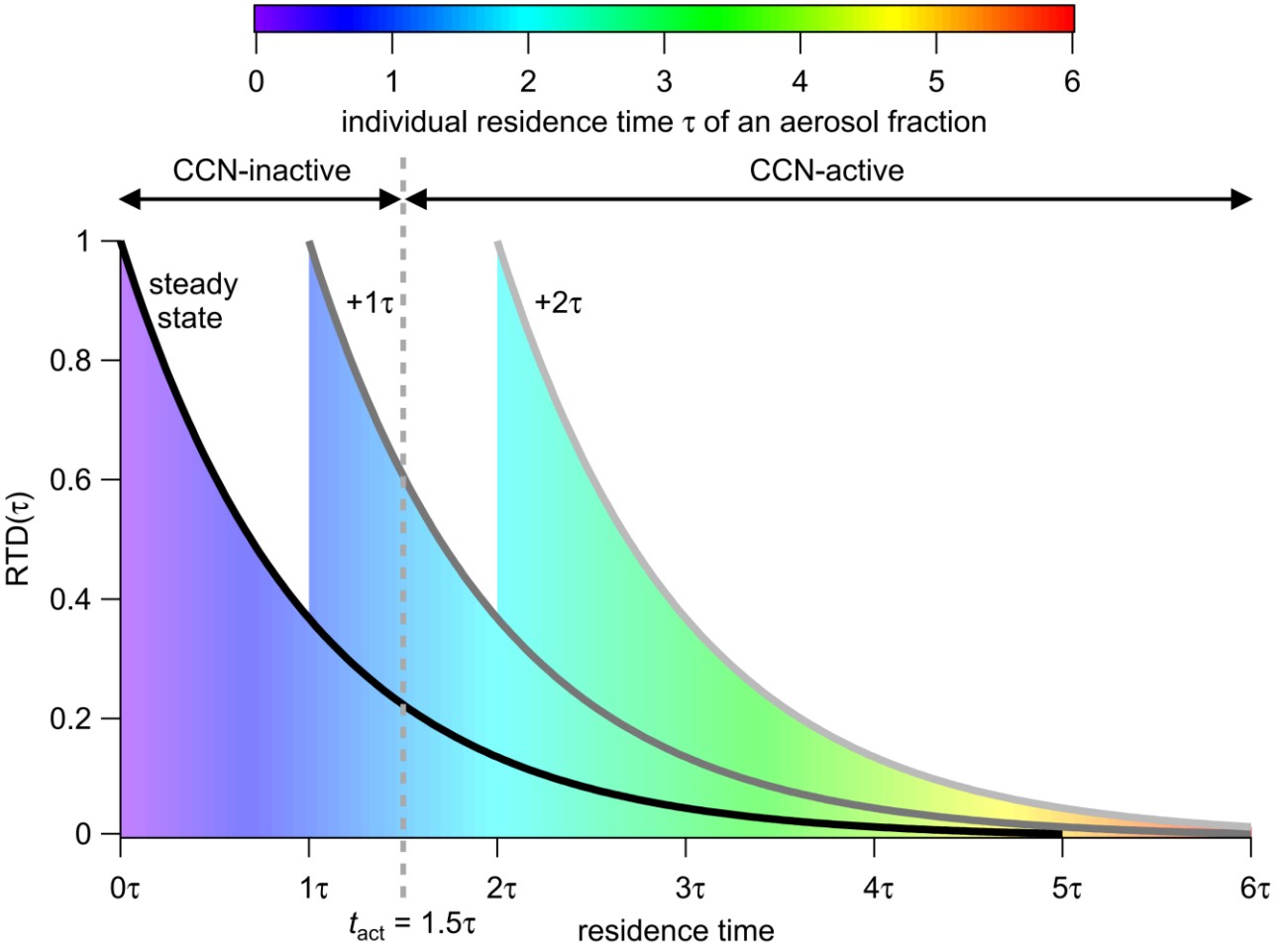

**Figure 1: RTD inside the CSTR within steady state (black line) and for different time steps after the CSTR operation was switched to the flushing regime. The area below the curve is proportional to the fraction of aerosol particles at a specific residence time. The individual residence time of a specific particle fraction is indicated by the color-coding. The time on the x-axis is plotted as**

**dimensionless time in multiples of the hydrodynamic residence time $\tau_{CSTR}$.**

While the initial RTD is preserved, the total aerosol concentration ($[A_{CSTR}(t)]$) is declining exponentially due to flushing with reactant free gas. $[A_{CSTR}(t)]$ at any point in time can be calculated using eq. (5) taking the aerosol concentration in the CSTR at the switching point ($[A(t=t_{switch})]$) into account.

$$[A_{CSTR}(t)] = [A(t = t_{switch})] \cdot e^{\left(-\frac{t - t_{switch}}{\tau_{CSTR}}\right)} \qquad (5)$$

## 2.4 Comparison with aerosol aging experiments in batch-mode

The use of a CSTR provides additional opportunities in performing aerosol aging experiments, but comes with a more complex experimental setup compared to aerosol chambers that are run in batch-mode. An ideal batch-aerosol chamber has to be filled instantly but in reality this ideal filling procedure is almost impossible to be achieved. If the filling time is short compared to the total aging time, the initial RTD can be ignored in data analysis. In contrast to that, the filling of the chamber is already part of CSTR experiments and data analysis can be performed on the partially aged aerosol. The flushing regime in both reactor types is similar, but at the start of flushing the aerosol inside the CSTR is already partly aged as defined by the RTD. Therefore, operating an aerosol chamber of defined volume in CSTR-mode allows to achieve higher aging times than compared to batch mode operation at the same sample extraction flow rate.

## 3 Introduction of the activation time ($t_{act}$) for transitions in binary systems

Due to the residence time distribution, data acquired from CSTR experiments require a different analysis approach than data from batch-mode experiments. While there are analysis concepts to describe continuous changes on the level of single particles in CSTR experiments (e.g. condensational growth; Kuwata and Martin, 2012), so far there is no concept that describes transitions in binary systems (in atmospheric sciences) to the best of the knowledge of the authors.

Binary systems can be considered as systems that show a step-wise change in a particle property as a function of an external parameter. Since this is opposite to a continuous/gradual change in a particle property, it can be also described as a non-gradual transition. As mentioned in the introduction, soluble aerosol particles such as ammonium nitrate exhibit a significant change in diameter with increasing relative humidity ($RH$) due to deliquescence. Similarly, the change from cloud condensation nuclei (CCN) to activated droplets due to exposure to a super-critical super saturation (SS) results in a fast increase of the particle diameter from the nanometer to the micrometer scale that is hard to be continuously tracked by standard measurement instrumentation. A defined diameter threshold is hereby used to distinguish between an aerosol particle and a solute droplet in the case of deliquescence. This is the same between non-activated CCN and cloud droplets. In both examples the relative humidity ($RH$) in the surrounding gas phase can be considered the external parameter that controls if an aerosol particle is in either of the two states of the binary system (effloresced vs. deliquesced/CCN vs. cloud droplet).

We may assume a system in which all external parameters stay constant but the particle itself undergoes a continuous transformation, e.g. due to oxidation. After a certain period of time, this continuous transformation, in this specific case oxidation, can lead to a change in a binary property, e.g. CCN-activity. Ultimately, the step-wise or non-gradual transition is a function of time. We define the required time span (e.g. necessary aging time) that leads to a change in a specific particle property, resulting in a transition in a binary system in another particle property as the activation time ($t_{act}$). This concept is generally valid and can be applied to any kind of transition in a system defined as binary either by intrinsic or operational parameters.

Nevertheless, in the effort to increase the comprehensibleness of the following introduction of $t_{act}$, we focus on the example of an aerosol particle aging process resulting in an increased CCN-activity. The aging process is a continuous and irreversible process that changes how a single particle can accumulate water at super saturated (SS) conditions. Once a particle reaches the necessary aging time $t_{act}$ it is considered to be CCN-active at a respective SS.

In reality, no time-dependent transition can be found that truly follows a step-function or that is truly non-gradual. If at all, some transitions can rather be described by a steep sigmoidal function. However, some transitions occur on a time scale that

is so short compared to e.g. the time resolution of the measurement that they appear to be non-gradual. Thus, we consider it a valid simplification to treat these transitions as step-wise and non-gradual and to define them as transitions in a binary system.

## 3.1 Aerosol particle activation and activation time in a CSTR

The fraction of particles acting as CCN (activated fraction; $AF$) is defined as the ratio of activated particles divided by the total number of particles in the sample volume. The measured $AF$ can be used to obtain the activation time $t_{act}$. Vice versa, if $t_{act}$ is know the theoretical $AF$ can be calculated throughout the entire experiment. Here, the three different regimes (filling, steady state, flushing) have to be treated individually.

### 3.1.1 Particle activation during the filling regime

Assuming that only aerosol particles are CCN-active which have an individual residence time in the aerosol chamber that is above $t_{act}$, the theoretical $AF$ can be calculated according to eq. (6). Two different time ranges within the experimental durations need to be considered. If the experimental duration $t$ is below $t_{act}$, $AF$ is 0 as even the particles that entered the aerosol chamber at the very beginning have an individual residence time shorter than $t_{act}$ and therefore cannot be CCN active yet (eq. 6a). If the experimental duration $t$ is above $t_{act}$, $AF$ is greater than 0 as a subset of the particles will have an individual residence time longer than $t_{act}$ and therefore can be CCN active (eq. 6b). Application of eq. (3), which describes RTD and rearrangement of eq. 6 allows for the calculation of the activation time $t_{act}$ based on an experimentally determined $AF$ as shown in eq. (7). This equation is valid throughout the entire filling regime including steady state.

$$t \leq t_{act} : \quad AF(t) = 0 \tag{6a}$$

$$t > t_{act} : \quad AF(t) = \frac{\text{activated particles}}{\text{all particles}} = \frac{\int_{t=t_{act}}^{t} RTD(t)\, dt}{\int_{0}^{t} RTD(t)\, dt} = \frac{RTD_{sum}(t) - RTD_{sum}(t=t_{act})}{RTD_{sum}(t)} \tag{6b}$$

$$t_{act} = \ln\left( 1 - \left( (1 - AF(t)) \cdot \left( 1 - e^{\frac{-t}{\tau_{CSTR}}} \right) \right) \right) \cdot (-\tau_{CSTR}) \tag{7}$$

### 3.1.2 Particle activation during steady state

After the conditions in the aerosol chamber reached steady state, the measured $AF$ does not change anymore. This is due to the fundamental concept of a CSTR which entails a continuous addition of fresh particles and simultaneous withdrawal of sample at equal flow rates resulting in a dynamic equilibrium and a constant RTD.

To simplify matters, the reason for the constant $AF$ within this dynamic equilibrium can be visualized when focusing on three distinct time periods within the continuum of the RTD and thereby on three specific particle fractions. Fraction one is within the right tail of the RTD and consists of particles with a residence time that is above $t_{act}$. They are only a few compared to the total number of particles and a fraction of these is constantly flushed out with the sample stream. This would lead to a hypothetical reduction of $AF$ if not simultaneously the second particle fraction of interest was in the situation to have an individual residence time that is just about to exceed $t_{act}$. The particles within fraction two are thereby transitioning from the CCN inactive particle fraction within the aerosol chamber to the CCN active particle fraction. The hypothetical loss of CCN inactive particles would lead to an increase in $AF$ if not again simultaneously the third particle fraction of interest consisting of fresh and CCN inactive particles was about to be added to the chamber volume.

Due to this dynamic equilibrium, eq. (7) can be simplified to eq. (8) assuming that the experimental duration $t$ approaches infinity $\left( \lim_{t \to \infty} eq.\ (7) = eq.\ (8) \right)$

$$t_{act} = -\ln(AF) \cdot \tau_{CSTR} \tag{8}$$

While the experiment can run theoretically for an infinite time, each individual particle fraction has in fact a limited lifetime within the aerosol chamber. Metaphorically speaking the particle fraction travels along the RTD curve from the left (residence time = 0 min) to the right in Fig 2 within its lifetime. Since the RTD is an exponential curve asymptotically approaching zero, in theory there should always be at least an infinitesimal small fraction of particles with a residence time equal to the experimental duration. In reality though, the maximal residence time of an individual particle fraction is defined by the characteristic parameters of the CSTR $\tau_{CSTR}$ and the detection limit of the measurement instrument. Once the particle concentration is below the detection limit of the measurement instrument, there are de facto no particles with a higher residence time than the one corresponding to this detection limit.

Another important aspect is the non-linear, but exponential correlation between $AF$ and $t_{act}$ as can be seen in eq. (8) and Fig 2. In Figure 2 a RTD (black curve) inside a CSTR with $\tau_{CSTR}$ =120 min in steady state is shown. The area under the curve represents the total particle population inside the CSTR and is equal to 1. By definition, $AF$ is the ratio of activated particles to the total number of particles in the respective volume. According to the $t_{act}$-concept, only particles with a residence time beyond $t_{act}$ (grey dashed vertical line in Fig.2) are CCN-active. The activation time $t_{act}$ therefore separates CCN-inactive ($t \leq t_{act}$) from CCN-active ($t_{act} > t$) particles.

In case $t_{act}$ is known, $AF$ can be obtained by integrating the RTD from $t = t_{act}$ to $t = \infty$, which is shown in eq. (6b). Vice versa, the fraction of CCN-inactive particles can be obtained by integrating the RTD from $t = 0$ min to $t = t_{act}$.

Supposed, an $AF$ of 0.368 is experimentally determined. This corresponds to 0.368 of the area under the RTD curve. In accordance with the discussion above, we can imagine to start the integration from the right ($t = \infty$) till a value of 0.368 is achieved which is equal to the area under the blue curve and the lower limit of the residence time (vertical blue bar) then corresponds to $t_{act}$. In a second step, we examine the case of an experimentally determined $AF$ of 0.134. Following the procedure outlined for the first case ($AF = 0.368$), the integration results in the entire area under the turquois curve and a $t_{act}$ of 240 min (turquoise vertical bar). In the third case with an experimentally determined $AF$ of 0.049, which corresponds to the area under the green curve, we determine a $t_{act}$ of 120 min (vertical green bar). Ultimately, $AF$ declines exponentially with increasing $t_{act}$ and vice versa.

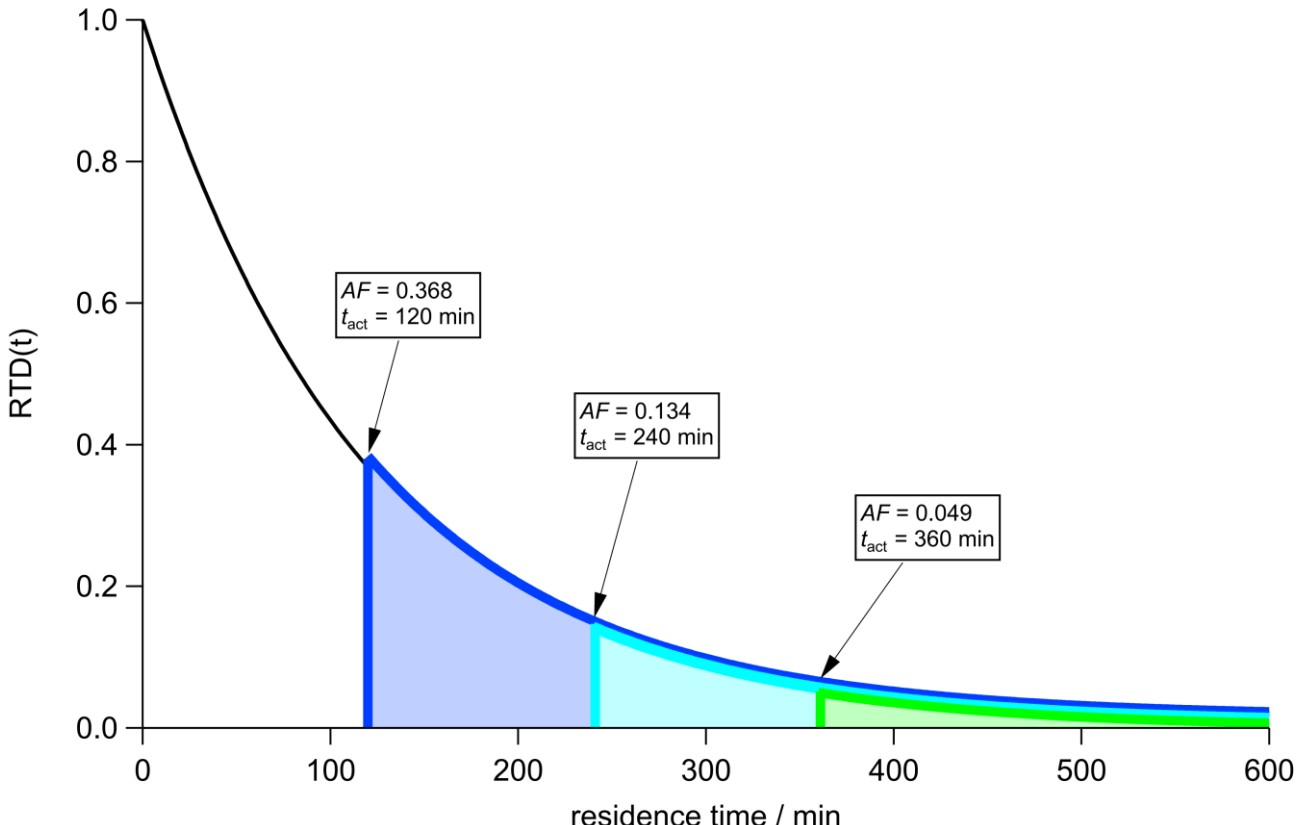

**Figure 2: Residence time distribution (RTD; black curve) for particles in a CSTR with $\tau_{CSTR}$ = 120 min in steady state. The area under the black curve represents the total particle number concentration. Three subsets of this area are highlighted corresponding to the fraction of CCN-active particles in dependence of the respective activation time $t_{act}$.**

### 3.1.3 Particle activation during flushing regime

At the beginning of the flushing regime ($t_{switch}$), the afflux of fresh particles is stopped and replaced by a particle free gas stream. As the sample extraction is maintained the total particle number (concentration) within the aerosol chamber depletes with time. Nevertheless, the particles that entered the aerosol chamber before $t_{switch}$ continue to age during their increasing individual residence time. This causes a transformation of the RTD along the x-axis/residence time but no transformation of the shape of the RTD as the ratio of particles of different residence time stays constant. This process is indicated by the multiple curves in Fig. 1. The RTD of the particles in steady state is represented by the solid black line. As no fresh particles are added anymore, but the aging of the present particles continues the RTD curve is shifted along the x-axis towards the right.

For example, after a time period equaling the hydrodynamic residence time of the CSTR (+1 $\tau$) the particle RTD is represented by the dark grey solid line. As only the fraction of particles with an individual residence time above $t_{act}$ is CCN active, $AF$ at +1 $\tau$ is significantly higher than during steady state. This is indicated by a larger area under the curve that crossed $t_{act}$ (grey dashed vertical line in Fig 1.). Throughout additional flushing time the RTD is shifted further towards longer residence times. At some point all particles have a residence time beyond $t_{act}$. This means that all particles are CCN-active resulting in an $AF$ of 1 which is for example the case for the particles in the area underneath the light grey curve in Fig 2 (+2 $\tau$). In reality this is not a stepwise process with time increments of 1 $\tau$, but a continuous process that involves an exponential increase of $AF$ inside the CSTR until $AF$ = 1.

This change in $AF$ can be mathematically captured. The first step is to derive an equation that describes what fraction of the RTD has crossed the point $t_{switch}$ after flushing has been initiated. Since this not equal to the $AF$, a second step is needed where an offset-parameter is introduced that converts the "fraction of particles older than $t_{switch}$" into the $AF$ (=fraction of particles older than $t_{act}$).

The first step can be achieved by integrating the RTD backwards starting from $t = t_{switch}$. This is an unfavorable approach since it is not compatible with a constantly increasing experimental duration $t$. This can be avoided, by flipping the RTD horizontally at $t = t_{switch}$ and integrating forward in time from $t = t_{switch}$ to $t$, which is done in eq. (9). For a simpler integration the experimental duration $t$, was normalized by dividing it by the hydrodynamic residence time $\tau_{CSTR}$.

$$AF(t)_{flushing} = \int\limits_{t_{switch}/\tau_{CSTR}}^{t/\tau_{CSTR}} e^{\left(\frac{t-2\cdot t_{switch}}{\tau_{CSTR}}\right)} d\left(\frac{t}{\tau_{CSTR}}\right) \tag{9}$$

As mentioned before, eq. (9) only describes the fraction of particles that are older than $t_{switch}$. Since we defined $AF$ as the fraction of particles with an age above the threshold time $t_{act}$, eq. (9) describes $AF$ only if $t_{act} = t_{switch}$ holds true. To determine $AF$ for conditions when $t_{act} < t_{switch}$ ($AF(t=t_{switch}) > 0$) or for a delayed activation, $t_{act} > t_{switch}$ ($AF(t=t_{switch}) = 0$), an additional parameter has to be introduced. This parameter is an offset of the $AF$-curve along the time-axis and is therefore called $t_{offset}$. Taking $t_{offset}$ into account, eq. (10) can be obtained after integrating eq. (9).

$$AF(t)_{flushing} = e^{\left(\frac{t+t_{offset}-2\cdot t_{switch}}{\tau_{CSTR}}\right)} - e^{\frac{-t_{switch}}{\tau_{CSTR}}} \tag{10}$$

The parameter $t_{offset}$ is initially unknown and has to be calculated. For this we need to differentiate between two cases. First, if $t_{act}$ is larger than $t_{switch}$ and therefore $AF$ is 0 at the switching point, $t_{offset}$ can be obtained by subtracting $t_{act}$ from $t_{switch}$ (eq. 11a). Second, if $AF$ at $t = t_{switch}$ is above 0, $t_{offset}$ has to be calculated by solving eq. 10 for $t_{offset}$. For this $AF(t)_{flushing}$ has to be set to $AF(t=t_{switch})$ and eq. 10 has to be rearranged as shown in eq. (11b).

$$AF(t=t_{switch}) = 0 \qquad\qquad t_{offset} = t_{switch} - t_{act} \tag{11a}$$

$$AF(t=t_{switch}) > 0 \qquad\qquad t_{offset} = \ln\left(AF(t=t_{switch}) + e^{\frac{-t_{switch}}{\tau_{CSTR}}}\right) \cdot \tau_{CSTR} + t_{switch} \tag{11b}$$

### 3.1.4 Particle activation throughout an entire CSTR experiment

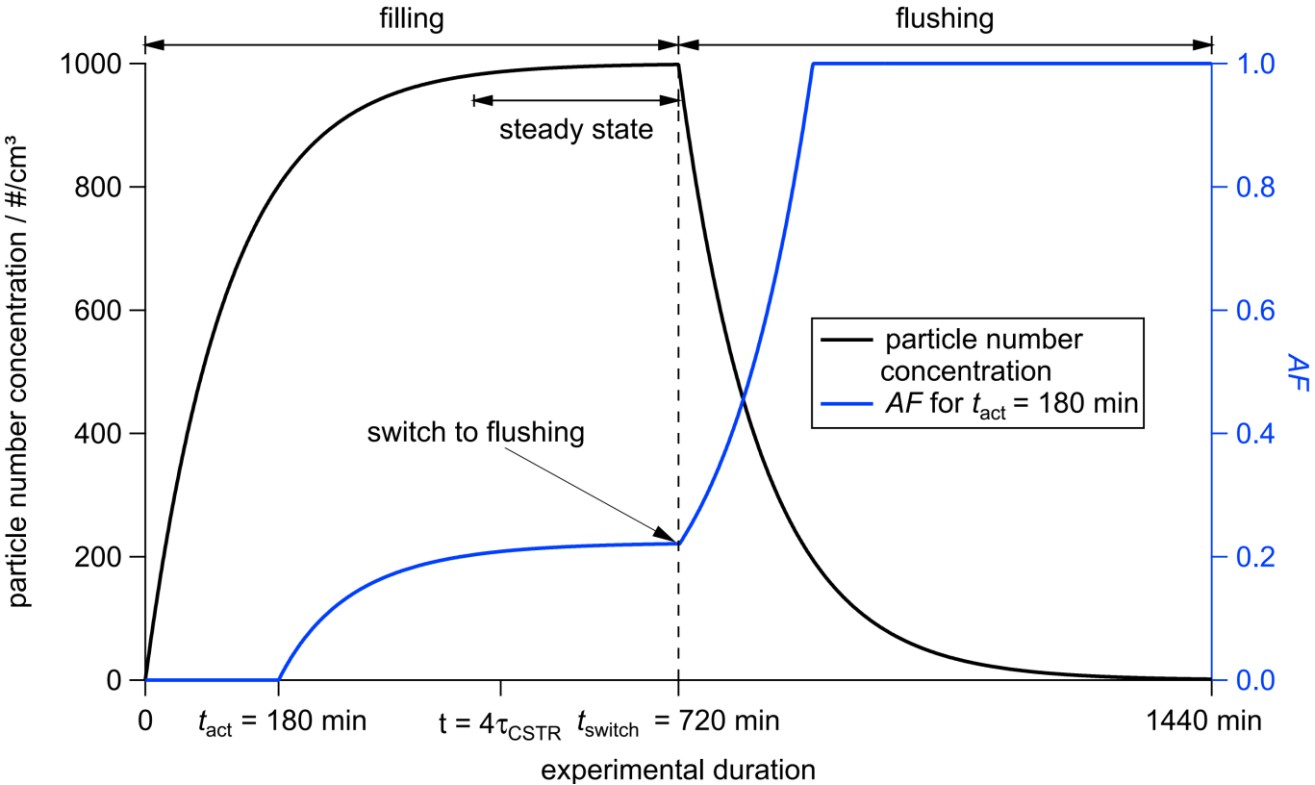

Figure 3: Calculated change of the activated fraction ($AF$; blue line, right axis) and particle number concentration (black line, left axis) throughout filling regime, steady state, and flushing regime based on a CSTR experiment with $\tau_{CSTR} = 120$ min,

$t_{act}$ = 180 min, $t_{switch}$ = 720 min, and an input particle number concentration of 1000 cm$^{-3}$. The different operation regimes are indicated on the top of the figure.

The particle number concentration inside the CSTR can be calculated throughout all operation regimes using equations (2) and (5). *AF* can be calculated using equations (6) and (10). Figure 3 shows how the particle number concentration (black line, left axis) and *AF* (blue line, right axis) change during an experiment with $\tau_{CSTR}$ = 120 min, $t_{act}$ = 180 min, $t_{switch}$ = 720 min, and [A$_{feed-in}$] = 1000 cm$^{-3}$. A summary is presented in Table 1.

**Table 1: Evolution of the aerosol particle number concentration and the *AF* in different CSTR-regimes.**

|  | experimental duration | *AF* inside CSTR | aerosol particle number concentration |
|---|---|---|---|
| **Filling regime** | 0 min (start) to 480 min ( 4$\tau$ - criterion) | *AF* = 0 from start till $t_{act}$. Increasing *AF* (asymptotically approaches a constant value) after $t_{act}$. | Increasing particle number concentration, that asymptotically approaches a constant value. |
| **Steady state** | 480 min ( 4$\tau$ - criterion) to 720 min ($t_{switch}$) | Stable *AF* at 0.221. | Stable aerosol particle number concentration (changes below detection limit) |
| **Flushing regime** | 720 min ($t_{switch}$) to 1440 min (end) | Exponentially increasing *AF* until 1 | Exponentially declining aerosol particle concentration. |

### 3.2 Introducing the activation time distribution *P(t$_{act}$)*

The approach discussed so far is based on the assumption that all aerosol particles are identical and therefore a specific property of the whole aerosol population can be described with a single parameter. In other words, all particles have the same tact in case of CCN activity being the specific property. However, this is not the case for many parameters. In case of the particle diameter, for example, every aerosol particle has its individual diameter and the total population can be described by a distribution of particle diameters around a mean diameter. An eventual size-selection does impact the mean diameter and the width of the distribution. Still, the size selected particles will not have the identical diameter. Furthermore the aerosol population might be mono-modal and narrowly-distributed with respect to one parameter such as the aerosol particles electrical mobility diameter, but it can be multi-modal or broader distributed with respect to another parameter e.g. the aerodynamic diameter.. Therefore, it has to be expected that the activation time ($t_{act}$) is also characterized by a distribution. For this we introduce the activation time distribution *P(t$_{act}$)* and discuss its theoretical impact on transitions within binary systems in CSTR-experiments. In contrast to a uniform $t_{act}$ valid for all particles the activation time distribution *P(t$_{act}$)* is more realistic as it takes the individual $t_{act}$ of the individual particles within the population into account. Nevertheless, only one value for *AF* can be determined experimentally. This single value, from now on referred to as global *AF*, represents the average *AF* over all *AF*s of the individual sub-fractions within the population as will be explained in more detail in the upcoming sections.

### *3.3* Impact of the activation time distribution on the individual *AF*

In section 3.1.4 we showed how *AF* evolves throughout a whole CSTR-experiment (blue curve in Fig. 3). This curve was calculated based on the assumption of uniformity, i.e. every aerosol particle that is older than $t_{act}$ = 180 min is CCN active. While, this assumption can be valid for some conditions it surely cannot be representative for all real-world conditions. To discuss the impact of an activation time distribution *P(t$_{act}$)* on the evolution of *AF* in a CSTR we consider a model system with *P(t$_{act}$)* representing a Gaussian distribution with an exemplary mean ($\mu$) of 180 min and an exemplary standard deviation ($\sigma$) of 30 min (eq. (12)).

$$P(t_{act})= \frac{1}{\sqrt{2\pi\sigma^2}} e^{-\left(\frac{(t_{act}-\mu)^2}{2\sigma^2}\right)} \tag{12}$$

For simplicity we discuss the impact of the activation time distribution ($P(t_{act})$) in steady state first but the concept is the same throughout the entire experiment including the filling as well as the flushing regime. In a CSTR particles with different individual residence times are present at the same time due to the continuous feed in of fresh particles and the active mixing. For a better understanding the "individual residence time of a particle" will be referred to "particle age" from here on. With

an increasing particle age the number of particles (=area under the curve) declines in a CSTR during steady state (Graph A, Fig. 4). Nevertheless, the activation time distribution $P(t_{act})$ is the same for all particles (Graph B, C, D in Fig. 4), regardless of their age. The fraction of activated particles inside the CSTR (global *AF*) therefore has to be described as an overlap of the residence time distribution (RTD; black curve in Fig. 4.A) and the activation time distribution ($P(t_{act})$; red curves in Fig. 4.B, C, and D).

In Fig 4.A, three individual sub-populations are indicated by red vertical bars. The first sub-population at $t = 60$ min $= 0.5\ \tau$ consists of rather young and fresh particles. Their corresponding activation time distribution $P(t_{act})$ is shown in sub-panel B. While there are a lot of particles (indicated by the large area under the red curve) only a small fraction of these particles is CCN-active. The active fraction is indicated by the red colored area and corresponds to the particles with a very low individual $t_{act}$. The contribution of this sub-population to the global *AF* is therefore small. In sub-panel D, a sub-population at

$t = 360$ min $= 2.0\ \tau$ of old and well-aged particles is shown. Due to their high age, litterally all particle in this sub-population are CCN-active as their individual particle age is more than 6 sigma beyond the mean value of the exemplarily discussed Gaussian activation time distribution $P(t_{act})$. Since the overall fraction of these old particles is low as indicated by the significantly reduced area underneath the red curve compared to the first sub-population (panel B), they contribute only little to the global *AF*. In sub-panel C, a medium aged sub-population at $t = 180$ min $= 1.5\ \tau$ is shown. On the one hand, there are

significantly less particles than in the first sub-population (panel B), which is indicated by the reduced area underneath the red curve. On the other hand, the fraction of CCN active particles within this sub-population is significantly larger than in the first one. In fact, the fraction is 0.5 as this sub-population has an individual residence time that is equal to the mean ($\mu$) of $P(t_{act})$.

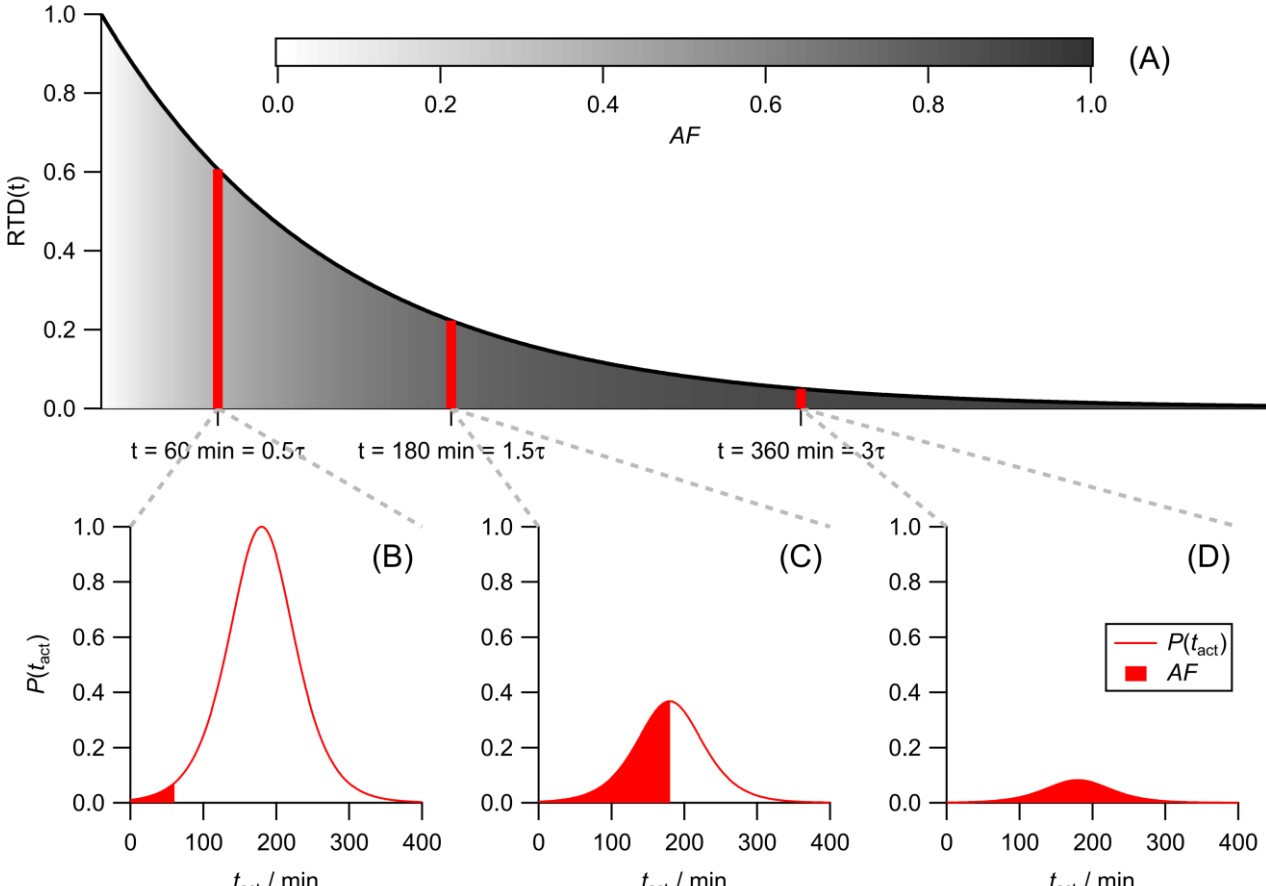

**Figure 4: (A):** The RTD in steady state is shown (black line). The grey shaded area underneath the curve represents the increasing fraction of activated particles with increasing particle age.

**(B-D):** Activation time distribution at different particle ages. The total area under the activation time distribution curve (red line) represents the relative abundance of particles at their specific age, which is equal to the area covered by the respective red bar in panel A. The red-colored area represents the fraction of activated particles within the population of this particular particle age.

### 3.4    Calculation of the total activated fraction (global *AF*)

The global *AF* at any point in time can be calculated by multiplication of $AF(t_{act},t)$ for each individual $t_{act}$ with the relative abundance of particles (obtained from $P(t_{act})$) with this respective $t_{act}$ and integration over the whole range of possible $t_{act}$'s. (lower limit: $t_{act} = 0$ min; upper limit: $t_{act} = t$)

$$AF(t) = \int_{t_{act}=0}^{t_{act}=t} AF(t_{act},t) \cdot P(t_{act}) \ dt_{act} \tag{13}$$

In Fig 5 the differing evolutions of the global *AF* in the case of two different $P(t_{act})$ within a CSTR ($\tau_{CSTR} = 120$ min) is presented. The blue curve is the same as in Fig. 3 postulating all particles activate uniformly at $t_{act} = 180$ min ($AF_{step}(t)$, $P_{step}(t_{act})$). The red curve shows the global *AF* for a Gaussian shaped activation time distribution as discussed in the previous section and displayed in Fig. 4 with $\mu = 180$ min and $\sigma = 30$ min ($AF_{gaussian}(t)$, $P_{gaussian}(t_{act})$). While the uniform scenario shows no activity before reaching $t_{act}$, the Gaussian distribution scenario shows an earlier activation onset. This is because there are some particles in the population that activate earlier than the mean activation time. These are all particles within the red area left of $\mu = 180$ in Fig. 4.B to D. Both curves reach a constant global *AF* during steady state, but in the Gaussian distribution scenario the *AF* is higher ($AF_{gaussian} = 0.242$ vs $AF_{step} = 0.221$). In our specific case $AF_{gaussian}$ is higher than $AF_{step}$, but the actual difference between these two values is dependent on the specific values of $\tau_{CSTR}$, $t_{act}$, $\mu$ and $\sigma$. In the flushing regime the global *AF* grows exponentially in both scenarios. Within the Gaussian activation time distribution ($P_{gaussian}$) there are particles

activating later than the average $t_{act}$ of 180 min. These particles are represented by the area right of $\mu$ under the red curve as shown in Fig 4.B to D. Therefore, in the Gaussian distribution scenario (red curve in Fig 5) full activation is reached later than in the uniform scenario (blue curve in Fig 5). Generally speaking, a broader $P_{gaussian}(t_{act})$ leads to an earlier onset of $AF$ while full activation is reached later because the broader distribution extends over a wider range of individual $t_{act}$'s on the single particle level.

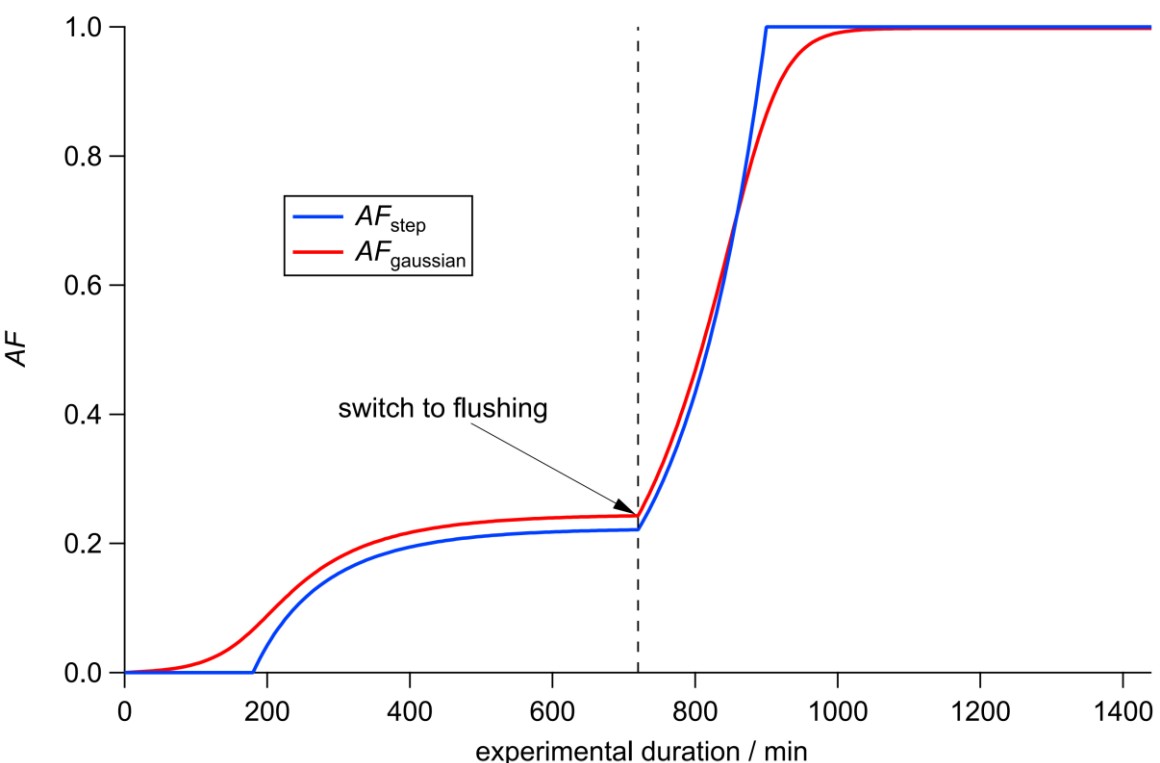

**Figure 5: Global $AF$ response functions inside a CSTR for a uniform aerosol population (blue line) and an aerosol population with an activation time distribution $P(t_{act})$ represented by a Gaussian distribution (red line).**

### 3.5     Equivalent parameters $t_{act}$ – onset and $t_{act}0.5$ vs $t_{act}$

In literature, different parameters are used to describe the CCN activity of particles. Results from batch chamber experiments as well as from oxidation flow reactor experiments are often presented in terms of SS-onset or critical SS. While the SS-onset is defined by a minimum threshold (e.g. 0.01 $AF$) the critical SS is reached when 0.5 of the particles activate (Friedman et al., 2011; Koehler et al., 2009; Rose et al., 2007). From this perspective, $t_{act}$ is a third parameter. Further we present the parameters $t_{act}$-onset and $t_{act}0.5$. Following the aforementioned nomenclature in the CCN community, we define $t_{act}$-onset as the time when the global $AF$ inside the CSTR crosses a defined value, here $AF = 0.01$. Opposite to this $t_{act}0.5$ does not refer to the global $AF$ that can be determined experimentally but we define it as the time after which 0.5 of the particle within the activation time distribution is activated (Fig. 4.C). In Table 2 the three parameters are compared for the two scenarios of a uniform $t_{act}$ ($P_{step}$) and an activation time distribution ($P_{gaussian}$), respectively.

**Table 2: Influence of the $t_{act}$-distribution on $t_{act}$-onset, $t_{act}$, and $t_{act}0.5$.**

| Parameter | Reference population | $P_{step}(t_{act})$ | $P_{gaussian}(t_{act})$ |
|---|---|---|---|
| **$t_{act}$-onset** | global $AF = 1$ % | 185 min | 87 min |
| **$t_{act}$** | global $AF$ in steady state (Fig. 5; eq. (8)); | 180 min | 170 min |
| | | $AF_{step} = 0.221$ | $AF_{gaussian} = 0.242$ |
| **$t_{act}0.5$** | Obtained from $P(t_{act})$ | 180 min | 180 min |

As can be seen in Table 2, the individual values deviate with the biggest deviation in the case of $t_{act}$-onset. However, the presented deviations are solely caused by the underlying distributions of the activation time. In addition, $t_{act}$-onset, $t_{act}$, and $t_{act}0.5$ are determined at different experimental times. While $t_{act}$-onset is directly determined by measuring the entire particle population within the CSTR (global $AF$), $t_{act}$ is calculated from the global $AF$ in steady state and $t_{act}0.5$ is obtained from the activation time distribution itself. In the case of $t_{act}$-onset, there is a significant share of particles activating significantly earlier than the nominal activation time ($\mu = 180$) in the case of a Gaussian distribution. Therefore, a fraction of 0.01 of CCN active particles within the entire particle population is already present after 87 min. Opposite to this, the threshold value of 0.01 is crossed later than the nominal activation time in the case of the step distribution. This is because even though every single particle activates after exactly 180 min of individual aging time, it takes some additional time before a fraction of 0.01 of the entire particle population within the CSTR is older than 180 min leading to a $t_{act}$-onset of 185 min. The difference of 10 min in $t_{act}$ between the two $P(t_{act})$-approaches is due to the application of eq. (8) which allows for the calculation of $t_{act}$ from the global $AF$ in steady state. Strictly speaking, this equation is defined for the ideal step function ($P_{step}(t_{act})$) only. Therefore the higher global $AF$ value for $P_{gaussian}(t_{act})$ in steady state has to lead to a lower $t_{act}$ value compared to $P_{step}(t_{act})$. Note, $t_{act}0.5$ is referring to the particle activation distribution $P(t_{act})$ only leading to a concordant value of 180 min in both cases. This can be seen in Graph C of Fig. 4, where 0.5 of the particles with a residence time equal to the nominal activation time are activated in the case of a Gaussian distribution corresponding to $t_{act}0.5$. In the case of a step function, all particles are activated once the respective particle population is older than $t_{act}$. In the following we will show how the actual activation time distribution $P(t_{act})$ can be retrieved from real CSTR experimental data.

## 4    Application of the new $t_{act}$-concept to experimental data from CSTR-aging experiments

In the laboratories at ETH Zurich we performed aging experiments in a 2.78 m$^3$ stainless steel aerosol chamber operated in CSTR mode. A detailed description of the chamber can be found in Kanji et al., (2013). The chamber was actively mixed with a fan, but had no further features to enhance mixing e.g. baffles. All instruments were connected to the chamber with stainless steel tubing with 4 mm inner diameter. Since the maximal tubing length from the aerosol chamber to the analysis instruments was 3 m at flow rates were between 2 and 5 lpm the impact on the overall residence time (0.45 to 1.13 s) is negligible compared to an average residence time on the order of hours within the chamber.

We investigated the change in CCN-activity of soot particles rich in organic carbon due to heterogeneous ozone oxidation. The soot particles were generated with the miniature Combustion Aerosol STandard (miniCAST, Model 4200, Jing Ltd., Zollikofen, Switzerland) which is propelled with propane and operates with a laminar diffusion flame. The miniCAST was operated under fuel-rich conditions (set point 6 according to the manual) in order to generate a soot which was rich in organic compounds (fuel-to-air ration: 1.03).

The particles were size selected at 100 nm by a Differential Mobility Analyzer (DMA). These size selected aerosol particles were diluted with particle-free and VOC-filtered air in order to achieve a constant aerosol flow of 25 lpm with a particle concentration of ~1200 cm$^{-3}$. The aerosol flow was fed into the aerosol chamber, where a constant ozone background concentration of 100 and 50 ppb, respectively, was maintained throughout the entire experiment. Downstream of the aerosol chamber the CCN-activity was measured with a Cloud Condensation Nuclei Counter (CCNC; Roberts and Nenes, 2005) and the size distribution data was acquired by a Scanning Mobility Particle Sizer (SMPS) system from which the total particle number concentration was derived. In Fig. 6 data for two experiments conducted on two different days are shown. While there is no difference in the experimental instrumentation, the two data sets differ by the SS conditions set in the CCNC and the ozone background concentration (A: 1.0 %, 100 ppb; B: 1.4 %, 50 ppb). The data was analyzed focusing on the three following aspects:

   1) Can the aerosol chamber be operated in CSTR-mode throughout an entire day, which requires a constant aerosol feed-in flow and a good internal mixing?

2) Can the change in CCN-activity of soot particles due to oxidation with ozone be investigated with CSTR-mode aging experiments?
3) Can $t_{act}$ and its distribution ($P(t_{act})$) be retrieved from experimental data?

The graphs A1 and B1 in Fig. 6 show the particle concentration (black crosses; left axis), the measured global *AF* (red crosses) and the fitted global *AF* (blue dashed line, both right axis). The particle number concentration curves (black crosses) follow the theoretical filling and flushing curves as expected in a CSTR (Fig. 3). The slight decline in the concentration in steady state in graph A1 is due to a slight reduction in the particle input concentration that was experienced during the experiment. Vice versa the slight increase in the number concentration in graph B1 is due to a slight increase in the particle input concentration over time.

In the flushing regime the particle number concentration declines exponentially in both experiments. Eq. (5) describes the ideal/theoretical evolution of the particle number concentration in the flushing regime when taking the hydrodynamic residence time $\tau_{CSTR}$ according to eq. 1 into account. In the ideal case the decay is solely caused by the flushing process. In reality, the decay is a combination of flushing as well as additional particle losses e.g. wall losses or coagulation. Therefore, the real residence time can be obtained by fitting equation 5 to the experimental data after rearrangement for $\tau$, to which we refer to as $\tau_{flush}$ from now on (Kulkarni et al., 2011). In both experiments $\tau_{flush}$ coincides at 104 min, which is lower than the hydrodynamic residence time $\tau_{CSTR}$ of 111 min. In other words, the particle concentration declines faster than expected. This difference is caused by particle losses to the chamber wall, which acts as an additional particle sink parallel to flushing and reduces the particle lifetime. Nevertheless, statistical analysis of the experimental data results in purely statistical noise centered on the fitting curve used to determine $\tau_{flush}$. This indicates that in terms of mixing no difference between an ideal CSTR and the aerosol chamber used here can be detected with the applied instrumentation.

When dividing the real particle life time ($\tau_{flush}$) into its individual components, a particle life time upon wall losses ($\tau_{wall\text{-}loss}$) of 1600 min can be determined in accordance with first order wall loss kinetic (Crump et al., 1982; Wang et al., 2018). The influence of particle coagulation can be considered negligible due to the low coagulation rate of 100 nm particle at concentrations of maximum 1500 cm$^{-3}$ (Kulkarni et al., 2011).

Based on the discussion above, the measured *AF*s (red crosses) show the expected change throughout the entire experiment in Fig 6 A1 and B1. In the beginning of both experiments *AF* is 0. After a minimum aging time each *AF* starts to increase until it reaches a constant level (A1: *AF* = 0.091, 1.0 % SS; B1: *AF* = 0.233, 1.4 % SS). The gaps in the curves during steady state are due to changes in the operation of the CCNC form running on a constant SS (1.0% and 1.4%, respectively) to scanning over a range of SS. In the flushing regime, each measured *AF* increases exponentially. CCN data could be acquired successfully throughout the entire experiment until the global *AF* reached ~1.0 (> 1000 min) in the first experiment presented in graph A1. In the second experiment presented in graph B1, instrumental issues caused the acquisition of the global *AF* to end prematurely after approx. 800 min of experimental duration.

The graphs A2 and B2 in Fig. 6 show the activation time distribution $P(t_{act})$ (blue solid line) retrieved from the measured global *AF*s. The $P(t_{act})$'s presented were obtained from curve fitting the measured *AF*-curves using eq. (13), which describes the evolution of *AF* taking the activation time distribution into account. For this, assumptions concerning the type of distribution had to be made. Here, we assumed that $P(t_{act})$ can be described by a mono-modal Gaussian distribution as presented in eq. (12). A brute-force algorithm was used that optimized the characteristic parameters $\mu$ (=mean) and $\sigma$ (=standard deviation) in order to achieve the best fit to the measured global *AF* using the least-square method. The results of this fitting procedure are presented in Table 3 as well as in A2 and B2 of Fig. 6. In the first experiment with the experimental settings at 1.0 % SS and 100 ppb O$_3$ $\mu$ as well as $\sigma$ of $P(t_{act})$ are larger (253.7 min and 35.5 min) compared to the results obtained for the second experiment at 1.4 % SS and 50 ppb O$_3$ (153.6 min and 24.6 min). From a theoretical perspective, there are two competing aspects. On the one hand, due to the higher ozone concentration the threshold of chemical transformation leading to CCN activity of the particles should be reached earlier. On the other hand, the threshold of chemical transformation should be lower

at higher SS. Our results presented here could indicate that the difference in SS in this specific range might be more important than the difference in ozone background concentration within the considered range. At the current stage we cannot draw any final conclusions on how these two competing aspects actually interplay but additional experiments are planned to resolve this issue.

In addition, we list $t_{act}$ obtained from $AF$ during steady state following eq. (8) as described in section 3.1.2 in Table 3. Based on error propagation calculation, the instrumental uncertainty for obtaining $t_{act}$ from steady state is $\pm 11.6$ min. In our experimental setup the differences between $t_{act}$ and $\mu$ are 3.9 min and 2.1 min, respectively, and therefore below the instrumental uncertainties. This is a very beneficial aspect when considering a broad application of the CSTR-concept in atmospheric science experiments. In general, an accurate determination of $P(t_{act})$ requires a sufficiently high time resolution

throughout the whole experiment. This can be difficult to achieve depending on the general experimental conditions such as the type of instrument, since running SS-scans with a CCNC can be time consuming. However, if a characterization of the aged aerosol during steady state is sufficiently precise, a potentially time consuming acquisition of a large number of data points for the determination of $P(t_{act})$ does not provide additional benefits.

     As a cross check, we implemented the activation time distributions determined from the experimental data into eq. (13), which

allows for the determination of $AF$ throughout the entire experiment. The results are presented in Fig. 6 panel A1 and B1, respectively. While the calculated $AF$ (blue dashed line) and the measured AF (red crosses) superimpose in the early filling regime, the steady state, and in the flushing regime, some deviation can be recognizes in the period when $AF$ increases. This deviation is caused by the slight changes in particle number concentration during stead state, which reflects a change in the particle input concentration. Therefore, this cross check represents an addition way to inspect the experimental conditions

throughout the experiment revealing potential deviations from ideality. Overall this section demonstrates the applicability of the CSTR concept for aerosol experiments that require long experimental durations.

**Table 3 Comparison of $t_{act}$ and $t_{act}0.5$ for both experiments.**

| Parameter | A: 1.0 % SS / 100 ppb O₃ | B: 1.4 % SS / 50 ppb O₃ |
|---|---|---|
| $t_{act}$ | 249.8 min | 151.5 min |
| $t_{act}0.5 / \mu$ | 253.7 min | 153.6 min |
| $\sigma$ | 35.5 min | 24.6 min |


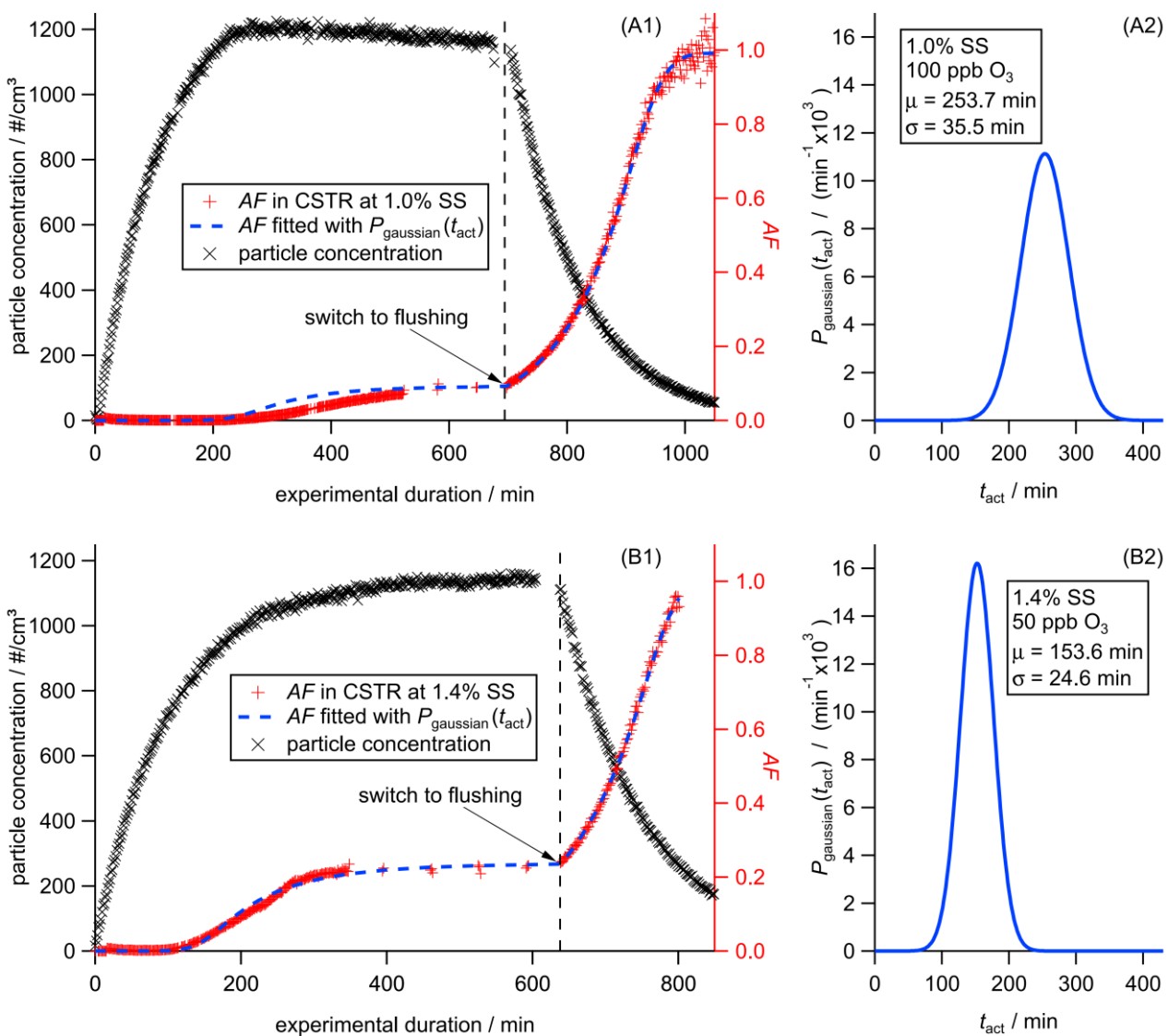

**Figure 6: Particle number concentration (black crosses, left axis), measured global *AF* (red crosses, right axis), and fitted global *AF* (blue dotted line, right axis) are presented in panel A1 and B1, respectively. The experimentally determined activation time distributions $P(t_{act})$ are shown in panel A2 and B2, respectively.**

## 5    Application of $t_{act}$ to other continuous flow aerosol chambers

The application of oxidation flow reactors (OFRs) in atmospheric science became increasingly popular. These chamber types are designed to generate an aged aerosol that is as homogenous as possible. The ideal OFR would be an ideal plug flow reactor (PFR) with a RTD being a Dirac-Delta-function, often referred to as impulse-function. However, all OFRs have a RTD that lies between an ideal CSTR and an ideal PFR and is further dependent on the individual design of the OFR (George et al., 2007; Huang et al., 2017; Ihalainen et al., 2019; Kang et al., 2007; Simonen et al., 2017). Lambe et al., (2011) already suggested that the RTD reduces the comparability of results from different OFR-types. In the following we discuss the applicability of the $t_{act}$-concept introduced here to real continuous flow aerosol chambers like OFRs which entail partial mixing.

Besides various home-build OFRs (e.g., (Ezell et al., 2010; Huang et al., 2017; Keller and Burtscher, 2012), the commercially available Potential Aerosol Mass Chamber (PAM, Aerodyne; Kang et al., 2007) is an instrument widely used for the investigation of aerosol aging within the atmospheric science community. All these chambers have in common that an aerosol flow is exposed to OH-radicals. OH-radicals are typically produced by UV irradiation of ozone causing the production of excited oxygen atoms [O(D[1])] which react with water vapor. In OFRs the OH-concentration tends to be significantly higher than the average atmospheric concentration in order to mimic several days of atmospheric aging within a few minutes of

experimental duration. For intercomparison amongst chambers and for extrapolation to atmospheric conditions the total OH-exposure is used as a metric, which is often calculated by multiplying the OH-concentration with the exposure time. The exposure time is hereby equal to the residence time within the OFR which can be calculated the same way as in the CSTR-concept following eq. (1).

In the case of an intercomparison of the Toronto Photo-Oxidation Tube (TPOT; George et al., 2007) and the PAM chamber presented by Lambe et al., (2011), the OH-exposure was determined from $SO_2$-oxidation experiments following eq. A1 therein. Amongst other parameters, they investigated the secondary organic aerosol (SOA) formation from volatile organic compounds (VOCs) as well as the impact of heterogeneous oxidation on the CCN-activity of bis(2-ethylhexyl)sebacate (BES) particles. They found very good agreement concerning the average H/C and O/C ratios of the SOA particles produced *insitu* and the

BES-particles. This indicates that the reaction with OH-radicals follows the same kinetic in both chambers. However, the CCN-activity of BES-particles aged in the PAM chamber is reported to be significantly higher than in the TPOT chamber at low OH-exposure levels and vice versa at high OH-exposure levels as can be seen in Fig A2(a) in the respective publication (Lambe et al., 2011). The authors identify differing residence time distributions between the two chambers and suggest this to results in a difference in chemical composition that is not captured by the average H/C and O/C ratios but to result in the

deviation in CCN-activity. In addition to major improvements in terms of operating the PAM chamber as well as in terms of analysis of PAM chamber data within the last couple of years, a range of modeling and experimental studies have been published investigating this specific aspect (e.g. Mitroo et al., 2018). In the following we show that the application of $t_{act}$ can contribute significantly to the explanation of the aforementioned discrepancies in terms of CCN-activity of the BES particles. In Fig. 7 we show the RTDs for 145 nm BES particles using the parameters for the bimodal Taylor-dispersion model given by

Lambe et al., (2011) in Appendix A4 (Fig A3). We normalize the area under the curve to be one causing the area under each curve to be directly proportional to the *AF*s for a better visual comparison. Here, PAM chamber data is indicated by the dotted line/green area and TPOT chamber data is indicated by the dashed line/blue area. As can be seen, the two curves are not perfectly superimposed with the peak of the PAM chamber RTD being earlier than in the TPOT chamber RTD followed by a steep decline causing the two curves to cross at approximately 40 s. Overall the PAM chamber RTD (dotted line) shows a

stronger dispersion causing the two lines to cross again at approximately 180 s.

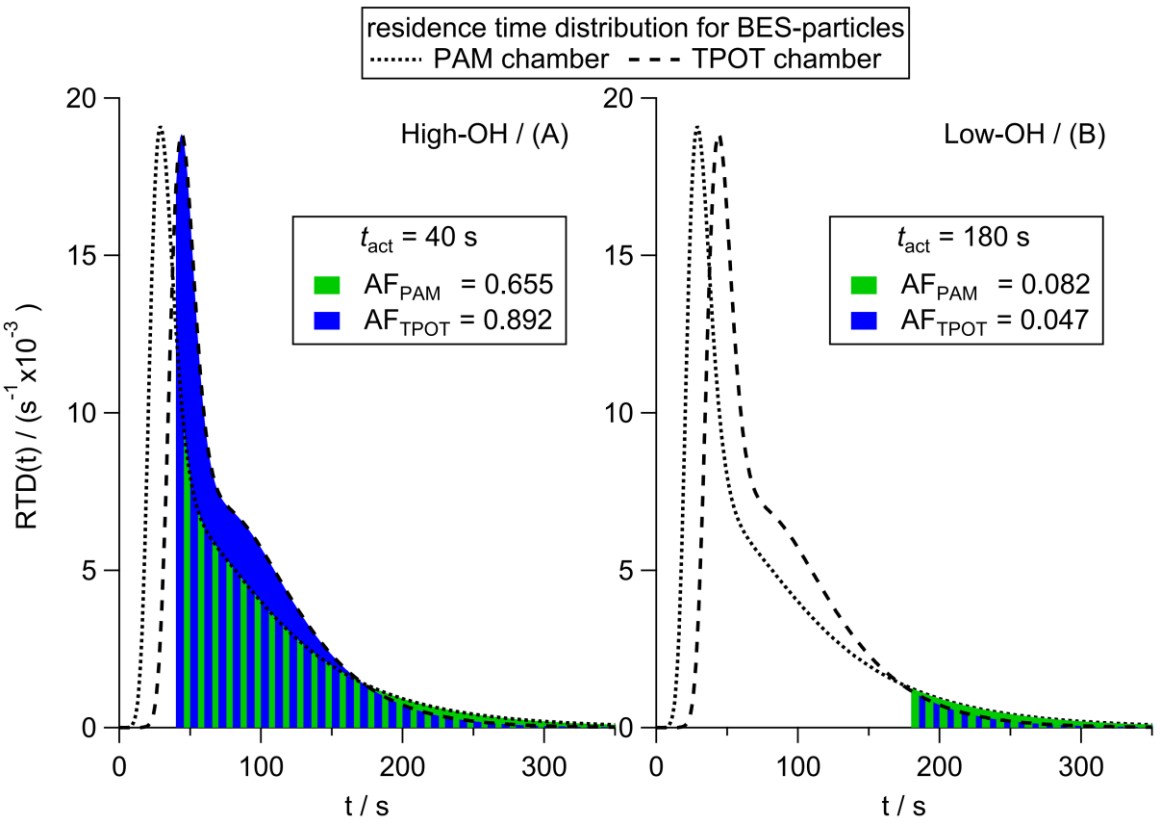

**Figure 7: Global *AF* in the PAM and TPOT chamber for $t_{\text{act}}$ of 40 and 180 s, respectively.**

Assuming a high OH-concentration leads to a higher reaction speed and therefore shorter $t_{\text{act}}$ we present two scenarios. Scenario A representing a high OH-concentration is based on a $t_{\text{act}}$ of 40 s (Fig. 7 A). Scenario B representing a low OH-concentration is based on a $t_{\text{act}}$ of 180 s (Fig. 7 B). In both cases the BES-particles show CCN-activity, but the global *AF* differs significantly

between both chamber types. In the high-OH scenario A, the TPOT chamber is more efficient in producing CCN-active BES-particle ($AF_{\text{TPOT}} = 0.892$; blue area) than the PAM chamber ($AF_{\text{PAM}} = 0.655$; green area) as can be perceived by the blue area being larger than the green area in the left panel. Opposite to this, the PAM chamber is more efficient ($AF_{\text{PAM}} = 0.082$) than the TPOT chamber ($AF_{\text{TPOT}} = 0.047$) in case of the low-OH scenario B, as can be seen in the right panel. These calculations indicate how the new $t_{\text{act}}$ concept can contribute to the understanding and interpretation of experimental data that has been

acquired in non-CSTR reaction chambers. At the same average OH-exposure, aging in different OFRs causes the same global *AF* only if the RTDs are the same. Since the RTDs of the PAM chamber and the TPOT chamber are not the same, the same global *AF* can only be obtained if the $t_{\text{act}}$'s differ. Three examples of how $t_{\text{act}}$ has to deviate between the PAM chamber and TPOT chamber to lead to the same global *AF* are given in the supplement.

While some parameters such as *AF* can differ between two OFR chambers, other parameters can still agree very well. Such

parameters could be the average H/C and O/C-ratios measured with an aerosol mass spectrometer (AMS). The chemical modification of an aerosol particle is a continuous process and the CCN-activity is a function of this continuous chemical modification as discussed in section 3. However, once a certain modification threshold is reached no further increase in the CCN-activity of a single aerosol particle can be achieved at a constant SS, even if the chemical aging proceeds. Therefore, *AF* does not correlate linearly with the average OH-exposure and the individual aging degree of a single particle, but with the

fraction of particles older than a certain $t_{\text{act}}$. However, if the H/C and O/C-ratio would be measured on a single particle level, a distribution of chemical properties would be recorded similar to the RTD of the respective chamber.

Up to now, the discussion did not include many important processes that are relevant in aging chambers e.g. particle wall-interaction, gas-phase-partitioning, fluctuating input concentrations while field measurements, or inhomogeneities inside the

OFR. These aspects are important for many processes such as the formation of SOA and can be incorporated to the $t_{\text{act}}$-concept by modifying eq. (13). As the actual calculation requires a multidimensional data array and detailed knowledge about the chamber of interest, this subject matter is beyond the scope of this publication and will not be discussed further. Nevertheless, the overall conclusion is that application of the original/non-adjusted $t_{\text{act}}$-concept can explain why measurements within different OFR chambers agree in parameters, which dependent on the bulk properties of the aerosol particle population (e.g.

average O:C ratio) and at the same time disagree in parameters, which are dependent on the condition/status of the individual particle (e.g. CCN-activity). Therefore, we suggest to apply the concept of the activation time $t_{\text{act}}$ or the activation time distribution $P(t_{\text{act}})$ as metric in addition to calculating average values, such as the global *AF* and OH-exposure if following conditions are met. One, the system or parameter of interest can be described as a binary system and undergoes step-wise / non-gradual transitions such as CCN-activity. Two, the OFR used has a RTD broad enough to influence the outcome. Three,

the conditions inside the reactor are either homogeneous or a correction for inhomogeneities (e.g. different oxidants concentrations inside the reactor) is implemented.

## 6    Conclusion

This work investigates the potential of aerosol chambers to be operated in Continuous flow Stirred Tank Reactor (CSTR) mode for simulating atmospheric aging processes of aerosol particles and retrieve data that is comparable to other methods. This

approach was motivated by the possibility to achieve longer aging times if the same chamber was operated in CSTR-mode instead of in batch-mode while at the same time a significantly lower aerosol number concentration was required enabling the

use of e.g. size selected aerosol particles. One of the main obstacles to implement the CSTR concept in atmospheric science has been hitherto consideration of the residence time distribution in data analysis. Inside a CSTR chamber the particle population consists of particles at different aging stages. In order to address this, we introduced the activation time $t_{act}$ as a new parameter to disentangle the non-uniform aerosol population and generate data that is comparable to that acquired in other

experimental setups. This concept was developed based on the assumption that continuous aging processes on the level of single particles can lead to a step-wise change in individual particle properties referred to as non-gradual transitions. The new parameter $t_{act}$ describes the time needed to reach this transition. On a more fundamental level, $t_{act}$ only requires a time-dependent change of a single particle property that can be used to distinguish between two states, below and above a defined threshold in a binary system. Since particle properties are typically distributed around a mean value, we also introduced the

activation time distribution $P(t_{act})$. The impact of $t_{act}$ and $P(t_{act})$ on the parameters measured downstream a CSTR aerosol chamber was simulated with the newly developed mathematical framework and compared to experimental data. Data presented herein was acquired from experiments on soot particles transitioning from initial CCN-inactivity to CCN-activity over the course of several hours due to ozone exposure. We show that our theoretical concept describes the observed changes in the CCN-activity very well. Additionally, we show that the discrepancy between $t_{act}$ and $t_{act}0.5$ is lower than the instrumental error

in the model system in CSTR mode. Therefore, the data acquired during steady state is representative for the whole particle population. Finally, we generalize this concept and apply it to data of two OFR aging chambers (Lambe et al., 2011) that are operated in steady state as well but are characterized by none ideal internal mixing. Through the application of our new $t_{act}$-concept we can explain qualitatively why the results from the PAM chamber and TPOT chamber agree for some parameters (bulk O/C and H/C-ratio) but show significant differences for other parameters (CCN-activity). We recommend re-analysis of

other OFR data to gain further insight to non-gradual transformation processes.

*Author contribution.*

FF and AAM wrote the manuscript and designed and carried out the experiments. FF developed the analysis concept and carried out the data analysis.

*Acknowledgements.*

This research was funded by the Swiss National Science Foundation; SNSF-grant #PZ00P2_161343. We thank Zamin A.

Kanji, Oliver F. Bischof, Thomas Peter, and Prem Lobo for their valuable discussions and the whole group of Ulrike Lohmann for their support.

*Data availability*

The data presented in this publication is available at the following DOI: https://doi.org/10.3929/ethz-b-000303444  (Friebel

and Mensah, 2018)

*Competing interests.*

The authors declare that they have no conflict of interest.

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
