# Peer review of "Aging aerosol in a well-mixed continuous flow tank reactor: An introduction of the activation time distribution"

_Atmospheric Measurement Techniques, 2018_

## Referee Comment (RC1) · Anonymous Referee #1 · 9 Dec 2018

Referee for Friebel and Mensah 2018
Journal: Atmos. Meas. Tech.

**Summary:**

The authors provide a theoretical framework that introduces the concept of activation time, which is the reaction time necessary to produce an observable chance in an aerosol population's property (CCN ability in this case). This is done in an effort to interpret data from a reactor operated in continuous mode rather than batch mode, the case being made because of the increased use of OFRs and OFR-like reactors compared to environmental or smog chambers. They describe the entire reactor operation from fill-up to shut-down. They provide a mathematical description which, to my understanding, is a piecewise solution in time for activation fraction of aerosols in question (soot aerosols subject to heterogeneous ozonolysis). This manuscript is front-heavy with concepts of chemical reactor engineering (ideal reactors and their residence time distributions), and how these are used to develop analytical expressions for the time profiles. Then their expressions are overlaid to data from two experiments to observe model agreement. Finally, they depict a theoretical example based on preexisting data in the literature to show how their parameter $t_{act}$ can be used to compare data from independent experiments, or even better, how future users chose to operate their reactors.

This is very important work that fits the scope of AMT; particularly the illustrative example in Fig. 7. However, I think the manuscript be improved. An in-depth revision of Sections 1-4 is necessary, mostly for emphasis on technical details and wording to reach a broader audience (that is, one unfamiliar with reactor design or operation). For example, the authors need to clarify what they mean by 'non-gradual' as soon as it is mentioned. Are they referring to fast reactions, e.g., heterogeneous nucleation? Or are they simply referring to non-steady state?

There is a bit of a disconnect between the theory and application. Probably because the nature of the subject is challenging. The authors are encouraged to make it clear in the Introduction that they are looking at CCN. Also, because $t_{act}$ is yet to be explained, words like 'parameters' have no meaning thus far; they do eventually by the end, but I think not to confuse the reader a revision is necessary (I offer suggestions in the Major and Minor comments sections for the authors' considerations). From what I can tell, their data is centered on reactors operated in continuous mode, yet the word 'OFR' is mentioned only in Section 5, when the introduction is focused on the large batch reactors. For that matter, comparison of the mathematical framework to that of a PFR is not present. If so, I think some mention as to why should be made. It seems to me that PFR-like reactors (e.g., flow tubes) work well. Why are CSTRs preferred by the authors? Mathematically, it would appear to me you need an RTD, and the PFR has one (Delta function), so why not compare?

**Major Comments:**

Abstract: Details can be improved; I offer suggestions for the authors to consider in the Minor Comments.

1. Introduction: I strongly encourage the authors to be more precise in their sentences. It appears that what the authors communicate is not what they mean, and to reach a broader audience, I think details should be made clear. While Sections 5 and 6 are very clear and logical, at least in my view, Sections 1-4 are not. I encourage OFRs like the PAM (e.g., TPOT, CPOT, etc.) to be addressed early on. The authors can read more in Lambe et al. and Mitroo et al., already cited by them. Also what is not clear is whether the authors have a new CSTR design (different from that of conventional OFRs or Teflon chambers) or if they just develop a mathematical approach for data coming from a CSTR. Or both. Mention of the SAPHIR, in my view, belongs here.

2. Introduction to CSTR: This section is of course important for readers who are not familiar with environmental reactor engineering, however, is not only available in any chemical engineering textbook, but also summarized by Mitroo et al. (Appendix A). If the authors see fit, I would suggest renaming this section as 'SAPHIR operation' or something similar, and then have Filling, SS, and Flushing sections. It seems that from Sections 5-6, their math can be applied to non-CSTRs like the PAM and TPOT, so I wonder if when the authors say 'CSTR' they mean 'non-batch'. Reactors operated in continuous mode range from CSTRs to PFRs, from a mixing perspective.

3. Introduction of the activation time ($t_{act}$) for non-gradual transitions: A major comment I have here that I alluded to prior to this section is to be explicit when talking about parameters. E.g., P6 L20 "If all other parameters stay constant…" what does this mean? Flow parameters? Temp and RH? If so, what is the parameter that is changing? I don't think the reader thinks of AF by now. Also, P6 L24-26 seem to me like the crux of the study (unless I'm mistaken). Are the authors looking at a specific scenario where they keep RH constant but slowly react aerosol with (e.g, ozone for sake of argument) and there is a very small time window where enough reaction occurred to make the aerosols in the reactor cloud nuclei at that supersaturation? Is that time window what current reactors cannot accurately allow determination of, but this method does? Why can't a PFR be used to detect that? If so, this concept needs to go in the introduction, with specific application to CCN if helpful. Finally, for the authors' consideration, it appears they want to keep the x-axis uniform in their equations by introducing $t_{switch}$ and $t_{offset}$. Seems to me like these are just substitutes for a Heaviside function. Would the authors consider using a Heaviside function instead to make the math simpler?

4. Application in first experiments: I don't think this section header reflects the content. Maybe change to something else? Section 4.2 was described well. My only major comment here is why Figure 3 has a lag (noticed after seeing Table 2 and Figure 5) Why does the 'step' or 'non-gaussian' have a lag? Even in the filling regime a CSTR gives no lag. E.g., in P14 L10-13 I remain unconvinced that the blue line in Fig. 5 should have a lag. I think that assumption (P7 L7; see comments for Fig. 5) is highly questionable. I think that leads to an artifact in the calculation, and that is reflected by the stark difference in $t_{act}$-onset (Table 2). If the authors provide a counter, I'd be happy to know why.

5. Application to experimental data: No major comments here other than those that stem from the previous section.

6. Application of $t_{act}$ to other continuous flow chambers: I think this section would be very useful for OFR users on how to use OFRs for CCN meaurements! Still, neither the PAM nor the TPOT

are CSTRs, so how have the authors applied $t_{act}$ to their RTDs? Also, what if aerosol content is not well known (e.g., field sampling)? How is their mathematical framework applied? I am still unclear as to what parameters are needed experimentally.

7. Conclusion: No major comments here.

**Minor Comments:**

P1 L6: Consider changing "atmospheric" to "realistic".

P1 L7: Arguably a small point, but I'd encourage a revision of "achieve extended observation times" to "obtain measurable reaction rates, due to long residence times" or words to that effect. I think owing to the small reactor design community there's often scant attention paid to the details of a reactor and how it operates by the average reader; and this work has potential for a broad audience, so ensuring the readers become educated about reactor design and meaningful parameters to evaluate its performance is important in my opinion.

P1 L8-9: Change "…in a CSTR mode." to "…as a CSTR." Also, if the authors wish to introduce the concept of a CSTR this early on, perhaps the opening sentence could mention the use of environmental chambers as batch or semi-batch reactors.

P1 L9: Consider changing "…which…" to "…that…".

P1 L9-10: Mean values of what? Perhaps 'its mean value', referring to the data.

P1 L10: Consider a colon, e.g.: "…metric: …".

P1 L13: "Furthermore, we show…"

P1 L14: Are the authors referring to the PAM? Perhaps give an example.

P1 L14-15: Rephrase sentence. $t_{act}$ explains or $t_{act}$ helps explain? What are the different chambers? Are they smog chambers vs. OFRs? Are they aerosol flow tubes vs. OFRs?

P1 L17: This may apply throughout but "Aerosol particles" should be "Aerosols".

P1 L17-18: Small detail, but stating aerosols are 'emitted' into the atmosphere implies they may not be generated by nucleation. The authors can consider the following rephrase: "Atmospheric aerosols undergo various reactions; the timescale for which depends on their lifetime."

P1 L19: Citation for sea salt aerosol lifespan (textbooks are appropriate as well) please. Also, replace comma with '"whereas".

P1 L21: Please check 'aerosol particles', as per my previous comment.

P1 L21-22: The authors can remove "…the fate of aerosol particles in the atmosphere and…"

P1 L22: I'd change 'parameter for' with 'process affecting'.

P1 L23: "task" should be plural; also, I'd change 'the investigation of aerosol' with 'understanding atmospheric aging'.

P1 L23-27: I'd encourage the authors to reword this section and not to gloss over how large reactor are 'technically' unfeasible, but instead be more explicit (e.g., wall losses, time dependencies, etc.). Also, I don't think the SAPHIR has a set 16 h operation time, so perhaps the authors can say '12-24h' to indicate a range.

P1 L28-29: This is a technical detail I would like not to be overlooked. "…in order to reduce the reaction time…" is not the objective; it is a consequence (advantageous, admittedly, for investigating physiochemical properties of SOA or LVOCs). The objective is to artificially augment the reaction rate. I believe the authors know this, but a reader may not, and I encourage the idea of having these details be clear. This is important work and should be presented as such!

P2 L4: Please consider adding a citation of Renbaum and Smith, doi: 10.5194/acp-11-6881-2011

P2 L6: In the engineering literature, CSTRs are well described, but the acronym is use as a general term for any well-mixed vessel. The authors choose to apply (or design?) a CSTR for their work in addition to the mathematical expression for $t_{act}$. Have I understood this correctly? Might I suggest them to give their reactor a more personalized name?

P2 L8: An ideal CSTR is perfectly mixed. A real CSTR is well-mixed.

P2 L9: Perhaps the authors can rephrase "…close to real processes in the atmosphere…" to "…mimics mixing in the free troposphere more accurately than [other reactors]". But more importantly, is the mixing state in this reactor important because it mimics atmospheric dynamics or because it allows more accurate data retrieval from laboratory experiments?

P2 L11-12: That's absolutely the case for a CSTR! At steady state, the distribution of ages is fixed, and is dependent only on reactor volume and flowrate. I think this needs to be clearer.

P2 L14: In addition to refining the sentence to make it sound less informal, I would encourage the authors to cite also Levenspiel's Omnibook (I think they cite it later, but it's missing in the Reference Section).

P2 L19: I'm confused, what do the authors mean when they say 'non-gradual' (see Major Comments)? Do they mean time-dependent? Do they mean non-steady state? This is a key concept in their work, so I would ask them to define it explicitly for the reader.

P3 L3-4: More than a physico-chemical (physio-chemical?) perspective, I'd say from a flow or mixing state perspective. Also, PFR can be placed in acronym in brackets (although PFRs can be mentioned in the introduction), and CSTR has already been spelled out earlier, so just the acronym should suffice here.

P3 L9: Can the authors make the case that environmental / smog chambers are batch-type reactors?

P3 L9: Again, I would urge the authors to be detailed. A PFR (which is the idealized reactor design on which flow tubes are built) allows no axial mixing (as the authors point out), but is perfectly mixed radially! The ADM (mentioned by Lambe et al.) allows for deviation from the PFR and is closer to describing flow tubes, but that discussion can be briefly mentioned, if needed at all.

P3 L17: Residence time of what? The large chambers?

P3 L19: Consider replacing "During a subsequent…" with "Following steady state, upon shut down, is the…"

P3 L20: To better illustrate their point, I think the authors can put an arbitrary schematic in the Supplement rather than alluding to a figure that has not yet been explained.

P3 L21: I don't think 'hydrodynamic' is necessary, but I could be wrong.

Equation (2): A suggestion to simplify notation, perhaps the subscript 'CSTR' can be removed, seen as it is implied. Also, (t) can be placed outside of the square brackets, as can the subscript 'feed-in', which I would also suggest be replaced with a subscript zero.

P4 L1: This is a good point by the authors! I would encourage a citation of Lambe et al., seen as what the authors are describing here is essentially the result of a tracer study (A is a chemically inert tracer essentially).

P4 L11: I would encourage a citation of Mitroo et al.

Equations (3-4): These are E and F-curves as described by Mitroo et al.; it may be worthwhile to mention.

P6 L16-19: This needs to go either at the end of the introduction, or at P2 L19 in my view.

P6 L20-21: This sentence needs to be rewritten as it is too handwavy and comes across as pseudo-science. "…a particle that undergoes changes that result in a non-gradual transition…" made no sense in my mind until I finished reading the manuscript. Could the authors come up with a physical example to help convey what change has been 'undergone' that resulted in a 'transition'? Or is the 'change' itself rapid (e.g., heterogeneous nucleation)? Are the authors implying they can model a process this fast as a function of time, and decouple it from other timescales within the reactor? Is a CSTR the best approach?

P10 L4: "aerosol particles"

P10 L6: "aerosol particles", but more importantly, what properties are distributed around a mean value? If they are physical (e.g., dpg, sigmag, etc.) maybe. If they are chemical (e.g., nitrate content) then not really.

P10 L7-9: I don't follow the logic here. If I understand correctly, the authors are saying that, due to multiplicity of charges on some aerosols, an aerosol population that follows a lognormal distribution if plotted by mobility diameter doesn't follow a lognormal distribution by aerodynamic diameter? I don't see how an aerosol population that is unimodal in mobility diameter can be multimodal in aerodynamic (or geometric) diameter.

P10 L15: Maybe "…has the potential to activate." instead of "…activates.", because after t=180 min, they don't all activate.

P10 L17: Why was 30 min chosen as standard deviation?

P14 L1: Unless I'm mistaken, $t_{act}$s don't really differ; only $t_{act}$-onset for $P_{Gaussian}$ differs.

P14 L12: Fix "tact", but more importantly, please address the Major Comment surrounding this sentence (the lag in Fig. 3 before $t_{act}$).

P14 L13: Fix "P(tact)"

P14 L15: Fix "Pstep(tact)".

P13 L5-6: Please provide appropriate citations.

P14 L21: I would appreciate either a description of the chamber or literature that describes it. I'd really like to know, as I think is important for the reader, if this chamber is indeed well mixed (does it have impellers, fans, baffles?) to where the equations can be applied to the data, or is this chamber not really well mixed? What about residence time in the tubing? The tracer data may require some convincing (see four comments down P15 L6).

P14 L22: For those not familiar with soot generation, what is a miniCAST, set point 6?

P14 L30-35: Would the authors see fit to put these two points at the end of the Introduction Section?

P14 L31: Again, I'd encourage the authors to refrain from using the word 'perfectly mixed' when talking about a real reactor. Might I suggest 'well-mixed'. More to my point: no RTD is available until Fig. 6; can a description of the chamber, or literature on it be presented?

P15 L6: Following the comment above: How the particles depict a CSTR would be more believable if the authors provide some way of showing it. Maybe plot an E-curve for the data and overlay that of an ideal CSTR over it? If I calculated it right, 2.78 $m^3$ / 25 LPM is ~111 min. Why is $t_{act}$ more than twice that? In P7 L7 the authors claim $t_{act}$ is one mean residence time for a CSTR. If their chamber is not as well mixed as believed that's OK, but it should be stated (and at least be better mixed than OFRs!).

**Tables and Figures:**

Figure 1: Please indicate a unit for the x-axis (I think it's seconds). Also, this figure is confusing because it should just be one curve representative of SS, but the authors mention in the caption "…while flushing the CSTR." I understand what the authors mean, but maybe the reader won't so this figure or its citation in the text should be made clearer.

Figure 2: No major comments.

Figure 3: No major comments here, other than the curiosity of how a graph like this would look like for a PFR.

Figure 4: No major comments.

Figure 5: Upon seeing Fig. 5, I struggle to now understand Fig. 3 (or, the blue line in Fig. 5). I was under the impression $t_{act}$ is when reactants are introduced. If that is the case, why does the red line show AF > 0 at $t < t_{act}$? Or am I missing something? A CSTR has no lag by design; only PFRs have lags. Even in the 'filling regime'. I think the root of my misunderstanding can be traced back to P7 L7. Why is AF = 0 when t < tact? Even for a system with no Gaussian spread, purely based on CSTR design, at $t = 0^+$ AF (however small) is non-zero. If the authors can explain their assumption in P7 L7, I think it would clear this up (at least for me).

Table 1: No major comments.

Table 2: No major comments on the table itself (maybe capitalize the subscript 'gaussian'?); but I have comments on how the authors choose to explain the difference in values of $t_{act}$-onset for Step and Gaussian (see comment section).

Table 3: No major comments.

Figure 6: No major comments, but I do have a question: it's unclear how the authors' fit matches data well. Was it a fit? E.g., if instead of soot they used salt, what is needed experimentally to determine the blue dotted line in this Figure? Did I miss something in the text?

Figure 7: No major comments, but to be clear, is this illustrative? That $AF_{TPOT} > AF_{PAM}$ at high [OH], and the reverse for low [OH], is subject to experimental data, right?

**References:**

---

## Referee Comment (RC2) · Anonymous Referee #2 · 21 Dec 2018

**General comments:**

This manuscript presented an improved experimental approach to perform atmospheric oxidation of soot particles using a Continuous Flow Stirred Tank Reactor (CSTR), which enables extended sampling time within a small-size conventional aerosol chamber. A new metric of activation time ($t_{act}$) was developed to characterize the change of activated fraction (AF) in different regimes (i.e., filling, steady state, and flushing) for soot particles following heterogeneous ozone oxidation. Good agreements between theoretical calculations and parameterized CCN activities using $t_{act}$ were achieved for their experimental data. The $t_{act}$ concept was also applied into some previous studies with continuous flow chambers. Discrepancies in the CCN activity of BES particles can be better explained with considering $t_{act}$ and residence time distribution, in comparison to those initially interpreted by the bulk H/C and O/C ratios, which couldn't fully characterize the detailed change in particle chemical compositions. This work is worth further application in atmospheric sciences, yet some details and interpretations could be clarified, reorganized, and improved accordingly. I would recommend for the final publication in AMT upon major revisions, as detailed below.

**Major comments:**

1. In the motivation section (Page 2, Line 19): The "non-gradual transition" case of CCN activation suddenly appeared, with no prior introduction or definition of this new concept (instead, which was included in Sect.3). This content seemed to be disconnected with the information detailed in the last sentence, and I didn't catch the importance/necessity of developing a mathematical analysis for the non-gradual transitions in the following statements.

The authors have introduced the concept of CSTR and suggested that *"The steady state in the CSTR is characterized by constant concentration of all compounds and constant reaction rates."*. It is a bit confusing that how the assumed "perfectly internal mixing" is achieved, even if without considering the influences of particle wall loss and coagulation during different experimental regimes. How should readers understand the "constant concentration of all compounds" during aging reactions in the CSTR, where the corresponding compositions/concentration of reactants/products are supposed to vary with such processes?

Another question is about the configuration of the CSTR in this study: did the authors use a real CSTR device for their experiments or not? what kinds of equipment (and how) were actually coupled with the CSTR, in addition to a CCN counter which enables the CCN activation measurements (i.e., the AF results) of aged soot particles? Corresponding details are suggested to be provided especially for those who are unfamiliar with such systems. From my perspective, the organization of this section could be improved for better delivery of the key points.

2. The Section 6, especially the last paragraph of which, is quite confusing. It is good to see the application of the activation time concept ($t_{act}$) into data interpretation of previous chamber studies, with improved agreements among different datasets. Nevertheless, there are several concerns need to be addressed. First of all, the previously used chambers such as PAM, they are actually not CSTR or far from the ideal mixing condition during oxidation. As a result, how can you simply apply the $t_{act}$ or RTD concept for CSTR system into the data interpretation of OFR/PAM reactors? Necessary information is needed to clarify this point.

Another issue is that discrepancies in CCN activity of SOA formed from chamber oxidation experiments could be influenced by various factors, such as gas-particle partitioning and particle-phase reactions during SOA production as well as liquid-liquid phase separation during activation processes. Additionally, the variability in different operation parameters such as relative humidity, initial concentration of

VOC precursors, and acidity in the OFR/PAM chamber can affect the SOA formation process even for a same average OH concentration condition, further influencing the subsequent CCN activation process. In this sense, how to evaluate or exclude the impacts of these factors on the agreement of CCN activity (or AF) measurements for different types of OFR or PAM experiments? Namely, how can we confirm that the discrepancies are predominantly introduced by the activation time (or RTD) rather than by the other influencing parameters, although the application of $t_{act}$ can better capture the deviation of CCN activity (likely due to change in chemical compositions) than what the bulk H/C and O/C ratios do? Further discussion is needed to clarify the abovementioned points.

**Specific comments:**

1. **Abstract**: What does the "non-gradual transitions" refer to here (Line 12)? In the last sentence, what specific kinds of "discrepancies" are you suggesting? It is better to clarify these concepts precisely, as which are important points to show the significance and applicability of this study.

2. Page 2, line 8: How is the *"perfectly mixed"* defined here? It is unclear especially to readers those are unfamiliar with the CSTR technique. Following which, what do you mean that *"real processes in the atmosphere where aerosols are constantly emitted, mixed and removed"*? Are you sure of the "constantly" condition in the ambient environment? Which specific atmospheric processes have you included in this statement, any references can be provided to support the idea?

3. **Equation 5**: Why is the exponential part not expressed as "$e^{-\frac{t-t_{switch}}{\tau_{CSTR}}}$" for the flushing regime? Please check the conversion carefully.

4. Page 6, line 20: As a crucial parameter introduced in this study, the activation time ($t_{act}$) for non-gradual transitions was developed. However, what do you mean "If all the other parameters stay constant" during non-gradual transitions, which specific parameters are you referring to? Is it easy to achieve in practical conditions of laboratory chamber experiments?

5. **Equation 7**: I think it should be "$e^{\frac{-t_{switch}}{\tau_{CSTR}}}$" in the exponent. Beside, it's better to add a pair of parentheses for "$-\tau_{CSRT}$", since it appeared after "·".

6. **Equation 8**: Why is the simplified equation not expressed as '$t_{act} = -\ln(AF(t))·\tau_{CSTR}$'? I'm wondering how will the value of $AF(t\rightarrow\infty)$ be, could it be 0 as suggested by the exponentially decreased curve in Fig.2, or probably approaching 1 like what AF responds when switching to the flushing regime as shown in Fig.3? How should the readers understand the corresponding physical meaning of $AF(t\rightarrow\infty)$ in this steady state condition? Corresponding details are necessary.

7. Page 8, line 7: It sounds a bit strange of "global" AF? Is the "global" trying to represent the specific exponentially increased AF inside CSTR or just to show a different AF case with other non-CSTR chamber experiments?

   Line 10: *"… and therefore the global AF only if $t_{act} = t_{switch}$."* Some information was missed in this sentence.

8. **Title of Sect.4**: What does the "first experiments" mean? Try to update the message into a more informative one.

9. Page 12, line 10: What does the "uniform" mean: "activate uniformly" here and "a uniform aerosol population" in the caption of Fig.5? Are you trying to say the initial particles with the same particle size and chemical composition? If so, how to understand the Gaussian distribution scenario (i.e., *"This is because there are some particles in the population, that activate earlier than the mean activation time."*), as all the uniform particles are supposed to activate at a same activation time? More

straightforward/concise descriptions would be useful to explain the scenario clearly.

10. Page 15, line 10: How was the particle wall loss rate of $k = 0.000625 \ \text{min}^{-1}$ estimated? Where can the readers find the corresponding clues/data for calculation?

11. Page 15, line 15: What is the meaning of the last sentence? What does the "other SS" refer to? Where can readers find the corresponding details? Necessary information is needed.

12. **Figure** 2: Is the "particle age" of x-axis with the same meaning of the "residence time" in Fig.1? If not, please specify accordingly in the corresponding places.

13. **Figure** 6: Why is the unit of particle concentration in Fig.6(A1) and (B1) different from those in Figure 3?

    In Fig.6 (B1), why are the data after 800 min missing? As assumed early in this study that all compounds in CSTR have perfectly mixed thus with constant concentrations during steady state, how to explain the increasing trend in observed particle concentration in the duration of 400-600 min, i.e., AF almost reached a stable level around 0.2 at 1.4% SS conditions)? More detailed discussion should be provided in the corresponding data interpretation sections.

14. Page 19, line 6: The last sentence is a bit confusing. It is better to clarify the "metric" here, e.g. metric of what specific aspects.

**Technical corrections:**

1. **Abstract**, Page 1, line 10: *"… the newly introduced metric: activation time"*

2. Page 3, line 27: *"… can be calculated as a function of …"*. A similar issue exists in Line 16, Page 6.

3. Page 6, line 13: *"... to describe  continuous changes"*?

4. Page 6, line 19: *"... can be considered as a non-gradual change."*

5. Page 6, line 20: *"If  all the other parameters stay constant, while a particle undergoes changes that result in a non-gradual transition ..."*

6. **Equation 9**: Why do you use different multiplication signs in these equations, e.g., "*" and "·"? It makes more sense to keep consistent within the same manuscript.

7. **Table 1**: Why is the layout of this table so different from other two tables in this manuscript? The corresponding details could be better organized.

8. Title of **Sect.4.3**: *"Calculation of the total activated fraction"*

9. Page 12, line 12: *"While the uniform scenario shows no activity be for reaching $t_{act}$ ..."* Do you mean 'before'?

10. Page 14, line 8-9: *"As there is a significant share of particles activating significantly earlier than the nominal activation time ($\mu = 180$ ) in the case of a Gaussian distribution a fraction of 1 % of the entire particle population within the CSTR is already activated after 87 min."* A comma is needed to clarify the point.

11. Page 14, line 12: *"The difference in tact of 10 min between the two P(tact)-approaches is due to the application ..."*

    It is very common to see that $t_{act}$ was written as $tact$. Similar issues also exist in some other expressions, e.g., $Pstep$, which should be $P_{step}$. Please check through the manuscript carefully and make necessary updates accordingly.

    In the same paragraph, there are many long sentences without proper splits or connections, which might make the readers difficult or even confused to catch the meaning effectively. For instance:

12. Page 14, line 15-16: *"As can be seen in Graph C of* Fig. 4, 50 % *of the particles with a residence time equal to the nominal activation time are activated in the case of a Gaussian distribution corresponding to $t_{act}$0.5."*

13. Page 14, line 23: *"... were diluted with* particle-free *and VOC-filtered air…"*

14. Page 14, line 25: *"The aerosol flow was fed into the aerosol chamber, where a constant*  ozone *concentration of 200 ppb was …"*

15. Page 14, line 27: *"The size distribution data was acquired by a … (SMPS) system from which the*  *total particle concentration could be derived."*

16. Page 15, line 5: The *"(blue solid line)"* is not needed, since there is only one curve in the corresponding subplots.

17. Page 15, line 18: *"… μ as well as σ*  is *larger for P($t_{act}$) at a 1.0 % SS*  *compared to the results obtained for 1.4 % SS. The mean activation time being larger for 1.0 %* SS *indicates that the longer the chemical aging* proceeds, *the initially inactive soot particles activate*  at a *lower SS."*

18. Page 15, Line 23: The comma between "*P($t_{act}$)*" and "requires" is unnecessary.

19. Page 17, line 1 and 3: *"Within these* types *of chambers …"*

20. Page 17, line 10: *"secondary* organic *aerosol (SOA)"*, and the "VOCs" should be defined before when it appeared for the first time.

21. Page 17, line 24: *"...to be directly proportional* to *the AFs…"*

22. Page 18, line 3: *"...we*  present *two scenarios."*

23. Page 18, line 9: *"...other* parameters *can agree very well."*

24. Page 19, line 22: *"...soot particles transitioning*  from *initial CCN-inactivity to CCN-activity over the course of …"*

---

## Author Comment (AC1) · 22 Feb 2019

**We thank the reviewer for the comprehensive feedback on our work. With the help of the reviewers' comments we greatly improved the understandability of our work and made it more accessible to a broader audience. Detailed answers to the individual comments are given below. For clarity, the reviewers' comments are written in black, and our response in** red**. Texts from the old version of the manuscript are typed in** green **and texts from the revised manuscript in** blue**.**
* * *
Referee for Friebel and Mensah 2018
Journal: Atmos. Meas. Tech.

**Summary:**

The authors provide a theoretical framework that introduces the concept of activation time, which is the reaction time necessary to produce an observable chance in an aerosol population's property (CCN ability in this case). This is done in an effort to interpret data from a reactor operated in continuous mode rather than batch mode, the case being made because of the increased use of OFRs and OFR-like reactors compared to environmental or smog chambers. They describe the entire reactor operation from fill-up to shut-down. They provide a mathematical description which, to my understanding, is a piecewise solution in time for activation fraction of aerosols in question (soot aerosols subject to heterogeneous ozonolysis). This manuscript is front-heavy with concepts of chemical reactor engineering (ideal reactors and their residence time distributions), and how these are used to develop analytical expressions for the time profiles. Then their expressions are overlaid to data from two experiments to observe model agreement. Finally, they depict a theoretical example based on preexisting data in the literature to show how their parameter tact can be used to compare data from independent experiments, or even better, how future users chose to operate their reactors. This is very important work that fits the scope of AMT; particularly the illustrative example in Fig.7.

However, I think the manuscript be improved. An in-depth revision of Sections 1-4 is necessary, mostly for emphasis on technical details and wording to reach a broader audience (that is, one unfamiliar with reactor design or operation). For example, the authors need to clarify what they mean by 'non-gradual' as soon as it is mentioned. Are they referring to fast reactions, e.g., heterogeneous nucleation? Or are they simply referring to non-steady state?

We understand that many readers are not familiar with the wording we used. We thank the reviewer for pointing this out. Our approach to improve the understandability is to add synonyms commonly used in the atmospheric science community. Additionally we added examples to illustrate this concepts. For this we chose processes commonly investigated in the atmospheric science community. Nevertheless, these concepts are not limited to atmospheric science and can be applied in different fields as well.

In case of the term "non-gradual" we neither refer to fast reactions nor non-steady state. "Non-gradual" describes changes like phase-transitions where a property changes step-wise. This is the opposite of a gradual or continuous change of a property. An example would be the freezing of water. Below or above 0°C the density of liquid water/ice changes gradually with the temperature. At 0°C the density does not change gradually but changes step-wise.

To clarify what is meant by "non-gradual" we extended the introduction of this phrase and added "step-wise change" and "transition between binary states" as alternative explanations. "Transition between binary states" hereby means that a system/particle can be described by two distinct states. Either a droplet is liquid or frozen. A transition from one state to another one can be described as "non-gradual" as well.

We added a list of possible transition that can be described as "non-gradual", "step-wise change" and "transition between binary states"

(P2 L26-39)

> Such transitions in binary systems are step-wise, also referred to as non-gradual changes in a particle property, such as:
>
> 1) Freezing of a water droplet: Step-wise and therefore non-gradual change in the particle density; the water is either in liquid or solid state.
>
> 2) Deliquescence of soluble aerosol particles: The particles show a step-wise i.e. non-gradual increase in diameter.
>
> Binary particle properties are not necessarily intrinsic particle properties, but can also be defined by the measurement protocol.
>
> 3) CCN-activity: The chemical and physical properties of an aerosol particle can vary, but the particle is either CCN-inactive or CCN-active at a defined super saturation (SS).
>
> 4) Growth beyond a threshold: Condensational growth of an aerosol particle leads to a continuous and gradual increase of the particle diameter. A binary system can be defined by introducing a threshold diameter that can be arbitrarily chosen. The aerosol particle is either smaller or larger than this defined threshold diameter. The same holds true when particles are separated e.g. in aerosol impactors.
>
> Therefore, the concept of non-gradual transitions/transitions within binary systems can be used to describe a multitude of changes in particle properties.

There is a bit of a disconnect between the theory and application. Probably because the nature of the subject is challenging. The authors are encouraged to make it clear in the Introduction that they are looking at CCN. Also, because tact is yet to be explained, words like 'parameters' have no meaning thus far; they do eventually by the end, but I think not to confuse the reader a revision is necessary (I offer suggestions in the Major and Minor comments sections for the authors' considerations).

This manuscript introduces two new concepts that were developed side by side and support each other. However these concepts are not limited to one particular application. The first concept is the use of aerosol chambers in CSTR-mode. The second concept is the idea of analyzing data with the activation time concept, which relies on "non-gradual" transitions. We try to make a clear distinction between the experimental approach rather new in atmospheric sciences and the $t_{act}$-concept. We hope that the revision of the sections where non-gradual" transitions (section 1 "Motivation") and the $t_{act}$-concept (section 3

"Introduction of the activation time ($t_{act}$) for transitions in binary systems") are introduced has significantly increased the understandability. Nevertheless, we admit to be challenged in the attempt to introduce these new concepts in a way that their general applicability is not undermined.

From what I can tell, their data is centered on reactors operated in continuous mode, yet the word 'OFR' is mentioned only in Section 5, when the introduction is focused on the large batch reactors.

The experimental part of the manuscript paper focusses on the application of the CSTR approach in atmospheric sciences. A CSTR is a continuous mode reactor just as an OFR but in many aspects the opposite of OFRs. To support the reader in recognizing the differences between the individual concepts of CSTRs, PFRs/ORFs and batch reactors, we expanded the introduction of alternative reactor concepts in section 2 "Introduction of the CSTR". We further added references to the chamber operated in the respective modes.

(P3 L15-L39)

From a technical perspective, generally three types of reactors are distinguished: the batch-reactor, the plug flow (PFR) or flow tube reactor, and the Continuous flow Stirred Tank Reactor (CSTR).

In an aerosol chamber operated in batch mode, the reaction volume is first filled with the sample aerosol as fast as possible to achieve high homogeneity of the sample. After the desired start concentration is reached further addition of the sample aerosol is stopped and the aging is initiated e.g. by addition of the oxidant. This point in time is generally defined as the start of the experiment and referred to as $t = 0$. Data acquisition of the ageing sample takes place while the reaction volume is flushed with sample-free gas. The composition throughout the chamber is homogeneous but evolving in time, therefore no steady state conditions are ever achieved. This concept is used to operate many large scale environmental chambers (Cocker et al., 2001; Leskinen et al., 2015; Nordin et al., 2013; Paulsen et al., 2005; Platt et al., 2013; Presto et al., 2005; Rohrer et al., 2005).

A PFR is a steady state reactor in which no mixing along the flow path (axial mixing) but perfect mixing perpendicular to the flow (radial mixing) takes place. Further, a continuous feed-in of reactants and withdrawal of sample take place at equal flow rates simultaneously. This results in a constant composition of the output solely depending on the residence time within the reactor. This ideal system is approximated by many Oxidation Flow Reactors (OFR) e.g. PAM chamber (George et al., 2007), TPOT Chamber (Kang et al., 2007), Micro Smog Chamber (MSC; Keller and Burtscher, 2012), or the TUT Secondary Aerosol Reactor (TSAR ; Simonen et al., 2017). The main difference between an ideal PFR and real OFRs is that in OFRs significant but unintentional mixing of the aerosol along the flow path takes place (Mitroo et al., 2018). Therefore, OFRs show a significant residence distribution.

The CSTR is a steady state reactor with a constant reactant feed in and sample withdrawal as well but opposite to OFRs, the volume is actively stirred to achieve a

homogeneous composition throughout the reactor volume. Due to the active mixing, sample stream composition and conditions are the same as within the entire chamber volume. The concept of the CSTR requires perfect internal mixing, which cannot be achieved in real systems. However, due to the good miscibility and low viscosity of gases and the aerosol particles being homogenously dispersed, it is possible to achieve a degree of mixing which is very close to a perfectly mixed system. Especially in the case of mimicking atmospheric processes, residence times of several hours are achieved. Compared to that, the time needed for dissipating all gradients, which is in the order of seconds to minutes, can be considered small.

For that matter, comparison of the mathematical framework to that of a PFR is not present. If so, I think some mention as to why should be made. It seems to me that PFR-like reactors (e.g., flow tubes) work well. Why are CSTRs preferred by the authors? Mathematically, it would appear to me you need an RTD, and the PFR has one (Delta function), so why not compare?

The mission of this publication is to present a reaction chamber operation mode that is not that prominent within the atmospheric community yet but comprises important benefits for the investigation of atmospheric processes. These benefits include extended reaction times and low reactant concentrations. The limited popularity of the CSTR concept within the atmospheric community might be partly due to limited availability of analysis procedures of data resulting from such experiments. We developed the activation time concept to allow for the analysis of reactions and processes relevant to the atmospheric community. Therefore, this publication intentionally focusses on the activation time concept and the CSTR-mode operation of aerosol chambers. We refer to OFRs and large batch aerosol chambers to discuss the differences, benefits, and disadvantages of the CSTR concept in comparison to these reaction chamber concepts established in the atmospheric community. The mathematical framework for the analysis of data from ORFs is not comparable to the $t_{act}$ concept for CSTR data presented here. Instead the parameters of interest of OFR data are average residence time, average exposure to oxidants and critical super saturation/super saturation onset. We discuss the different metrics and their comparability in section 3.5 "Equivalent parameters $t_{act}$ – onset and $t_{act}$ vs $t_{act}0.5$" and section 5 "Application of $t_{act}$ to other continuous flow aerosol chambers" now in greater detail.

**Major Comments:**
Abstract: Details can be improved; I offer suggestions for the authors to consider in the Minor Comments.
1. Introduction: I strongly encourage the authors to be more precise in their sentences. It appears that what the authors communicate is not what they mean, and to reach a broader audience, I think details should be made clear. While Sections 5 and 6 are very clear and logical, at least in my view, Sections 1-4 are not. I encourage OFRs like the PAM (e.g., TPOT, CPOT, etc.) to be addressed early on. The authors can read more in Lambe et al. and Mitroo et al., already cited by them.

We expanded the introduction section and made a clearer distinction between the application of the CSTR approach for atmospheric experiments, the development of a CSTR-specific mathematical framework, the newly developed $t_{act}$-concept, as well as the application of the $t_{act}$-concept to other continuous flow steady state chamber, namely OFRs. (P2 L40- P3 L7)

> In the following, we discuss a theoretical basis for the analysis of time-dependent changes in binary systems within well-mixed continuous flow aerosol aging chambers (CSTR-approach). We developed a mathematical framework which allows the retrieval of characteristic parameters from the system of interest (e.g. CCN activity) and which allows for the calculation of the parameter of interest throughout the entire duration. Key element in this framework is the activation time ($t_{act}$) which marks the time after which the individual aerosol particle undergoes a transition within a binary system. We start by introducing an idealized system in which $t_{act}$ can be described by a single number and proceed to a more realistic setting in which we incorporate a distribution of particles with different individual $t_{act}$'s (activation time distribution, P($t_{act}$)). Further, we test the $t_{act}$-concept on real experimental data and finally apply it to other types of continuous flow aging chambers such as OFRs. We show that application of the $t_{act}$-concept is capable of giving new insights to ORF data and further significantly improves the understanding of discrepancies in experimental results obtained in intercomparison studies Lambe et al., (2011) with different reactors such as the Potential Aerosol Mass Chamber (PAM) chamber and the Toronto Photo-Oxidation Tube (TPOT).

Also what is not clear is whether the authors have a new CSTR design (different from that of conventional OFRs or Teflon chambers) or if they just develop a mathematical approach for data coming from a CSTR. Or both.

We do not present a new CSTR design but the application of the concept, which is well established in chemical engineering but not that prominent in atmospheric sciences, yet. The authors would like to highlight the fact, that CSTRs are neither OFRs nor Teflon chambers operated in batch mode. Teflon chambers could be operated in CSTR mode but this demands the installation of a fan as the concept of the CSTR requires perfect internal mixing of the sample.

Mention of the SAPHIR, in my view, belongs here.

The SAPHIR-chamber is mentioned as an example of a large reaction chamber:

(P1 L28-29)

> Here aging durations of up to 16 hours and beyond at atmospherically relevant reactant concentrations can be achieved, which has been shown e.g. for the SAPHIR chamber of FZ Julich with a volume of 270 m3 (Rohrer et al., 2005; Rollins et al., 2009).

2. Introduction to CSTR: This section is of course important for readers who are not familiar with environmental reactor engineering, however, is not only available in any chemical engineering textbook, but also summarized by Mitroo et al. (Appendix A).

We agree to the fact that Mitroo et al. discusses concepts relevant to for describing residence time distributions in OFRs within their paper. The equations presented in section "3.2 Tank-in-series model for indirect deconvolution" within their publication allow to calculate RTD and $RTD_{sum}$ – curves. This is a general description of multiples CSTRs that are connected in series. For the special case where N=1 the mentioned equations become the equations we used in our study.

Instead of using RTD and $RTD_{sum}$ Mitroo et al. labeled the curves as E and F curves, which is common in the engineering community. We prefer the first option since it is a more intuitive notation.

In "Appendix A: The use of E and F curves" of their publication Mitroo et al. describe how their RTDs were determined and how to calculate moments of the RTD.

However, none of this contains a description of the CSTR and its RTD, therefore we do not refer to Mitroo et al. in this section.

In section "5 Application of $t_{act}$ to other continuous flow aerosol chambers" of our manuscript we discuss how different RTDs in different OFR-designs lead to differing results of a parameter of interest. A quantitative application of $t_{act}$, requires precise knowledge of the chambers RTD. The concepts of Mitroo et al. can be used to obtain that. However, within this work we only focus on a qualitative application of $t_{act}$ to continuous flow aerosol chambers that are characterized by none perfect mixing

If the authors see fit, I would suggest renaming this section as 'SAPHIR operation' or something similar, and then have Filling, SS, and Flushing sections.

We recognize the suggestion of renaming this section but remain at the initial naming. This is due to the fact, that this section is intended to introduce the CSTR concept in general as well as its specific aspects and characteristics. The SAPHIR chamber is a batch reactor and not a CSTR. While within the operation of a CSTR the three different regimes of filling, steady state, and flushing can be achieved, no steady state can be achieved throughout the operation of a batch reactor. This is one of the fundamental differences between these two reactor types.

It seems that from Sections 5-6, their math can be applied to non-CSTRs like the PAM and TPOT, so I wonder if when the authors say 'CSTR' they mean 'non-batch'. Reactors operated in continuous mode range from CSTRs to PFRs, from a mixing perspective.

We agree to the reviewer that both CSTR and PFR are continuous mode reactors and include a phase of steady state operation. Nevertheless, we want to highlight the critical difference between these two reactor types. While perfect/well-mixing is an additional prerequisite exclusive to the concept of the CSTR, OFR are characterized by a partial mixing of the sample. As this section is titled 2 "Introduction of the CSTR" we focus on the mathematical framework relevant for CSTRs exclusively. For a comprehensive introduction of the new

concept and to allow the reader to first get acquainted to it, we postpone the introduction of the transferability of this new mathematical framework to other reactor types to a later section of this manuscript (section "Application of $t_{act}$ to other continuous flow aerosol chambers"). Further, we attempt to clarify the distinction between well-mixed (CSTR) and partially-mixed (OFR) systems and mention this now in the abstract:

(P1 L15-18)

> Furthermore, we show how $t_{act}$ can be applied for the analysis of data originating from other reactor types such as Oxidation Flow Reactors (OFR), which are widely used in atmospheric sciences. The new $t_{act}$ concept significantly supports the understanding of data acquired in OFRs especially these of deviating experimental results in intercomparison campaigns.

The new mathematical framework presented in this publication is developed for the extraction of quantitative data from CSTR measurements. Nevertheless, equation 13 (P13 L11) represents a general expression that can be applied to any continuous flow but non-CSTR chambers as well. For the retrieval of quantitative date, additional information about the specific chamber is required. We show that despite the lack of access to such information the application of the $t_{act}$-concept allows not only for a qualitative comparison of the TPOT and PAM chambers but allows even for an explanation of the differences in the measurement results.

$$AF(t)= \int_{t_{act} = 0}^{t_{act} = t} AF(t_{act},t) \cdot P(t_{act}) \, \mathrm{d}\, t_{act} \tag{1}$$

The following section was added to discuss a quantitative application of $t_{act}$ for data from OFR-experiments.

(P20 L23-36)

> Up to now, the discussion did not include many important processes that are relevant in aging chambers e.g. particle wall-interaction, gas-phase-partitioning, fluctuating input concentrations while field measurements, or inhomogeneities inside the OFR. These aspects are important for many processes such as the formation of SOA and can be incorporated to the $t_{act}$-concept by modifying eq. (13). As the actual calculation requires a multidimensional data array and detailed knowledge about the chamber of interest, this subject matter is beyond the scope of this publication and will not be discussed further. Nevertheless, the overall conclusion is that application of the original/non-adjusted $t_{act}$-concept can explain why measurements within different OFR chambers agree in parameters, which dependent on the bulk properties of the aerosol particle population (e.g. average O:C ratio) and at the same time disagree in parameters, which are dependent on the condition/status of the individual particle (e.g. CCN-activity). Therefore, we suggest to apply the concept of the activation time $t_{act}$ or the activation time distribution $P(t_{act})$ as metric in addition to calculating average values, such as the global $AF$ and OH-exposure if following

conditions are met. One, the system or parameter of interest can be described as a binary system and undergoes step-wise / non-gradual transitions such as CCN-activity. Two, the OFR used has a RTD broad enough to influence the outcome. Three, the conditions inside the reactor are either homogeneous or a correction for inhomogeneities (e.g. different oxidants concentrations inside the reactor) is implemented.

3. Introduction of the activation time (tact) for non-gradual transitions: A major comment I have here that I alluded to prior to this section is to be explicit when talking about parameters. E.g., P6 L20 "If all other parameters stay constant…" what does this mean? Flow parameters? Temp and RH? If so, what is the parameter that is changing? I don't think the reader thinks of AF by now.

"If all other parameters stay constant…" refers to the previously mentioned external parameters. These external parameters can trigger changes in single particles. For example a change in ambient temperature can lead to freezing or melting of a particle. In this section we discuss that such changes can also be the result of e.g. a chemical reaction. If the temperature stays constant but the particle becomes liquid due to a chemical modification we can use the time as parameter to describe this phase transitions.

Examples for the external parameters were added:

Changed from:

> If the all other parameters stay constant, while a particles undergoes changes that result in a non-gradual transitions, this transition can be described as a function of time.

To (P6 L29-32):

We may assume a system in which all external parameters stay constant but the particle itself undergoes a continuous transformation, e.g. due to oxidation. After a certain period of time, this continuous transformation, in this specific case oxidation, can lead to a change in a binary property, e.g. CCN-activity. Ultimately, the step-wise or non-gradual transition is a function of time.

Also, P6 L24-26 seem to me like the crux of the study (unless I'm mistaken).

This is not the crux of the study. The $t_{act}$-concept and the idea of describing changes in parameter on the particle level as transition within a binary system/a stepwise-change/a non-gradual transitions is the crux itself. The mentioned lines are an application of this general idea to the specific process of CCN-activation of soot particles due to aging. This application affects the experimental design but not the overall theoretical framework.

To (P6 L32-35):

We define the required time span (e.g. necessary aging time) that leads to a change in a specific particle property, resulting in a transition in a binary system in another particle property as the activation time ($t_{act}$). This concept is generally valid and can be applied to any kind of transition in a system defined as binary either by intrinsic or operational parameters.

Are the authors looking at a specific scenario where they keep RH constant but slowly react aerosol with (e.g, ozone for sake of argument) and there is a very small time window where enough reaction occurred to make the aerosols in the reactor cloud nuclei at that supersaturation? Is that time window what current reactors cannot accurately allow determination of, but this method does? Why can't a PFR be used to detect that? If so, this concept needs to go in the introduction, with specific application to CCN if helpful. Finally, for the authors' consideration, it appears they want to keep the x-axis uniform in their equations by introducing tswitch and toffset. Seems to me like these are just substitutes for a Heaviside function. Would the authors consider using a Heaviside function instead to make the math simpler?

The section 3 "Introduction of the activation time ($t_{act}$) for transitions in binary systems" is a general introduction of $t_{act}$ and does not include any specific chamber design. This concept is also not limited to any specific scenario. We focus our introduction of the rather theoretical and abstract concept with the help of 2 processes (deliquescence and CCN-activation) that are well known in the atmospheric science community.

(P9 L19-28)

Binary systems can be considered as systems that show a step-wise change in a particle property as a function of an external parameter. Since this is opposite to a continuous/gradual change in a particle property, it can be also described as a non-gradual transition. As mentioned in the introduction, soluble aerosol particles such as ammonium nitrate exhibit a significant change in diameter with increasing relative humidity (RH) due to deliquescence. Similarly, the change from cloud condensation nuclei (CCN) to activated droplets due to exposure to a super-critical super saturation (SS) results in a fast increase of the particle diameter from the nanometer to the micrometer scale that is hard to be continuously tracked by standard measurement instrumentation. A defined diameter threshold is hereby used to distinguish between an aerosol particle and a solute droplet in the case of deliquescence. This is the same between non-activated CCN and cloud droplets. In both examples the relative humidity (RH) in the surrounding gas phase can be considered the external parameter that controls if an aerosol particle is in either of the two states of the binary system (effloresced vs. deliquesced/CCN vs. cloud droplet).

Up to this point, the $t_{act}$-concept and the equations are independent of the overall time-scale. It doesn't matter if the processes need seconds, minutes, or hours to proceed. Therefore we do not discuss applicability of PFRs or CSTRs in this section.

Nevertheless we developed the $t_{act}$-concept to retrieve data from aerosol aging experiments in a 3 m$^3$ CSTR. The aim of this approach was to achieve aging times of several hours. As can

be seen in Fig.6 (P17 L1) this long time spans are indeed needed to investigate the CCN-activation of soot particle due to ozone oxidation. So far, to the best of our knowledge, no OFR-design exists that can reach residence time of several hours which is applied in atmospheric science experiments on a general basis. Increasing the ozone concentration to mimic several hours of atmospheric aging in 2 min (which a typical residence time in OFRs) is a very different approach.

As assumed by the reviewer we want to keep the x-axis uniform. The x-axis is hereby the experimental duration $t$ starting with the beginning of the experiment. Avoiding the parameters $t_{switch}$ and $t_{offset}$ would lead to equally complex equations while potentially increasing the confusion of the reader, since multiple time-axises/x-axises would be needed. Therefore we remain with the math presented so far.

The parameter $t_{switch}$ was introduced since this time is defined be the experiment itself, namely the time after which the chamber is switched to flushing mode.

The explanation of parameter $t_{switch}$ was extended from:

> However, eq. (9) only describes the fraction of particles that are older than $t_{switch}$ and therefore the global $AF$ only if $t_{act} = t_{switch}$. To determine the $AF$ for conditions when $t_{act} < t_{switch}$ ($AF(t=t_{switch}) > 0$) or for a delayed activation, $t_{act} > t_{switch}$, a new parameter
> $t_{offset}$, is introduced. This parameter is an offset of the $AF$-curve along the time-axis. Taking $t_{offset}$ into account, eq. (10) can be
> obtained after integrating eq. (9)

To (P10 L5-9):

As mentioned before, eq. (9) only describes the fraction of particles that are older than $t_{switch}$. Since we defined $AF$ as the fraction of particles with an age above the threshold time $t_{act}$, eq. (9) describes $AF$ only if $t_{act} = t_{switch}$ holds true. To determine $AF$ for conditions when $t_{act} < t_{switch}$ ($AF(t=t_{switch}) > 0$) or for a delayed activation, $t_{act} > t_{switch}$ ($AF(t=t_{switch}) = 0$), an additional parameter has to be introduced. This parameter is an offset of the $AF$-curve along the time-axis and is therefore called $t_{offset}$. Taking $t_{offset}$ into account, eq. (10) can be obtained after integrating eq. (9).

4. Application in first experiments: I don't think this section header reflects the content. Maybe change to something else?

The caption "Application in first experiments" was indeed misleading and has been rephrased to:

> 4 "Application of the new tact to experimental data from CSTR-aging experiments"

Section 4.2 was described well. My only major comment here is why Figure 3 has a lag (noticed after seeing Table 2 and Figure 5) Why does the 'step' or 'non-gaussian' have a lag? Even in the filling regime a CSTR gives no lag. E.g., in P14 L10-13 I remain unconvinced that

the blue line in Fig. 5 should have a lag. I think that assumption (P7 L7; see comments for Fig. 5) is highly questionable. I think that leads to an artifact in the calculation, and that is reflected by the stark difference in tact-onset (Table 2). If the authors provide a counter, I'd be happy to know why.

First we want to highlight the distinction between the activation time $t_{act}$ and the experimental duration $t$. $t_{act}$ is the time needed to modify a particle to such a degree that a particle's property changes step-wise. The experimental duration $t$ is the time that passed since the start of the experiment. We extended the respective section 3.1.2 "Particle activation during steady state" substantially in the attempt to clarify this distinction

Further, In Figure 3 only $AF$ has a lag but not the particle number concentration. As mentioned by the reviewer there is no lag during filling of a CSTR. If we recall, that a chemical transformation of a certain degree is necessary before an individual particle can be CCN active, no CCN activity can be detected prior to this minimum time threshold which we refer to as $t_{act}$. The lag in AF up to $t_{act}$ = 180 min is therefore an intrinsic behavior as the experimental duration $t$ is shorter than this this minimum time $t_{act}$. This underlying concept defines the shape of the curves in Fig. 5 as well. We attempt to clarify this issue in section 3.1.1 "Particle activation during the filling regime":

(P7 L12-19)

Assuming that only aerosol particles are CCN-active which have an individual residence time in the aerosol chamber that is above $t_{act}$, the theoretical $AF$ can be calculated according to eq. (6). Two different time ranges within the experimental durations need to be considered. If the experimental duration $t$ is below $t_{act}$, $AF$ is 0 as even the particles that entered the aerosol chamber at the very beginning have an individual residence time shorter than $t_{act}$ and therefore cannot be CCN active yet (eq. 6a). If the experimental duration $t$ is above $t_{act}$, $AF$ is greater than 0 as a subset of the particles will have an individual residence time longer than $t_{act}$ and therefore can be CCN active (eq. 6b). Application of eq. (3), which describes RTD and rearrangement of eq. 6 allows for the calculation of the activation time $t_{act}$ based on an experimentally determined $AF$ as shown in eq. (7). This equation is valid throughout the entire filling regime including steady state.

The AF-curve in a CSTR appears to be much smoother for the case of a Gaussian shape $t_{act}$-distribution. This is due to the fact that this scenario comprises particles with different individual $t_{act}$'s as introduced in section 3.3 "Impact of the activation time distribution on the individual $AF$". A small subset of the particle population activates much earlier than 180min, therefore a global $AF$ = 0.01 is reached much earlier as well. At the same time an $AF$=1 is reached much later than in the case of the step-wise $t_{act}$-distribution because of particles with a significantly higher individual $t_{act}$. The difference in the tact-onsets from Table 2 is therefore no artifact from the calculations but a direct (and expected) result of applying the $t_{act}$ – concept to the CSTR-approach. The text discussing Table 2 now reads:

(P15 L1-18)

As can be seen in Table 2, the individual values deviate with the biggest deviation in the case of $t_{act}$ -onset. However, the presented deviations are solely caused by the underlying

distributions of the activation time. In addition, $t_{act}$-onset, $t_{act}$, and $t_{act}0.5$ are determined at different experimental times. While $t_{act}$-onset is directly determined by measuring the entire particle population within the CSTR (global $AF$), $t_{act}$ is calculated from the global $AF$ in steady state and $t_{act}0.5$ is obtained from the activation time distribution itself. In the case of $t_{act}$-onset, there is a significant share of particles activating significantly earlier than the nominal activation time ($\mu$ = 180) in the case of a Gaussian distribution. Therefore, a fraction of 0.01 of CCN active particles within the entire particle population is already present after 87 min. Opposite to this, the threshold value of 0.01 is crossed later than the nominal activation time in the case of the step distribution. This is because even though every single particle activates after exactly 180 min of individual aging time, it takes some additional time before a fraction of 0.01 of the entire particle population within the CSTR is older than 180 min leading to a $t_{act}$-onset of 185 min. The difference of 10 min in $t_{act}$ between the two $P(t_{act})$-approaches is due to the application of eq. (8) which allows for the calculation of $t_{act}$ from the global $AF$ in steady state. Strictly speaking, this equation is defined for the ideal step function ($P_{step}(t_{act})$) only. Therefore the higher global $AF$ value for $P_{gaussian}(t_{act})$ in steady state has to lead to a lower $t_{act}$ value compared to $P_{step}(t_{act})$. Note, $t_{act}0.5$ is referring to the particle activation distribution $P(t_{act})$ only leading to a concordant value of 180 min in both cases. This can be seen in Graph C of Fig. 4, where 0.5 of the particles with a residence time equal to the nominal activation time are activated in the case of a Gaussian distribution corresponding to $t_{act}0.5$. In the case of a step function, all particles are activated once the respective particle population is older than $t_{act}$. In the following we will show how the actual activation time distribution $P(t_{act})$ can be retrieved from real CSTR experimental data.

5. Application to experimental data: No major comments here other than those that stem from the previous section.

6. Application of tact to other continuous flow chambers: I think this section would be very useful for OFR users on how to use OFRs for CCN meaurements! Still, neither the PAM nor the TPOT are CSTRs, so how have the authors applied tact to their RTDs? Also, what if aerosol content is not well known (e.g., field sampling)? How is their mathematical framework applied? I am still unclear as to what parameters are needed experimentally.

The PAM and TPOT chambers are indeed no CSTRs and cannot be described with the CSTR-specific mathematical framework introduced here. However, they show a significant residence time distribution due the mixing along the flow path as already discussed by e.g. Lambe et all. 2011 and Mitroo et al. 2018, therefore it is possible to apply the $t_{act}$ concept. An RTD means that the aerosol particles that leave the OFR stayed inside the chamber for different individual residence times. If the measured $AF$ behind the OFR is 0.3, we raise the question which particles of the whole aerosol particle distribution are the CCN-active particles. The $t_{act}$ concept implies that only the oldest 30 % of the particles are CCN-active and the youngest 70 % are CCN-inactive. The time that separates the youngest and CCN-inactive 70 % from oldest and CCN-active 30 % is the necessary aging time $t_{act}$.

In principle it is also possible that young and old particle activate equally well, however this seems to be unlikely for the BES-particles discussed here. Furthermore even this behavior could be captured be the $t_{act}$-distribution. In this unlikely case the activation time distribution would be a horizontal line ($P(t_{act})$ = 0.3 ) and neither a peak nor a Gaussian shape distribution.

In this manuscript we present 2 scenarios that illustrate the application of the $t_{act}$-concept to the RTDs reported by Lambe et al. 2011 (Figure 7 P19). In the first scenario (High-OH) we calculated the fraction of particle older than 40 s in the PAM and TPOT chamber, respectively. In the second scenario we did the same with a $t_{act}$ of 180 s. As can be seen, the fraction of particle older than these threshold times $t_{act}$ varies between both chambers. Since in our definition only the oldest particle can be CCN-active this leads to different values of $AF$-values depending on the chamber. In the high-OH scenario the $AF$ in the TPOT is higher than in the PAM chamber. In the low-OH scenario the $AF$ in the TPOT is lower than in the PAM chamber. The results obtained based on the application of our $t_{act}$ concept agree well with the trend reported by Lambe et al 2011. The respective text in section 5 "Application of $t_{act}$ to other continuous flow aerosol chambers" now reads:

(P19 L7 – P20 L13)

In the following we show that the application of $t_{act}$ can contribute significantly to the explanation of the aforementioned discrepancies in terms of CCN-activity of the BES particles.

In Fig. 7 we show the RTDs for 145 nm BES particles using the parameters for the bimodal Taylor-dispersion model given by Lambe et al., (2011) in Appendix A4 (Fig A3). We normalize the area under the curve to be one causing the area under each curve to be directly proportional to the $AF$s for a better visual comparison. Here, PAM chamber data is indicated by the dotted line/green area and TPOT chamber data is indicated by the dashed line/blue area. As can be seen, the two curves are not perfectly superimposed with the peak of the PAM chamber RTD being earlier than in the TPOT chamber RTD followed by a steep decline causing the two curves to cross at approximately 40 s. Overall the PAM chamber RTD (dotted line) shows a stronger dispersion causing the two lines to cross again at approximately 180 s.

Assuming a high OH-concentration leads to a higher reaction speed and therefore shorter $t_{act}$ we present two scenarios. Scenario A representing a high OH-concentration is based on a $t_{act}$ of 40 s (Fig. 7 A). Scenario B representing a low OH-concentration is based on a $t_{act}$ of 180 s (Fig. 7 B). In both cases the BES-particles show CCN-activity, but the global $AF$ differs significantly between both chamber types. In the high-OH scenario A, the TPOT chamber is more efficient in producing CCN-active BES-particle ($AF_{TPOT}$ = 0.892; blue area) than the PAM chamber ($AF_{PAM}$ = 0.655; green area) as can be perceived by the blue area being larger than the green area in the left panel. Opposite to this, the PAM chamber is more efficient ($AF_{PAM}$ = 0.082) than the TPOT chamber ($AF_{TPOT}$ = 0.047) in case of the low-OH scenario B, as can be seen in the right panel. These calculations indicate how the new $t_{act}$ concept can contribute to the understanding and interpretation of experimental data that has been acquired in non-CSTR reaction chambers. At the same average OH-exposure, aging in different OFRs causes the same global $AF$ only if the RTDs are the same. Since the RTDs of

the PAM chamber and the TPOT chamber are not the same, the same global $AF$ can only be obtained if the $t_{act}$'s differ. Three examples of how $t_{act}$ has to deviate between the PAM chamber and TPOT chamber to lead to the same global $AF$ are given in the supplement.

However this a qualitative application of the $t_{act}$-concept. We added following section that mentions what would be needed for quantitative application of $t_{act}$ – concept.

(P20 L23-36)

Up to now, the discussion did not include many important processes that are relevant in aging chambers e.g. particle wall-interaction, gas-phase-partitioning, fluctuating input concentrations while field measurements, or inhomogeneities inside the OFR. These aspects are important for many processes such as the formation of SOA and can be incorporated to the $t_{act}$-concept by modifying eq. (13). As the actual calculation requires a multidimensional data array and detailed knowledge about the chamber of interest, this subject matter is beyond the scope of this publication and will not be discussed further. Nevertheless, the overall conclusion is that application of the original/non-adjusted $t_{act}$-concept can explain why measurements within different OFR chambers agree in parameters, which dependent on the bulk properties of the aerosol particle population (e.g. average O:C ratio) and at the same time disagree in parameters, which are dependent on the condition/status of the individual particle (e.g. CCN-activity). Therefore, we suggest to apply the concept of the activation time $t_{act}$ or the activation time distribution $P(t_{act})$ as metric in addition to calculating average values, such as the global $AF$ and OH-exposure if following conditions are met. One, the system or parameter of interest can be described as a binary system and undergoes step-wise / non-gradual transitions such as CCN-activity. Two, the OFR used has a RTD broad enough to influence the outcome. Three, the conditions inside the reactor are either homogeneous or a correction for inhomogeneities (e.g. different oxidants concentrations inside the reactor) is implemented.

We want to point out that every additional variable increases the complexity of the math exponentially. Fluctuation input concentration during field sampling can be rather easily implemented. Fluctuation input concentrations distort the RTD in the OFR. This could be approach by normalizing the RTD not to 1 but to the particle input concentration and then to continuously integrate over all RTD for all time steps.

Having inhomogeneous conditions inside the OFR e.g. different concentration of OH-radical is rather difficult to implement, which was one motivation to use an internally mixed chamber.

7. Conclusion: No major comments here.

Minor Comments:

P1 L7: Arguably a small point, but I'd encourage a revision of "achieve extended observation times" to "obtain measurable reaction rates, due to long residence times" or words to that effect. I think owing to the small reactor design community there's often scant attention paid to the details of a reactor and how it operates by the average reader; and this work has potential for a broad audience, so ensuring the readers become educated about reactor design and meaningful parameters to evaluate its performance is important in my opinion.

P1 L8-9: Change "…in a CSTR mode." to "…as a CSTR." Also, if the authors wish to introduce the concept of a CSTR this early on, perhaps the opening sentence could mention the use of environmental chambers as batch or semi-batch reactors.

We understand and appreciate the suggestion of the reviewer to implement batch and semi-batch reactor early on in manuscript. However we present a different experimental approach and refrain from mentioning batch-mode operation of aerosol chambers in the abstract. In section "1 Motivation" we discuss environmental chambers as well as PFR/OFR and compare them with CSTRs (P1 L27 – P2 L11). The respective text now reads:

(P1 L8-10)

We present an experimental approach that enables long observation times at atmospherically relevant reactant concentrations in small chamber volumes by operating the aerosol chamber as a Continuous flow Stirred Tank Reactor (CSTR).

P1 L9-10: Mean values of what? Perhaps 'its mean value', referring to the data.

In OFR and environmental chamber experiments an average exposure to certain oxidants (e.g. photochemical age) is typically reported. Since this does not always represent the aging conditions well we introduced the $t_{act}$ as an alternative metric.

P1 L10-13

We developed a mathematical framework that allows the retrieval of data beyond calculating mean values such as $O_3$-exposure or equivalent atmospheric aging time, using the new metric: activation time ($t_{act}$).

P1 L14: Are the authors referring to the PAM? Perhaps give an example.
P1 L14-15: Rephrase sentence. tact explains or tact helps explain? What are the different chambers?
Are they smog chambers vs. OFRs? Are they aerosol flow tubes vs. OFRs?

As the reviewer encouraged as to give OFRs more space, we now mention the intercomparion of the PAM and TPOT in the abstract and mention that the $t_{act}$-concept can be applied to different OFRs in general and that it can explain discrepancies found between the TPOT and PAM chamber.

P1 L15-18:

Furthermore, we show how $t_{act}$ can be applied for the analysis of data originating from other reactor types such as Oxidation Flow Reactors (OFR), which are widely

used in atmospheric sciences. The new $t_{act}$ concept significantly supports the understanding of data acquired in OFRs especially these of deviating experimental results in intercomparison campaigns.

P1 L17: This may apply throughout but "Aerosol particles" should be "Aerosols".

In atmospheric sciences the term "aerosols" is often used when exclusively the condensed phase suspended in a gaseous phase is meant. However general definition of the term "aerosol" actually includes the condensed phase as well as gaseous phase. In the attempt to reach a broad audience even beyond the atmospheric science community, we remain with the wording "aerosol particle".

P1 L17-18: Small detail, but stating aerosols are 'emitted' into the atmosphere implies they may not be generated by nucleation. The authors can consider the following rephrase: "Atmospheric aerosols undergo various reactions; the timescale for which depends on their lifetime."

Please note, in accordance with the discussion above "Atmospheric aerosols" refers to the gaseous components as well. The sentence now reads:

P1 L21-22

Atmospheric aerosols undergo various chemical reactions and physical modification processes once they are emitted into the atmosphere.

P1 L19: Citation for sea salt aerosol lifespan (textbooks are appropriate as well) please.

Both numbers can be found in Textor et al.

Also, replace comma with '"whereas".

done

P1 L21: Please check 'aerosol particles', as per my previous comment.

We refer to aerosol particle only when excluding the surrounding gas phase, therefore we keep the original wording

P1 L23: "task" should be plural; also, I'd change 'the investigation of aerosol' with 'understanding atmospheric aging'.

task changed to tasks

We present a general experimental approach to study aerosols. This is not limited to "atmospheric aging" even though this is the main focus.

P1 L23-27: I'd encourage the authors to reword this section and not to gloss over how large

reactor are 'technically' unfeasible, but instead be more explicit (e.g., wall losses, time dependencies, etc.). Also, I don't think the SAPHIR has a set 16 h operation time, so perhaps the authors can say '12-24h' to indicate a range.

We appreciate the comment of the reviewer and rephrase the appropriate sentence. As Rollins et al., 2009 explicitly mentioned an aging time of 16h we continue to include this number:

P1 L28-30

Here aging durations of up to 16 hours and beyond at atmospherically relevant reactant concentrations can be achieved, which has been shown e.g. for the SAPHIR chamber of FZ Julich with a volume of 270 m$^3$ (Rohrer et al., 2005; Rollins et al., 2009).

P1 L28-29: This is a technical detail I would like not to be overlooked. "…in order to reduce the reaction time…" is not the objective; it is a consequence (advantageous, admittedly, for investigating physiochemical properties of SOA or LVOCs). The objective is to artificially augment the reaction rate. I believe the authors know this, but a reader may not, and I encourage the idea of having these details be clear. This is important work and should be presented as such!

We appreciate the comment of the reviewer and the revised sentence now reads:

P1 L31-34

The second option is to increase the concentration of the reactive compounds such as oxidants and aerosol particles, in order to trigger higher reaction rates and thereby reduce the reaction time (George et al., 2007; Huang et al., 2017; Kang et al., 2007; Keller and Burtscher, 2012; Simonen et al., 2017). This allows to significantly reduce the volume of the aerosol chamber.

P2 L4: Please consider adding a citation of Renbaum and Smith, doi: 10.5194/acp-11-6881-2011

done

P2 L6: In the engineering literature, CSTRs are well described, but the acronym is use as a general term for any well-mixed vessel. The authors choose to apply (or design?) a CSTR for their work in addition to the mathematical expression for tact. Have I understood this correctly? Might I suggest them to give their reactor a more personalized name?

We agree to the reviewer. We operate an aerosol chamber in CSTR mode. In the previous of the manuscript the distinction between the different concepts was not clear. We changed the wording throughout the manuscript to "aerosol chamber operated in CSTR mode". This does not include a new chamber design, but is a different experimental approach.

Since CSTRs are known for quite a while we also cannot give it a personalized name. There is also no new reactor design involved like in the most OFRs. Furthermore the tank used here

was so well mixed that no deviation from a perfect mixing could be detected. The respective sentence now reads.

P2 L3-5

Here we present an experimental approach that can be used to achieve long aerosol aging times with neither need for large chamber volumes nor high reactant concentrations by operating an aerosol chamber in the Continuous flow Stirred Tank Reactor (CSTR) mode.

P2 L8: An ideal CSTR is perfectly mixed. A real CSTR is well-mixed.

We agree with the reviewer's statement. We adjusted the wording throughout the manuscript to avoid any confusion. We only use "perfectly mixed" in context of an ideal CSTR. The respective sentence now reads:

P2 L14

The volume of the CSTR is actively stirred in order to achieve a homogenous aerosol mixture.

P2 L9: Perhaps the authors can rephrase "…close to real processes in the atmosphere…" to "…mimics mixing in the free troposphere more accurately than [other reactors]". But more importantly, is the mixing state in this reactor important because it mimics atmospheric dynamics or because it allows more accurate data retrieval from laboratory experiments?

This reference to atmospheric processes shall give the reader a better understanding what it means to operate an aerosol chamber in CSTR-mode. We clarified that we compare the CSTR-approach and the atmosphere in terms of mixing aerosols and measuring a non-uniformly aged aerosol. The respective text now reads:

P2 L14-19

Due to the mixing, the aerosol that is continuously extracted for analysis consists of a well-defined mixture of aerosols at different aging stages. From this perspective, the CSTR approach is closer to atmospheric processes than other reactor types as in the real atmosphere except for individual plume emissions aerosols are rather continuously emitted, mixed, and removed. This results in a mixture of aerosols at different aging stages, but of course, the atmospheric mixture is less well defined compared to an aerosol in a CSTR.

P2 L11-12: That's absolutely the case for a CSTR! At steady state, the distribution of ages is fixed, and is dependent only on reactor volume and flowrate. I think this needs to be clearer.

We focus on this specific aspect in section 2 "Introduction of the CSTR" and the respective text now reads:

P3 L3-5

The key parameter for the description of reactions within a CSTR is the hydrodynamic residence time ($\tau_{CSTR}$) which is also the mean residence time. It can be obtained from the reactor volume ($V_{CSTR}$) and the volumetric flow through the CSTR ($\dot{V}$) as shown in eq. (1) (Levenspiel, 1999).

P2 L14: In addition to refining the sentence to make it sound less informal, I would encourage the authors to cite also Levenspiel's Omnibook (I think they cite it later, but it's missing in the Reference Section).

> done

P2 L19: I'm confused, what do the authors mean when they say 'non-gradual' (see Major Comments)? Do they mean time-dependent? Do they mean non-steady state? This is a key concept in their work, so I would ask them to define it explicitly for the reader.

"Non-gradual" describes changes like phase-transitions where a property shows a step-wise change. The opposite is a gradual or continuous change of a property. An example would be the freezing of water. Below or above 0°C the density of liquid water/ice changes gradually with the temperature. When excluding super-critical conditions, at 0°C the density does not change gradually but changes step-wise. To clarify what is meant by "non-gradual", we extended the introduction of this phrase and added "step-wise change" and "transition between binary states" as alternative explanations. "Transition between binary states" hereby means that a system/particle can be described by two distinct states. Either a droplet is liquid or frozen. A transition from one state to another one can therefore be described as "non-gradual". Further, we added a list of possible transition that can be described as "non-gradual", "step-wise change" and "transition between binary states"

P2 L24-39

Such transitions in binary systems are step-wise, also referred to as non-gradual changes in a particle property, such as:

1) Freezing of a water droplet: Step-wise and therefore non-gradual change in the particle density; the water is either in liquid or solid state.

2) Deliquescence of soluble aerosol particles: The particles show a step-wise i.e. non-gradual increase in diameter.

Binary particle properties are not necessarily intrinsic particle properties, but can also be defined by the measurement protocol.

3) CCN-activity: The chemical and physical properties of an aerosol particle can vary, but the particle is either CCN-inactive or CCN-active at a defined super saturation (SS).

4) Growth beyond a threshold: Condensational growth of an aerosol particle leads to a continuous and gradual increase of the particle diameter. A binary system can be defined by introducing a threshold diameter that can be arbitrarily chosen. The aerosol particle is either smaller or larger than this defined threshold diameter. The same holds true when particles are separated e.g. in aerosol impactors.

Therefore, the concept of non-gradual transitions/transitions within binary systems can be used to describe a multitude of changes in particle properties.

P3 L3-4: More than a physico-chemical (physio-chemical?) perspective, I'd say from a flow or mixing state perspective. Also, PFR can be placed in acronym in brackets (although PFRs can be mentioned in the introduction), and CSTR has already been spelled out earlier, so just the acronym should suffice here.

Changed from

From the physico-chemical perspective

to

From a technical perspective.

PFR acronym added

P3 L9: Can the authors make the case that environmental / smog chambers are batch-type reactors?

Changed from:

In a batch-reactor the reactants are introduced at the beginning of the experiment aiming for homogeneity and then the reaction
is allowed to procede. The composition throughout the vessel is homogeneous but evolving in time, therefore no steady state
conditions are ever achieved. After a certain reaction time the sample is discharged or collected and subjected to further
analysis.

To (P3 L17-23):

In an aerosol chamber operated in batch mode, the reaction volume is first filled with the sample aerosol as fast as possible to achieve high homogeneity of the sample. After the desired start concentration is reached further addition of the sample aerosol is stopped and the aging is initiated e.g. by addition of the oxidant. This point in time is generally defined as the start of the experiment and referred to as $t = 0$. Data acquisition of the ageing sample takes place while the reaction volume is flushed with sample-free gas. The composition throughout the chamber is homogeneous but evolving in time, therefore no steady state conditions are ever achieved. This concept is used to operate many large scale environmental chambers (Cocker et al., 2001; Leskinen et al., 2015; Nordin et al., 2013; Paulsen et al., 2005; Platt et al., 2013; Presto et al., 2005; Rohrer et al., 2005).

P3 L9: Again, I would urge the authors to be detailed. A PFR (which is the idealized reactor design on which flow tubes are built) allows no axial mixing (as the authors point out), but is perfectly mixed radially! The ADM (mentioned by Lambe et al.) allows for deviation from the

PFR and is closer to describing flow tubes, but that discussion can be briefly mentioned, if needed at all.

The ADM describes how the RTD in a real OFR deviates from the RTD in an ideal PFR. For the application of the $t_{act}$-concept it is sufficient that the RTD is known. Therefore we do include different methods to characterize the RTD. We mention the PFR as reactor concept and point out that it is approximated by OFRs

Changed from

> A flow tube is a steady state reactor in which no mixing along the flow path takes place resulting in a constant output of products depending on the residence time within the reactor.

To (P3 L24-31)

A PFR is a steady state reactor in which no mixing along the flow path (axial mixing) but perfect mixing perpendicular to the flow (radial mixing) takes place. Further, a continuous feed-in of reactants and withdrawal of sample take place at equal flow rates simultaneously. This results in a constant composition of the output solely depending on the residence time within the reactor. This ideal system is approximated by many Oxidation Flow Reactors (OFR) e.g. PAM chamber (George et al., 2007), TPOT Chamber (Kang et al., 2007), Micro Smog Chamber (MSC; Keller and Burtscher, 2012), or the TUT Secondary Aerosol Reactor (TSAR ; Simonen et al., 2017). The main difference between an ideal PFR and real OFRs is that in OFRs significant but unintentional mixing of the aerosol along the flow path takes place (Mitroo et al., 2018). Therefore, OFRs show a significant residence distribution.

P3 L17: Residence time of what? The large chambers?

> "in large chambers" added

P3 L19: Consider replacing "During a subsequent…" with "Following steady state, upon shut down, is the…"

We thank the reviewer for the suggestion but we want to highlight the fact that the purpose of the flushing regime is to shut down the chamber exclusively but it allows to investigate the parameter of interest in an addition regime.

P3 L20: To better illustrate their point, I think the authors can put an arbitrary schematic in the Supplement rather than alluding to a figure that has not yet been explained.

> We remove the reference to Fig. 3 for clarity.

P3 L21: I don't think 'hydrodynamic' is necessary, but I could be wrong.

We clarified that it is the hydrodynamic residence time as well as the mean residence time. In case the reactor would have dead zones this would not the case.

Changed from:

The key parameter for the description of reactions within a CSTR is the hydrodynamic mean residence time ($\tau_{CSTR}$)

To (P4 L3-4):

The key parameter for the description of reactions within a CSTR is the hydrodynamic residence time ($\tau_{CSTR}$) which is also the mean residence time

Equation (2): A suggestion to simplify notation, perhaps the subscript 'CSTR' can be removed, seen as it is implied. Also, (t) can be placed outside of the square brackets, as can the subscript 'feed-in', which I would also suggest be replaced with a subscript zero.

Using $A_0$ is indeed a more common notification to describe a single-step process. Here we focus on a two-step process (filling and flushing) therefore it would be necessary to define two different $A_0$ ($A_{CSTR}$ = 0 and $A(t=t_{switch})$) which can confuse the reader. Since the square brackets are intended to indicate "concentration", we remain in the terminology used so far.

P4 L1: This is a good point by the authors! I would encourage a citation of Lambe et al., seen as what the authors are describing here is essentially the result of a tracer study (A is a chemically inert tracer essentially).

Unfortunately, the authors are not aware of the appropriate publication by Lambe et al. that the reviewer refers to. We want to highlight the fact that the beginning of steady state has to be chosen depending on the experimental conditions (e.g. resolution of detection) and can be different in different experiments.

P4 L11: I would encourage a citation of Mitroo et al.
Equations (3-4): These are E and F-curves as described by Mitroo et al.; it may be worthwhile to mention.

We added a sentence to explain our reasoning for the differing wording.

P4 L28-30:

Note, while we choose RTD($t$) and RTD$_{sum}$($t$) for a more intuitive denotation, generally E(t) and F(t), respectively, are the official formula symbols especially in the engineering community (Levenspiel, 1999).

P6 L16-19: This needs to go either at the end of the introduction, or at P2 L19 in my view.

We believe we have addressed this comment in the new version of the manuscript by significantly extending the introduction of the term "non-gradual".

P6 L20-21: This sentence needs to be rewritten as it is too handwavy and comes across as pseudoscience. "…a particle that undergoes changes that result in a non-gradual

transition…" made no sense in my mind until I finished reading the manuscript. Could the authors come up with a physical example to help convey what change has been 'undergone' that resulted in a 'transition'? Or is the 'change' itself rapid (e.g., heterogeneous nucleation)? Are the authors implying they can model a process this fast as a function of time, and decouple it from other timescales within the reactor? Is a CSTR the best approach?

We thank the reviewer for his comment and aim to improve the understanding by specifying what kind of transitions we refer to.

Changed from:

> If the all other parameters stay constant, while a particles undergoes changes that result in a non-gradual transitions, this transition can be described as a function of time.

To P6 L29-35:

We may assume a system in which all external parameters stay constant but the particle itself undergoes a continuous transformation, e.g. due to oxidation. After a certain period of time, this continuous transformation, in this specific case oxidation, can lead to a change in a binary property, e.g. CCN-activity. Ultimately, the step-wise or non-gradual transition is a function of time. We define the required time span (e.g. necessary aging time) that leads to a change in a specific particle property, resulting in a transition in a binary system in another particle property as the activation time ($t_{act}$). This concept is generally valid and can be applied to any kind of transition in a system defined as binary either by intrinsic or operational parameters.

P10 L4: "aerosol particles"

See discussion above

P10 L6: "aerosol particles", but more importantly, what properties are distributed around a mean value? If they are physical (e.g., dpg, sigmag, etc.) maybe. If they are chemical (e.g., nitrate content) then not really.

From a very fundamental point of view, even the chemical compositions of aerosol particles from rather pure source (maybe sea salt ) varies slightly between the particles. Often this distribution is so narrow that it does not matter. That is why it is often ignored.

To avoid any confusion of the reader we change the wording from:

> However, this is rarely as in reality an aerosol population consists of aerosol particles, whose properties are typically distributed around a mean-value (e.g. the mode of a particle size distribution).

To (P11 L11-23):

However, this is not the case for many parameters. In case of the particle diameter, for example, every aerosol particle has its individual diameter and the total population can be

described by a distribution of particle diameters around a mean diameter. An eventual size-selection does impact the mean diameter and the width of the distribution. Still, the size selected particles will not have the identical diameter.

P10 L7-9: I don't follow the logic here. If I understand correctly, the authors are saying that, due to multiplicity of charges on some aerosols, an aerosol population that follows a lognormal distribution if plotted by mobility diameter doesn't follow a lognormal distribution by aerodynamic diameter? I don't see how an aerosol population that is unimodal in mobility diameter can be multimodal in aerodynamic (or geometric) diameter.

The mobility diameter is a function of size and shape of the particles. The aerodynamic diameter is a function of the density as well. If the effective density of aerosol particle changes with the diameter, both size distribution can be quite different. For soot particles this often the case. Additionally a DMA selects particles that are monodisperse with respect to their mobility in an electric field. Nevertheless the same particles have a multimodal distribution of their mobility diameters due to different charges.

We clarified this in the manuscript.

Changed from:

> While the aerosol population might be mono-modal and narrowly-distributed with respect to one parameter such as the aerosol particle's electrical mobility diameter, it can be multi-modal and broader distributed with respect to another parameter (e.g. aerodynamic diameter).

To:

> Furthermore the aerosol population might be mono-modal and narrowly-distributed with respect to one parameter such as the aerosol particles electrical mobility diameter, but it can be multi-modal or broader distributed with respect to another parameter e.g. the aerodynamic diameter. Therefore, it has to be expected that the activation time ($t_{act}$) is also characterized by a distribution.

P10 L15: Maybe "…has the potential to activate." instead of "…activates.", because after t=180 min, they don't all activate.

Based on our definition all particles older than $t_{act}$=180 min are CCN active, but not all particles are older than $t_{act}$. We thank the reviewer for the comment and the sentence now ready:

P11 L24-25:

This curve was calculated based on the assumption of uniformity, i.e. every aerosol particle that is older than $t_{act}$ = 180 min is CCN active.

P10 L17: Why was 30 min chosen as standard deviation?

The numbers presented here are chosen to represent an exemplary Gaussian distribution. In addition, these values are close to our experimental value and we hope to improve the readability of the text by avoiding to discuss the same aspect with significantly different values depending on the section of the manuscript (3.3 vs. 4). The respective sentence now reads:

P11 L26-29:

> To discuss the impact of an activation time distribution $P(t_{act})$ on the evolution of $AF$ in a CSTR we consider a model system with $P(t_{act})$ representing a Gaussian distribution with an exemplary mean ($\mu$) of 180 min and an exemplary standard deviation ($\sigma$) of 30 min (eq. (12)).

P14 L1: Unless I'm mistaken, tacts don't really differ; only tact-onset for PGaussian differs.

This is a theoretical comparison of different ways to obtain $t_{act}$. The system is idealized and therefor no instrumental uncertainties affect the outcome. Therefore any deviation between the numbers is significant. We now have extended the discussions of the numbers presented in Table 2 in order to present the comparability of our parameters to the literature. The respective text now reads:

P14 L1-18

As can be seen in Table 2, the individual values deviate with the biggest deviation in the case of $t_{act}$ -onset. However, the presented deviations are solely caused by the underlying distributions of the activation time. In addition, $t_{act}$-onset, $t_{act}$, and $t_{act}0.5$ are determined at different experimental times. While $t_{act}$-onset is directly determined by measuring the entire particle population within the CSTR (global $AF$), $t_{act}$ is calculated from the global $AF$ in steady state and $t_{act}0.5$ is obtained from the activation time distribution itself. In the case of $t_{act}$-onset, there is a significant share of particles activating significantly earlier than the nominal activation time ($\mu = 180$) in the case of a Gaussian distribution. Therefore, a fraction of 0.01 of CCN active particles within the entire particle population is already present after 87 min. Opposite to this, the threshold value of 0.01 is crossed later than the nominal activation time in the case of the step distribution. This is because even though every single particle activates after exactly 180 min of individual aging time, it takes some additional time before a fraction of 0.01 of the entire particle population within the CSTR is older than 180 min leading to a $t_{act}$-onset of 185 min. The difference of 10 min in $t_{act}$ between the two $P(t_{act})$-approaches is due to the application of eq. (8) which allows for the calculation of $t_{act}$ from the global $AF$ in steady state. Strictly speaking, this equation is defined for the ideal step function ($P_{step}(t_{act})$) only. Therefore the higher global $AF$ value for $P_{gaussian}(t_{act})$ in steady state has to lead to a lower $t_{act}$ value compared to $P_{step}(t_{act})$. Note, $t_{act}0.5$ is referring to the particle activation distribution $P(t_{act})$ only leading to a concordant value of 180 min in both cases. This can be seen in Graph C of Fig. 4, where 0.5 of the particles with a residence time equal to the nominal activation time are activated in the case of a Gaussian distribution corresponding to $t_{act}0.5$. In the case of a step function, all particles are activated once the respective particle population is older than $t_{act}$. In the following we will show how the actual activation time distribution $P(t_{act})$ can be retrieved from real CSTR experimental data.

P14 L12: Fix "tact", but more importantly, please address the Major Comment surrounding this sentence (the lag in Fig. 3 before tact).

done

P14 L13: Fix "P(tact)"

done

P14 L15: Fix "Pstep(tact)".

done

P13 L5-6: Please provide appropriate citations.

Changed from:

> In the case of CCN-activation it is either an SS-onset often characterized by a minimum threshold (e.g. 1 % **AF**) or a critical SS, when 50 % of the particles activate.

To (P14 L11-14):

> Results from batch chamber experiments as well as from oxidation flow reactor experiments are often presented in terms of SS-onset or critical SS. While the SS-onset is defined by a minimum threshold (e.g. 0.01 *AF*) the critical SS is reached when 0.5 of the particles activate (Friedman et al., 2011; Koehler et al., 2009; Rose et al., 2007).

P14 L21: I would appreciate either a description of the chamber or literature that describes it. I'd really like to know, as I think is important for the reader, if this chamber is indeed well mixed (does it have impellers, fans, baffles?) to where the equations can be applied to the data, or is this chamber not really well mixed? What about residence time in the tubing? The tracer data may require some convincing (see four comments down P15 L6).

We picked up this comments and implemented it in the manuscript.

Changed from:

> In the laboratories at ETH Zurich we performed aging experiments in a 2.78 m3 stainless steel aerosol chamber run in CSTR mode.

To (P15 L20-24):

In the laboratories at ETH Zurich we performed aging experiments in a 2.78 $m^3$ stainless steel aerosol chamber operated in CSTR mode. A detailed description of the chamber can be found in Kanji et al., (2013). The chamber was actively mixed with a fan, but had no further features to enhance mixing e.g. baffles. All instruments were connected to the chamber with stainless steel tubing with 4 mm inner diameter. Since the maximal tubing length from the

CSTR chamber to the analysis instruments was 3 m the impact on the overall residence time is negligible.

P14 L22: For those not familiar with soot generation, what is a miniCAST, set point 6?

Changed from:

> We investigated the change in CCN-activity of soot particles rich in organic carbon from propane combustion (miniCAST, set point 6) due to heterogeneous ozone oxidation.

To (P15 L25-29):

We investigated the change in CCN-activity of soot particles rich in organic carbon due to heterogeneous ozone oxidation. The soot particles were generated with the miniature Combustion Aerosol STandard (miniCAST, Model 4200, Jing Ltd., Zollikofen, Switzerland) which is propelled with propane and operates with a laminar diffusion flame. The miniCAST was operated under fuel-rich conditions (set point 6 according to the manual) in order to generate a soot which was rich in organic compounds (fuel-to-air ration: 1.03).

P14 L30-35: Would the authors see fit to put these two points at the end of the Introduction Section?

We thank the reviewer for this suggestion, but in view of the complex nature of the concept discussed within this manuscript the authors are worried that moving this information to the beginning of the manuscript might confuse readers not that familiar with chemical reaction chamber designs as the reviewer.

P14 L31: Again, I'd encourage the authors to refrain from using the word 'perfectly mixed' when talking about a real reactor. Might I suggest 'well-mixed'. More to my point: no RTD is available until Fig. 6; can a description of the chamber, or literature on it be presented?

Changed from:

> 1) Can the aerosol chamber be operated in CSTR-mode for up to 12 h which requires a constant aerosol feed-in flow and a perfect internal mixing?

To (P15 L40-41):

1) Can the aerosol chamber be operated in CSTR-mode throughout an entire day, which requires a constant aerosol feed-in flow and a good internal mixing?

P15 L6: Following the comment above: How the particles depict a CSTR would be more believable if the authors provide some way of showing it. Maybe plot an E-curve for the data and overlay that of an ideal CSTR over it? If I calculated it right, 2.78 m3 / 25 LPM is ~111 min. Why is tact more than twice that? In P7 L7 the authors claim tact is one mean residence time for a CSTR. If their chamber is not as well mixed as believed that's OK, but it should be stated (and at least be better mixed than OFRs!).

The authors apologize for not being clear enough in the definition and introduction of the activation time $t_{act}$, the mean residence time $\tau$, and the individual residence time (i.e. particle age) in the original manuscript. The respective text now reads:

P16 L4-32

The graphs A1 and B1 in Fig. 6 show the particle concentration (black crosses; left axis), the measured global $AF$ (red crosses) and the fitted global $AF$ (blue dashed line, both right axis). The particle number concentration curves (black crosses) follow the theoretical filling and flushing curves as expected in a CSTR (Fig. 3). The slight decline in the concentration in steady state in graph A1 is due to a slight reduction in the particle input concentration that was experienced during the experiment. Visa versa the slight increase in the number concentration in graph B1 is due to a slight increase in the particle input concentration over time.

In the flushing regime the particle number concentration declines exponentially in both experiments. Eq. (5) describes the ideal/theoretical evolution of the particle number concentration in the flushing regime when taking the hydrodynamic residence time $\tau_{CSTR}$ according to eq. 1 into account. In the ideal case the decay is solely caused by the flushing process. In reality, the decay is a combination of flushing as well as additional particle losses e.g. wall losses or coagulation. Therefore, the real residence time can be obtained by fitting equation 5 to the experimental data after rearrangement for $\tau$, to which we refer to as $\tau_{flush}$ from now on (Kulkarni et al., 2011). In both experiments $\tau_{flush}$ coincides at 104 min, which is lower than the hydrodynamic residence time $\tau_{CSTR}$ of 111 min. In other words, the particle concentration declines faster than expected. This difference is caused by particle losses to the chamber wall, which acts as an additional particle sink parallel to flushing and reduces the particle lifetime. Nevertheless, statistical analysis of the experimental data results in purely statistical noise centered on the fitting curve used to determine $\tau_{flush}$. This indicates that in terms of mixing no difference between an ideal CSTR and the aerosol chamber used here can be detected with the applied instrumentation.

When dividing the real particle life time ($\tau_{flush}$) into its individual components, a particle life time upon wall losses ($\tau_{wall\text{-}loss}$) of 1600 min can be determined in accordance with first order wall loss kinetic (Crump et al., 1982; Wang et al., 2018). The influence of particle coagulation can be considered negligible due to the low coagulation rate of 100 nm particle at concentrations of maximum 1500 cm$^{-3}$ (Kulkarni et al., 2011).

Based on the discussion above, the measured $AF$s (red crosses) show the expected change throughout the entire experiment in Fig 6 A1 and B1. In the beginning of both experiments $AF$ is 0. After a minimum aging time each $AF$ starts to increase until it reaches a constant level (A1: $AF$ = 0.091, 1.0 % SS; B1: $AF$ = 0.233, 1.4 % SS). The gaps in the curves during steady state are due to changes in the operation of the CCNC form running on a constant SS (1.0% and 1.4%, respectively) to scanning over a range of SS. In the flushing regime, each measured $AF$ increases exponentially. CCN data could be acquired successfully throughout the entire experiment until the global $AF$ reached ~1.0 (> 1000 min) in the first experiment presented in graph A1. In the second experiment presented in graph B1, instrumental issues

caused the acquisition of the global *AF* to end prematurely after approx. 800 min of experimental duration.

**Tables and Figures:**

Figure 1: Please indicate a unit for the x-axis (I think it's seconds). Also, this figure is confusing because it should just be one curve representative of SS, but the authors mention in the caption "…while flushing the CSTR." I understand what the authors mean, but maybe the reader won't so this figure or its citation in the text should be made clearer.

We thank the reviewer for pointing out this issue. We have redesigned the entire figure to increase its readability. Please note, the values of the x-Axis are multitudes of the hydrodynamic residence time. The caption has been revised appropriately.

Caption change from:

> Figure 1: RTD inside the CSTR for steady state and for different time steps (multiples of $\tau$) while flushing the CSTR. The area below the curve is proportional to the fraction of aerosol particles at a specific residence time. The individual residence time of a specific particle fraction is indicated by the color coding.

To:

> Figure 1: RTD inside the CSTR within steady state (black line) and for different time steps after the CSTR operation was switched to the flushing regime. The area below the curve is proportional to the fraction of aerosol particles at a specific residence time. The individual residence time of a specific particle fraction is indicated by the color-coding. The time on the x-axis is plotted as dimensionless time in multiples of the hydrodynamic residence time $\tau_{CSTR}$.

In addition, we now highlight in the text only the solid black line labeled with "steady state" represents SS while the other curves represent the RTD at increasing time increments after switching to the flushing regime. The respective text now reads:

P3 L7-15

Figure **1** illustrates how the RTD changes in the flushing regime. Note, the time on the x-axis is plotted as dimensionless time in multiples of $\tau$. Each color in the area represents an individual aerosol fraction with a corresponding residence time. Blue stands for the lowest and red for the highest residence times. The dashed black curve labeled "steady-state" represents the RTD in steady state while the other curves show the RTDs for additional time increments after the flushing regime has been initiated ($t_{switch}$). For example, the area und the grey curve labeled "+1 $\tau$" represents the RTD 1 $\tau$ after initiation of the flushing regime. The grey dashed line stands for the activation time $t_{act}$, a threshold time that will be introduced later. Here it marks a threshold time. With increasing flushing time, the fraction of aerosol particles that have an individual residence time higher than this threshold time increases. From some point in time on all particles have crossed this threshold time as is the case for the particles under the light grey curve at "+2 $\tau$" after $t_{switch}$.

Figure 2: No major comments.
Figure 3: No major comments here, other than the curiosity of how a graph like this would look like for a PFR.

The particle concentration / F-curve from Mitroo et.al. would be a Step function. The AF-curve for ideal PFRs with no activation time distribution is either 0 or as step function as well.

Figure 4: No major comments.

Figure 5: Upon seeing Fig. 5, I struggle to now understand Fig. 3 (or, the blue line in Fig. 5). I was under the impression tact is when reactants are introduced. If that is the case, why does the red line show AF > 0 at t < tact? Or am I missing something? A CSTR has no lag by design; only PFRs have lags. Even in the 'filling regime'. I think the root of my misunderstanding can be traced back to P7 L7. Why is AF = 0 when t < tact? Even for a system with no Gaussian spread, purely based on CSTR design, at t = 0+ AF (however small) is non-zero. If the authors can explain their assumption in P7 L7, I think it would clear this up (at least for me).

The authors hope to have improved the introduction of $t_{act}$ chapter 3 in this revised version of the manuscript as $t_{act}$ is the time needed to modify a single particle to such a degree that it is CCN active. $t_{act}$ is **not** the time span since when reactants are introduced into the reaction chamber as this is defined as the experimental duration $t$.

Table 1: No major comments.
Table 2: No major comments on the table itself (maybe capitalize the subscript 'gaussian'?); but I have comments on how the authors choose to explain the difference in values of tact-onset for Step and Gaussian (see comment section).

Done in the comment section

Table 3: No major comments.
Figure 6: No major comments, but I do have a question: it's unclear how the authors' fit matches data well. Was it a fit? E.g., if instead of soot they used salt, what is needed experimentally to determine the blue dotted line in this Figure? Did I miss something in the text?

Changed from:

> The **P($t$**act) presented in A2 and B2 of Fig. 6, respectively, were obtained by performing a curve fitting operation assuming
> **P($t$**act) to be a mono-modal Gaussian distribution and with the parameters **μ** (= **t**act0.5) and **σ** to be optimized.

To:

The graphs A2 and B2 in Fig. 6 show the activation time distribution $P(t_{act})$ (blue solid line) retrieved from the measured global $AF$s. The $P(t_{act})$'s presented were obtained from curve fitting the measured $AF$-curves using eq. (13), which describes the evolution of $AF$ taking the activation time distribution into account. For this, assumptions concerning the type of distribution had to be made. Here, we assumed that $P(t_{act})$ can be described by a mono-modal Gaussian distribution as presented in eq. (12). A brute-force algorithm was used that optimized the characteristic parameters $\mu$ (=mean) and $\sigma$ (=standard deviation) in order to achieve the best fit to the measured global $AF$ using the least-square method. The results of this fitting procedure are presented in Table 3 as well as in A2 and B2 of Fig. 6. In the first experiment with the experimental settings at 1.0 % SS and 100 ppb $O_3$ $\mu$ as well as $\sigma$ of $P(t_{act})$ are larger (253.7 min and 35.5 min) compared to the results obtained for the second experiment at 1.4 % SS and 50 ppb $O_3$ (153.6 min and 24.6 min). From a theoretical perspective, there are two competing aspects. On the one hand, due to the higher ozone concentration the threshold of chemical transformation leading to CCN activity of the particles should be reached earlier. On the other hand, the threshold of chemical transformation should be lower at higher SS. Our results presented here could indicate that the difference in SS in this specific range might be more important than the difference in ozone background concentration within the considered range. At the current stage we cannot draw any final conclusions on how these two competing aspects actually interplay but additional experiments are planned to resolve this issue.

In addition, we list $t_{act}$ obtained from $AF$ during steady state following eq. (8) as described in section 3.1.2 in Table 3. Based on error propagation calculation, the instrumental uncertainty for obtaining $t_{act}$ from steady state is ± 11.6 min. In our experimental setup the differences between $t_{act}$ and $\mu$ are 3.9 min and 2.1 min, respectively, and therefore below the instrumental uncertainties. This is a very beneficial aspect when considering a broad application of the CSTR-concept in atmospheric science experiments. In general, an accurate determination of $P(t_{act})$ requires a sufficiently high time resolution throughout the whole experiment. This can be difficult to achieve depending on the general experimental conditions such as the type of instrument, since running SS-scans with a CCNC can be time consuming. However, if a characterization of the aged aerosol during steady state is sufficiently precise, a potentially time consuming acquisition of a large number of data points for the determination of $P(t_{act})$ does not provide additional benefits.

Figure 7: No major comments, but to be clear, is this illustrative? That AFTPOT > AFPAM at high [OH], and the reverse for low [OH], is subject to experimental data, right?

The specific values for $t_{act}$ are chosen for illustrative purposes. The AF-values are theoretical calculations based on the framework presented here and match the trend seen in the experimental data as presented by Lambe et al. 2011.

---

## Author Comment (AC2) · 22 Feb 2019

**We thank the reviewer for the comprehensive feedback on our work. With the help of the reviewers' comments we greatly improved the understandability of our work and made it more accessible to a broader audience. Detailed answers to the individual comments are given below. For clarity, the reviewers' comments are written in black, and our response in red. Texts from the old version of the manuscript are typed in green and texts from the revised manuscript in blue.**
* * *
**General comments:**

This manuscript presented an improved experimental approach to perform atmospheric oxidation of soot particles using a Continuous Flow Stirred Tank Reactor (CSTR), which enables extended sampling time within a small-size conventional aerosol chamber. A new metric of activation time ($t_{act}$) was developed to characterize the change of activated fraction (AF) in different regimes (i.e., filling, steady state, and flushing) for soot particles following heterogeneous ozone oxidation. Good agreements between theoretical calculations and parameterized CCN activities using $t_{act}$ were achieved for their experimental data. The $t_{act}$ concept was also applied into some previous studies with continuous flow chambers. Discrepancies in the CCN activity of BES particles can be better explained with considering $t_{act}$ and residence time distribution, in comparison to those initially interpreted by the bulk H/C and O/C ratios, which couldn't fully characterize the detailed change in particle chemical compositions. This work is worth further application in atmospheric sciences, yet some details and interpretations could be clarified, reorganized, and improved accordingly. I would recommend for the final publication in AMT upon major revisions, as detailed below.

**Major comments:**

1. In the motivation section (Page 2, Line 19): The "non-gradual transition" case of CCN activation suddenly appeared, with no prior introduction or definition of this new concept (instead, which was included in Sect.3). This content seemed to be disconnected with the information detailed in the last sentence, and I didn't catch the importance/necessity of developing a mathematical analysis for the non-gradual transitions in the following statements.

We understood that the wording we used does not precisely describe to what kind of changes we refer to. Our approach to improve the understandability is to add synonyms commonly used in the atmospheric science community. Additionally we added examples to illustrate this concepts. For the examples we chose common processes investigated by the atmospheric science community. Nevertheless, these concepts are not limited to atmospheric science and can be applied in different fields as well.

In case of the term "non-gradual" we refer to changes like phase-transitions where a property changes step-wise. This is the opposite of a gradual or continuous change of a property. An example would be the freezing of water. Below or above 0°C the density of liquid water/ice changes gradually with the temperature. At 0°C the density does not change

gradually but changes step-wise by jumping between 0.92 g/cm$^3$ and 1.00 g/cm$^3$ . To clarify what is meant by "non-gradual" we extended the introduction of this phrase and added "step-wise change" and "transition between binary states" as alternative explanations. "Transition between binary states" hereby means that a system/particle can be described by two distinct states. Either a droplet is liquid or frozen. A transition from one state to another can be described as "non-gradual" as well.

We added a list of possible transition that can be described as "non-gradual", "step-wise change" and "transition between binary states" (P2 L26-39)

> Such transitions in binary systems are step-wise, also referred to as non-gradual changes in a particle property, such as:
>
> 1) Freezing of a water droplet: Step-wise and therefore non-gradual change in the particle density; the water is either in liquid or solid state.
>
> 2) Deliquescence of soluble aerosol particles: The particles show a step-wise i.e. non-gradual increase in diameter.
>
> Binary particle properties are not necessarily intrinsic particle properties, but can also be defined by the measurement protocol.
>
> 3) CCN-activity: The chemical and physical properties of an aerosol particle can vary, but the particle is either CCN-inactive or CCN-active at a defined super saturation (SS).
>
> 4) Growth beyond a threshold: Condensational growth of an aerosol particle leads to a continuous and gradual increase of the particle diameter. A binary system can be defined by introducing a threshold diameter that can be arbitrarily chosen. The aerosol particle is either smaller or larger than this defined threshold diameter. The same holds true when particles are separated e.g. in aerosol impactors.
>
> Therefore, the concept of non-gradual transitions/transitions within binary systems can be used to describe a multitude of changes in particle properties.

Such transitions between binary states are a necessary requirement for the analysis of data from experiment conducted in a CSTR aerosol chamber. Some of the above mentioned transitions are not triggered by a change in the ambient conditions but by a change of the particle itself e.g. due to aging. Aging processes show typically a time-dependency. We focus on the time needed to modify an aerosol particle to such a degree that it changes its state in a binary system. The example on which we focus is the CCN-activation of soot particles due to oxidation with ozone. Initially, a single soot particle is CCN-inactive at a given super saturation. After a certain aging time it becomes able to accumulate water under super saturated conditions. The chemical modification of the soot particle is a continuous process. The changing CCN-activity is not a continuous process. A particle is either CCN-inactive or CCN-active. The transformation from CCN-inactive to CCN-active is hereby a step-wise change in the particle properties.

The authors have introduced the concept of CSTR and suggested that *"The steady state in the CSTR is characterized by constant concentration of all compounds and constant reaction rates."*. It is a bit confusing that how the assumed "perfectly internal mixing" is achieved, even if without considering the influences of particle wall loss and coagulation during different experimental regimes.

The steady state in CSTR is indeed characterized by constant concentrations. All processes and chemical reactions proceed, but new compounds are constantly introduced to the CSTR as well as old products are continuously removed from the CSTR. After a certain time span the CSTR reaches an dynamic equilibrium and therefore the concentration do not change anymore. The same is valid for oxidation flow reactors like the Potential Aerosol Mass chamber. These kind of aging chamber also operate in steady state mode. In contrast to that, environmental aging chamber operated in batch mode and are not continuously filled with new compounds. These type of chambers have constantly changing conditions.

Particle losses to the chamber wall do occur in CSTR-experiment as well, but due to the constant feed-in of fresh aerosol the Particle-wall-loss-rate reaches a constant speed once the CSTR is in its dynamic equilibrium/ steady state.

The "perfectly mixed" refers to the active mixing inside the CSTR. We used a fan to mix the freshly introduced aerosol with the aerosol that entered the CSTR before. A real CSTR is theoretically never perfectly mixed. However, a perfect internal mixing can be extremely well approximated by actively stirring the aerosol inside the chamber.

Section changed from: (old P2 L6-8)

> The continuous flow stirred tank reactor (CSTR) describes an aerosol chamber, which is continuously filled with an aerosol flow constant in composition over time. The aerosol inside the CSTR is perfectly mixed, therefore a mix of aged and unaged aerosols is continuously extracted from the CSTR for analysis.

To: (new P2 L13-19)

> The CSTR approach describes an aerosol chamber, which is continuously filled with an aerosol flow constant in composition over time. The volume of the CSTR is actively stirred in order to achieve a homogenous aerosol mixture. Due to the mixing, the aerosol that is continuously extracted for analysis consists of a well-defined mixture of aerosols at different aging stages.

How should readers understand the "constant concentration of all compounds" during aging reactions in the CSTR, where the corresponding compositions/concentration of reactants/products are supposed to vary with such processes?

As described above the CSTR can reach a dynamic equilibrium. In this state, particles do react, get lost to the chamber walls, coagulate and so on. Due to the constant feed-in of new aerosol and pumping out old aerosol the overall concentration stays constant over time. We greatly expanded the description of the CSTR and the ongoing processes. (new: P4 L7 – P5 L15)

**Filling regime**

[revised manuscript text omitted]

Another question is about the configuration of the CSTR in this study: did the authors use a real CSTR device for their experiments or not? what kinds of equipment (and how) were actually coupled with the CSTR, in addition to a CCN counter which enables the CCN activation measurements (i.e., the AF results) of aged soot particles?

The CSTR describes a reactor concept that is continuously filled with educts while products are constantly removed. The compounds inside the CSTR are ideally perfectly mixed, which is realized by actively stirring the compound with e.g. a fan.

In the context of atmospheric science, every environmental aging chamber can be operated as a CSTR. The only technical requirement is the presence of e.g. a fan to stir the aerosol. In our experiments we operated a 2.78m³ steel chamber in CSTR-mode .

The description of the aerosol chamber used was extended from: (old P14 L20 - 21)

In the laboratories at ETH Zurich we performed aging experiments in a 2.78 m³ stainless steel aerosol chamber run in CSTR mode.

To: (new P15 L20-29)

In the laboratories at ETH Zurich we performed aging experiments in a 2.78 m³ stainless steel aerosol chamber operated in CSTR mode. A detailed description of the chamber can be found in Kanji et al., (2013). The chamber was actively mixed with a fan, but had no further features to enhance mixing e.g. baffles. All instruments were

connected to the chamber with stainless steel tubing with 4 mm inner diameter. Since the maximal tubing length from the CSTR chamber to the analysis instruments was 3 m the impact on the overall residence time is negligible.

We investigated the change in CCN-activity of soot particles rich in organic carbon due to heterogeneous ozone oxidation. The soot particles were generated with the miniature Combustion Aerosol STandard (miniCAST, Model 4200, Jing Ltd., Zollikofen, Switzerland) which is propelled with propane and operates with a laminar diffusion flame. The miniCAST was operated under fuel-rich conditions (set point 6 according to the manual) in order to generate a soot which was rich in organic compounds (fuel-to-air ration: 1.03).

The aerosol chamber used is a "real CSTR" which and is very close to an "ideal CSTR". We added a section where we explicitly point out that in terms of changes in the particle number concentration no deviation between our "real CSTR" and an "ideal CSTR" could be detected. (new P16 L10-20)

In the flushing regime the particle number concentration declines exponentially in both experiments. Eq. (5) describes the ideal/theoretical evolution of the particle number concentration in the flushing regime when taking the hydrodynamic residence time $\tau_{CSTR}$ according to eq. 1 into account. In the ideal case the decay is solely caused by the flushing process. In reality, the decay is a combination of flushing as well as additional particle losses e.g. wall losses or coagulation. Therefore, the real residence time can be obtained by fitting equation 5 to the experimental data after rearrangement for $\tau$, to which we refer to as $\tau_{flush}$ from now on (Kulkarni et al., 2011). In both experiments $\tau_{flush}$ coincides at 104 min, which is lower than the hydrodynamic residence time $\tau_{CSTR}$ of 111 min. In other words, the particle concentration declines faster than expected. This difference is caused by particle losses to the chamber wall, which acts as an additional particle sink parallel to flushing and reduces the particle lifetime. Nevertheless, statistical analysis of the experimental data results in purely statistical noise centered on the fitting curve used to determine $\tau_{flush}$. This indicates that in terms of mixing no difference between an ideal CSTR and the aerosol chamber used here can be detected with the applied instrumentation.

We conducted a broad range of experiments we were investigated the impact of different aging conditions onto different soot particles. We investigate parameters like single particle mass, hygroscopicity, chemical composition, Ice nucleation activity as well. Results from these experiments will be published shortly. We refrain from mentioning these measurements in order to not distract the reader, to keep the focus on the CSTR-approach and on the applicability of this approach.

Corresponding details are suggested to be provided especially for those who are unfamiliar with such systems. From my perspective, the organization of this section could be improved for better delivery of the key points.

We expanded and rewrote the introduction section and made a clearer distinction between the application of the CSTR approach for atmospheric experiments,   the development of a CSTR-specific mathematical framework, the newly developed $t_{act}$-concept, as well as the

application of the $t_{act}$-concept to other continuous flow steady state chambers, namely OFRs. (new: P2 L40- P4 L7)

> In the following, we discuss a theoretical basis for the analysis of time-dependent changes in binary systems within well-mixed continuous flow aerosol aging chambers (CSTR-approach). We developed a mathematical framework which allows the retrieval of characteristic parameters from the system of interest (e.g. CCN activity) and which allows for the calculation of the parameter of interest throughout the entire duration. Key element in this framework is the activation time ($t_{act}$) which marks the time after which the individual aerosol particle undergoes a transition within a binary system. We start by introducing an idealized system in which $t_{act}$ can be described by a single number and proceed to a more realistic setting in which we incorporate a distribution of particles with different individual $t_{act}$'s (activation time distribution, P($t_{act}$)). Further, we test the $t_{act}$-concept on real experimental data and finally apply it to other types of continuous flow aging chambers such as OFRs. We show that application of the $t_{act}$-concept is capable of giving new insights to ORF data and further significantly improves the understanding of discrepancies in experimental results obtained in intercomparison studies Lambe et al., (2011) with different reactors such as the Potential Aerosol Mass Chamber (PAM) chamber and the Toronto Photo-Oxidation Tube (TPOT).

Furthermore, we expanded the section where we introduce the batch-mode aerosol chambers, the OFRs and compare both types with the CSTR. (new P3 L17-39)

> In an aerosol chamber operated in batch mode, the reaction volume is first filled with the sample aerosol as fast as possible to achieve high homogeneity of the sample. After the desired start concentration is reached further addition of the sample aerosol is stopped and the aging is initiated e.g. by addition of the oxidant. This point in time is generally defined as the start of the experiment and referred to as $t = 0$. Data acquisition of the ageing sample takes place while the reaction volume is flushed with sample-free gas. The composition throughout the chamber is homogeneous but evolving in time, therefore no steady state conditions are ever achieved. This concept is used to operate many large scale environmental chambers (Cocker et al., 2001; Leskinen et al., 2015; Nordin et al., 2013; Paulsen et al., 2005; Platt et al., 2013; Presto et al., 2005; Rohrer et al., 2005).

> A PFR is a steady state reactor in which no mixing along the flow path (axial mixing) but perfect mixing perpendicular to the flow (radial mixing) takes place. Further, a continuous feed-in of reactants and withdrawal of sample take place at equal flow rates simultaneously. This results in a constant composition of the output solely depending on the residence time within the reactor. This ideal system is approximated by many Oxidation Flow Reactors (OFR) e.g. PAM chamber (George et al., 2007), TPOT Chamber (Kang et al., 2007), Micro Smog Chamber (MSC; Keller and Burtscher, 2012), or the TUT Secondary Aerosol Reactor (TSAR ; Simonen et al., 2017). The main difference between an ideal PFR and real OFRs is that in OFRs significant but unintentional mixing of the aerosol along the flow path takes place (Mitroo et al., 2018). Therefore, OFRs show a significant residence distribution.

The CSTR is a steady state reactor with a constant reactant feed in and sample withdrawal as well but opposite to OFRs, the volume is actively stirred to achieve a homogeneous composition throughout the reactor volume. Due to the active mixing, sample stream composition and conditions are the same as within the entire chamber volume. The concept of the CSTR requires perfect internal mixing, which cannot be achieved in real systems. However, due to the good miscibility and low viscosity of gases and the aerosol particles being homogenously dispersed, it is possible to achieve a degree of mixing which is very close to a perfectly mixed system. Especially in the case of mimicking atmospheric processes, residence times of several hours are achieved. Compared to that, the time needed for dissipating all gradients, which is in the order of seconds to minutes, can be considered small.

2. The Section 6, especially the last paragraph of which, is quite confusing. It is good to see the application of the activation time concept (*tact*) into data interpretation of previous chamber studies, with improved agreements among different datasets. Nevertheless, there are several concerns need to be addressed. First of all, the previously used chambers such as PAM, they are actually not CSTR or far from the ideal mixing condition during oxidation. As a result, how can you simply apply the *tact* or RTD concept for CSTR system into the data interpretation of OFR/PAM reactors? Necessary information is needed to clarify this point.

The PAM and TPOT chambers are indeed no CSTRs, are not internally well-mixed and therefore cannot be described with the here introduced CSTR-specific mathematical framework. However, they show a significant residence time distribution due the mixing along the flow path, therefore it is possible to apply the $t_{act}$ concept. An RTD means that the aerosol particles that leave the OFR stayed inside the chamber for different individual times. If the measured AF behind the OFR is 0.3, we raise the question which particles of the whole aerosol particle distribution are the CCN-active particles. The $t_{act}$ concept implies that only the oldest 30% of the particles are CCN-active and the youngest 70% are CCN-inactive. The time that separates the youngest and CCN-inactive 70% from oldest and CCN-active 30% is the $t_{act}$ (necessary aging time).

In principle it is also possible that young and old particle activate equally well, however this seems to be unlikely for the here discussed BES-particles. Furthermore even this behavior could be captured by the $t_{act}$-distribution. In this unlikely case the activation time distribution would be a horizontal line ( P($t_{act}$) = 0.3 ) and neither a peak nor a Gaussian shaped distribution.

In this manuscript we define 2 scenarios that combine this $t_{act}$-concept with the RTD reported by Lambe et al. (Figure 7) In the first scenario (High-OH) we calculated the fraction of particles older than 40s in the PAM and TPOT chamber. In the second scenario we did the same with a time of 180s. As can be seen the fraction of particles older than this threshold time $t_{act}$ varies between both chambers. Since only the oldest particle can be CCN-active this leads to different measured AF-values. IN The High-OH scenario, the AF in the TPOT is higher than in the PAM chamber. In the Low-OH scenario The AF in the TPOT is lower than in the PAM chamber. The same trend was reported by Lambe et al.

However this a qualitative application of the $t_{act}$-concept. We added following section that mentions what would be needed for quantitative application of $t_{act}$ – concept.  (new: P20 L23-36)

> Up to now, the discussion did not include many important processes that are relevant in aging chambers e.g. particle wall-interaction, gas-phase-partitioning, fluctuating input concentrations while field measurements, or inhomogeneities inside the OFR. These aspects are important for many processes such as the formation of SOA and can be incorporated to the $t_{act}$-concept by modifying eq. (13). As the actual calculation requires a multidimensional data array and detailed knowledge about the chamber of interest, this subject matter is beyond the scope of this publication and will not be discussed further. Nevertheless, the overall conclusion is that application of the original/non-adjusted $t_{act}$-concept can explain why measurements within different OFR chambers agree in parameters, which dependent on the bulk properties of the aerosol particle population (e.g. average O:C ratio) and at the same time disagree in parameters, which are dependent on the condition/status of the individual particle (e.g. CCN-activity). Therefore, we suggest to apply the concept of the activation time $t_{act}$ or the activation time distribution $P(t_{act})$ as metric in addition to calculating average values, such as the global $AF$ and OH-exposure if following conditions are met. One, the system or parameter of interest can be described as a binary system and undergoes step-wise / non-gradual transitions such as CCN-activity. Two, the OFR used has a RTD broad enough to influence the outcome. Three, the conditions inside the reactor are either homogeneous or a correction for inhomogeneities (e.g. different oxidants concentrations inside the reactor) is implemented.

Another issue is that discrepancies in CCN activity of SOA formed from chamber oxidation experiments could be influenced by various factors, such as gas-particle partitioning and particle-phase reactions during SOA production as well as liquidliquid phase separation during activation processes. Additionally, the variability in different operation parameters such as relative humidity, initial concentration of VOC precursors, and acidity in the OFR/PAM chamber can affect the SOA formation process even for a same average OH concentration condition, further influencing the subsequent CCN activation process. In this sense, how to evaluate or exclude the impacts of these factors on the agreement of CCN activity (or AF) measurements for different types of OFR or PAM experiments? Namely, how can we confirm that the discrepancies are predominantly introduced by the activation time (or RTD) rather than by the other influencing parameters, although the application of tact can better capture the deviation of CCN activity (likely due to change in chemical compositions) than what the bulk H/C and O/C ratios do? Further discussion is needed to clarify the abovementioned points.

We only refer to the aging of BES-particle in the PAM and TPOT-chamber. Therefore, we do not discuss the application of the $t_{act}$-concept onto the formation and aging of SOA. We also only apply the $t_{act}$-concept qualitatively to the results obtained by (Lambe et. al 2011). To identify if the RTD of the two chambers is the only reason for a different measured AF a quantitative analysis would be needed. However, this greatly exceeds the scope this manuscript. Nevertheless, the $t_{act}$-concept predicts the overall trend in the AF measured downstream both chambers well. This is described in the section above.

The formation and subsequent aging of SOA in OFRs is a complex process and cannot by fully described by the here introduced $t_{act}$-concept. The main factor that inhibits a straightforward application is that SOA-particles only form in the chamber. As long as the educts are gaseous and therefore fully miscible, no air parcel can be separated from another air parcel by a measurement of the CCN-activity. After particles formed, each particle can have its individual trajectory inside the OFR. Therefore, particles can have different individual aging times and individual degrees of chemical modification. This can be detected by e.g. a CCN-Counter. For gases in an OFR that's not possible. Furthermore, the $t_{act}$-concept relies on the fact that a single particle is a closed system. If one particle has a high concentration of substance A while another particle has a low concentration of substance A, no exchange processes can dissipate this gradient. In the case of SOA, this can be an invalid assumption. Volatile compounds can evaporate and can condense on other particles and dissipate concentration gradients. An extended framework can potentially capture these processes and the $t_{act}$-concept has to be part of this.

Other aspects that should be implemented in such extended framework as well, would be the internal inhomogeneities of the OFR. For example, the concentration of OH-radicals inside the OFR is inhomogenously distributed. The OH-concentration for example increases the closer the light sources get. Knowledge of the temperature gradient/profile would also be relevant since the speed of chemical reactions is typically temperature dependent. At this point, we want to mention that every additional variable increases the complexity of the math exponentially. This degree of complexity was a major reason why we favored an internally mixed aerosol chamber. This allowed us to assume homogeneous conditions throughout the entire experiment.

**Specific comments:**
1. **Abstract**: What does the "non-gradual transitions" refer to here (Line 12)?

We greatly expanded the explanation of the phrase "non-gradual transition" throughout the entire manuscript (see major comment). In the abstract we changed it from: (old: P1 L12)

> We show that this concept can be applied to other systems investigating non-gradual transitions.

To: (new: P1 L14-15)

This experimental approach and data analysis concept can be applied for the investigation of any transition in aerosol particles properties that can be considered as a binary system

In the last sentence, what specific kinds of "discrepancies" are you suggesting? It is better to clarify these concepts precisely, as which are important points to show the significance and applicability of this study.

We show for the specific example of CCN-activation of photochemically aged BES-particles, that the activation time concept is beneficial for the data-analysis from OFRs. However it is not limited to this, since it is a general concept. These points are now intensively discuss in the expanded manuscript. We refrain from discussing specific discrepancies in the abstract since this would greatly expand the abstract.

Changed from: (old: P1 L13-15)

Furthermore we show how $t_{act}$ can be applied for the analysis of data originating from other oxidation flow reactors widely used in atmospheric sciences. This concept allows to explain discrepancies found in intercomparison of different chambers.

To: (new: P1 L15-19)

Furthermore, we show how $t_{act}$ can be applied for the analysis of data originating from other reactor types such as Oxidation Flow Reactors (OFR), which are widely used in atmospheric sciences. The new $t_{act}$ concept significantly supports the understanding of data acquired in OFRs especially these of deviating experimental results in intercomparison campaigns.

2. Page 2, line 8: How is the "perfectly mixed" defined here? It is unclear especially to readers those are unfamiliar with the CSTR technique.

To avoid the phrase "perfectly mixed" we mention instead that a homogenous aerosol mixture can be achieved by actively stirring the aerosol inside the chamber. This is a necessary requirement for a CSTR operation. At this point "perfectly mixed" is accurate since we refer to an ideal system. However, this can be confusing as the reviewer pointed out.

changed from: (old: P2 L7-8)

The aerosol inside the CSTR is perfectly mixed, therefore a mix of aged and unaged aerosols is continuously extracted from the CSTR for analysis.

To: (new: P2 L14-15)

The volume of the CSTR is actively stirred in order to achieve a homogenous aerosol mixture. Due to the mixing, the aerosol that is continuously extracted for analysis consists of a well-defined mixture of aerosols at different aging stages.

Following which, what do you mean that "real processes in the atmosphere where aerosols are constantly emitted, mixed and removed"? Are you sure of the "constantly" condition in the

ambient environment? Which specific atmospheric processes have you included in this statement, any references can be provided to support the idea?

As the reviewer pointed out earlier many readers are not familiar with CSTRs. This reference to atmospheric processes shall give the uniformed reader a better understanding what it means to operate an aerosol chamber in CSTR-mode. To the authors knowledge, all other experimental approaches that use flow tube or batch-chamber aim to generate a uniformly aged aerosol. This stands in a strong contrast to the here presented approach, where we are aiming for a non-uniformly aged aerosol output.

This loosly resembles the conditions in the atmosphere. Due to the persistent emission of aerosol into the atmosphere, the ongoing modification and the ongoing removal of aerosols from the atmosphere a mixture of young, medium-aged and old aerosols are present in the atmosphere. Similar to that a freshly produced aerosol is fed-in/emitted to the CSTR. Inside the CSTR, the aerosol is constantly chemically modified and mixed with the fresh aerosol. The aerosol that is removed from the chamber is a mixture of young, medium-aged and old aerosols. We acknowledge that the atmosphere is not a CSTR, however both are comparable in the mixing aspect.

We clarified that we compare the CSTR-approach and the atmosphere in terms of mixing aerosols and measuring a non-uniformly aged aerosol.

Changed from: (old: P2 L8-9)

> This approach is close to real processes in the atmosphere where aerosols are constantly emitted, mixed and removed as well.

To: (new: P2 L16-19)

> From this perspective, the CSTR approach is closer to atmospheric processes than other reactor types as in the real atmosphere except for individual plume emissions aerosols are rather continuously emitted, mixed, and removed. This results in a mixture of aerosols at different aging stages, but of course, the atmospheric mixture is less well defined compared to an aerosol in a CSTR.

3. **Equation 5**: Why is the exponential part not expressed as "$e^{\left(\frac{t-t_{switch}}{\tau_{CSTR}}\right)}$" for the flushing regime? Please check the conversion carefully.

We thank the reviewer for taking the time and checking the equations as well. That was a mistake on our side and we implemented a correction.

Equation changed from

$$[A_{CSTR}(t)]=[A(t=t_{switch})]\cdot e^{\left(\frac{-t}{\tau_{CSTR}}\right)} \tag{4}$$

To

$$[A_{CSTR}(t)]=[A(t=t_{switch})]\cdot e^{\left(\frac{t-t_{switch}}{\tau_{CSTR}}\right)} \tag{5}$$

4. Page 6, line 20: As a crucial parameter introduced in this study, the activation time (tact) for non-gradual transitions was developed. However, what do you mean "If all the other parameters stay constant" during non-gradual transitions, which specific parameters are you referring to?

"If all other parameters stay constant…" refers to the previously mentioned external parameters that can trigger non-gradual/step-wise changes in a particle. We added examples for relevant external parameters. We created this $t_{act}$-concept based on this assumption since we operated our experiment in a temperature controlled chamber, at a defined RH and at a ozone concentration that was actively kept constant.

Changed from: (old: P6 L20-21)

> If the all other parameters stay constant, while a particles undergoes changes that result in a non-gradual transitions, this transition can be described as a function of time.

To: (new: P6 L26-30)

> We may assume a system in which all external parameters stay constant but the particle itself undergoes a continuous transformation, e.g. due to oxidation. After a certain period of time, this continuous transformation, in this specific case oxidation, can lead to a change in a binary property, e.g. CCN-activity. Ultimately, the step-wise or non-gradual transition is a function of time. We define the required time span (e.g. necessary aging time) that leads to a change in a specific particle property, resulting in a transition in a binary system in another particle property as the activation time ($t_{act}$).

Is it easy to achieve in practical conditions of laboratory chamber experiments?

This section introduces a theoretical and idealized concept for which constant background concentrations are assumed. Experimental short-comings are therefore not discussed in this section.

Besides that, it is rather easy to keep certain parameters like temperature, relative humidty and Ozone concentration constant in our CSTR-experiments. The fan inside the chamber creates a homogenous atmosphere. The temperature can be actively controlled with a heater/chiller. The ozone concentration was constantly measured and kept stable with a feedback loop that regulated the ozone source. This is an experimental advantage towards many OFRs. Results from these experiments will be published within the next months.

5. Equation 7: I think it should be "$e^{\frac{-t_{switch}}{\tau_{CSTR}}}$" for the flushing regime? Please check the conversion carefully

The initial version of equation 7 is correct. $t_{switch}$ cannot have any influence in the filling regime since its not reached yet.

6. **Equation 8**: Why is the simplified equation not expressed as '***tact*** = -ln(AF(t))·τCSTR'? I'm wondering how will the value of AF(t→∞) be, could it be 0 as

suggested by the exponentially decreased curve in Fig.2, or probably approaching 1 like what AF responds when switching to the flushing regime as shown in Fig.3? How should the readers understand the corresponding physical meaning of AF(t→∞) in this steady state condition? Corresponding details are necessary.

The equation 6a. 6b and 7 describes how the AF inside the CSTR evolves over time while filling the CSTR.

$$t \leq t_{act} : \quad AF(t) = 0 \tag{6a}$$

$$t > t_{act} : \quad AF(t) = \frac{\text{activated particles}}{\text{all particles}} = \frac{\int_{t=t_{act}}^{t} RTD(t)\, dt}{\int_{0}^{t} RTD(t)\, dt} = \frac{RTD_{sum}(t) - RTD_{sum}(t=t_{act})}{RTD_{sum}(t)} \tag{6b}$$

$$t_{act} = \ln\left(1 - \left((1-AF(t)) \cdot \left(1 - e^{\frac{-t}{\tau_{CSTR}}}\right)\right)\right) \cdot (-\tau_{CSTR}) \tag{7}$$

During this time the particle concentration inside the CSTR is increasing continuously. Parallel to that the *AF* is also changing. After a certain time the changes in the particle concentration become negligible. The CSTR is then in a dynamic equilibrium called "steady state". In this dynamic equilibrium, aerosol particles still become CCN-active due to aging but already CCN-active particle are also constantly removed from the CSTR by flushing. This state can be maintained for an in theory infinite time span. This means that the experimental duration t grows continuously, but the conditions inside the CSTR are constant. Therefore, the distribution of young, medium-aged and old particle inside the CSTR is constant and decoupled from the experimental duration t. This also means that the AF and the activation time $t_{act}$ calculated from it do not change anymore. The same applies to OFRs which are operated as steady state reactor as well.

In the old version of the manuscript the x-axis in Fig. 2 was labeled with "particle age / min" which was now changed to "residence time / min". The individual residence time is equal to the particle age. However, this is only mentioned at a later point in the manuscript, which is indeed misleading. The curve in Fig. 2 does therefore not show how particle number concentration that declines with increasing experimental duration. Instead it shows the abundance of particles with a defined residence time (=individual particle age) in the total aerosol particle mixture.

Figure 2 shows the RTD during steady state. The area under the curve represents the total particle population. The colored area is the fraction of particles that is older than the defined $t_{act}$. This "oldest fraction" is equal the CCN-activated fraction in our model. Since this curve refers to the steady state it does not change over time, but stays constant. Therefore the AF does neither decline exponentially nor does it approach 1, but stays constant with increasing experimental duration. Also, the correlation between the individual necessary aging time $t_{act}$ with the measured AF is not linear but exponential as can be deduced from the different AF-values in Fig.2.

As a result of this, the mentioned equation 8 cannot contain any dependency on the experimental duration t. In fact, equation 8 is derived from eq 7 for (t→∞). This is now stated in the manuscript as: (new: P6 L20 –P7 L1)

> After the conditions in the aerosol chamber reached steady state, the measured $AF$ does not change anymore. This is due to the fundamental concept of a CSTR which entails a continuous addition of fresh particles and simultaneous withdrawal of sample at equal flow rates resulting in a dynamic equilibrium and a constant RTD.
>
> To simplify matters, the reason for the constant $AF$ within this dynamic equilibrium can be visualized when focusing on three distinct time periods within the continuum of the RTD and thereby on three specific particle fractions. Fraction one is within the right tail of the RTD and consists of particles with a residence time that is above $t_{act}$. They are only a few compared to the total number of particles and a fraction of these is constantly flushed out with the sample stream. This would lead to a hypothetical reduction of $AF$ if not simultaneously the second particle fraction of interest was in the situation to have an individual residence time that is just about to exceed $t_{act}$. The particles within fraction two are thereby transitioning from the CCN inactive particle fraction within the aerosol chamber to the CCN active particle fraction. The hypothetical loss of CCN inactive particles would lead to an increase in $AF$ if not again simultaneously the third particle fraction of interest consisting of fresh and CCN inactive particles was about to be added to the chamber volume.
>
> Due to this dynamic equilibrium, eq. (7) can be simplified to eq. (8) assuming that the experimental duration $t$ approaches infinity $\left( \lim_{t \to \infty} eq. \ (7) = eq. \ (8) \right)$

$$t_{act} = - \ln(AF) \cdot \tau_{CSTR} \tag{8}$$

A comparision to the flushing regime in Fig 3 is not useful since this a different operational setting, that is described with its own equations. In the flushing regime the particle concentration and the AF inside the CSTR are changing constantly which is similar to aerosol chamber operated in batch-mode. The equations to describe the changing AF while flushing are introduced in the section "Particle activation during flushing regime"

7. Page 8, line 7: It sounds a bit strange of "global" AF? Is the "global" trying to represent the specific exponentially increased AF inside CSTR or just to show a different AF case with other non-CSTR chamber experiments?

A consequence of the CSTR-approach is that aerosol particles with different individual aging times are present in the chamber at the same time. Each of these aerosol fractions is CCN-active to a different degree. The term "global AF" refers to the combination of all AF-values from different individual aerosol particle fractions combined (=global). This global $AF$ is the $AF$ that is measured downstream the chamber

The explanation of the global AF in the manuscript was changed from: (old: P8 L7)

> This leads to an exponential increase of the $AF$ inside the CSTR (global $AF$) until $AF$=1.

To: (new: P11 L21-22)

> This single value, from now on referred to as global *AF*, represents the average *AF* over all *AF*s of the individual sub-fractions within the population as will be explained in more detail in the upcoming sections.

Further the introduction of the term "global AF" was moved to alater point in the manuscript.

Line 10: ***"… and therefore the global AF only if tact = tswitch."*** Some information was missed in this sentence.

The here present concept assumes that the *AF* inside the CSTR is equal to the fraction of particles older than the threshold time $t_{act}$. Therefore, the *AF* can be calculated by calculating the fraction of particles older than this time. However, this comes with some difficulties in the flushing regime.

In Figure 1 can be seen how the RTD changes after the particle feed-in is stopped. The whole RTD-curve from steady state is shifted towards longer times. As a result of this, the fraction of particles older than a threshold time increases exponentially. This increase of "old particle fraction" change is captured in equation 9.

$$AF(t)_{flushing} = \int_{t_{switch}/\tau_{CSTR}}^{t/\tau_{CSTR}} e^{\left(\frac{t-2 \cdot t_{switch}}{\tau_{CSTR}}\right)} \, d\left(\frac{t}{\tau_{CSTR}}\right) \tag{9}$$

This equation 9 is the result of centering of the RTD during steady state at t = $t_{switch}$. Therefore, equation 9 describes which fraction of particles is older than $t_{switch}$. Since the fraction of particles older a defined threshold time is equal the AF, this equation would describe the changing AF for the special case that $t_{switch}$ and $t_{act}$ are equal. Implementing all other case requires a shift of the starting point of equation 9. This is done by introducing the parameter $t_{offset}$.

We rewrote the entire section, in order to increase the understandability (new: P9 L7 – P10 L14)

[revised manuscript text omitted]

**8. Title of Sect.4**: What does the "first experiments" mean? Try to update the message into a more informative one.

The caption "Application in first experiments" was removed. The following captions were rephrased.

**9. Page 12, line 10**: What does the "uniform" mean: "activate uniformly" here and "a uniform aerosol population" in the caption of Fig.5? Are you trying to say the initial particles with the same particle size and chemical composition? If so, how to understand the Gaussian distribution scenario (i.e., *"This is because there are some particles in the population, that activate earlier than the mean activation time."*), as all the uniform particles are supposed to activate at a same activation time? More straightforward/concise descriptions would be useful to explain the scenario clearly.

The reviewer understood the meaning of "activate uniformly" correctly. However, we do not consider this scenario as the most like scenario. It is more likely that some particles activate after a shorter/longer aging time than other particles. A reason for this can be that even in a size-selected aerosol particle flow some particles are slightly small/larger than the average diameter. Since the CCN-activity of particle shows a size-dependency this can lead to a non-uniform activation of the aerosol particles.

In the in previous sections "Introducing the activation time distribution $P(t_{act})$" and "Impact of the activation time distribution on the individual $AF$" we created a scenario where particles do not activate uniformly, but show a distribution of different $t_{act}$'s. We use a Gaussian distribution to capture the fraction of particles that activate after a certain necessary aging time $t_{act}$.

In section "Calculation of the total activated fraction (global $AF$)" we pick up the previously introduced $t_{act}$ and $P(t_{act})$-concept and calculate the changing global $AF$ throughout a full CSTR-experiment. This is done by integration of the contribution of individual aerosol fractions to the global AF over the whole range of possible $t_{act}$'s (equation 13)

$$AF(t) = \int_{t_{act}=0}^{t_{act}=t} AF(t_{act},t)\cdot P(t_{act})\; dt_{act} \qquad\qquad (13)$$

**10. Page 15, line 10**: How was the particle wall loss rate of $k$ = 0.000625 min-1 estimated? Where can the readers find the corresponding clues/data for calculation?

The paper the wall loss rate was obtained from the difference between theoretical and expected particle loss during flushing under the assumption of a first order loss kinetic.

We extended the explanation of how we calculated the particle wall loss numbers and refer to publications were the same calculation regarding particle losses was applied.

The section was changed from: (old: P15 L8-11)

> The measured particle flush rate $\tau_{flush}$ obtained during the flushing regime is 104 min in both experiments, which differs slightly from the theoretical flush rate $\tau_{CSTR}$ = 111 min. This difference is caused by particle losses to the chamber wall. From this difference the particle wall loss rate of $k$ = 0.000625 min-1 and a mean particle life time upon wall loss of 1600 min was determined assuming first order loss rates

To: (new: P16 L10 –L24)

> In the flushing regime the particle number concentration declines exponentially in both experiments. Eq. (5) describes the ideal/theoretical evolution of the particle number concentration in the flushing regime when taking the hydrodynamic residence time $\tau_{CSTR}$ according to eq. 1 into account. In the ideal case the decay is solely caused by the flushing process. In reality, the decay is a combination of flushing as well as additional particle losses e.g. wall losses or coagulation. Therefore, the real residence time can be obtained by fitting equation 5 to the experimental data after rearrangement for $\tau$, to which we refer to as $\tau_{flush}$ from now on (Kulkarni et al., 2011). In both experiments $\tau_{flush}$ coincides at 104 min, which is lower than the hydrodynamic residence time $\tau_{CSTR}$ of 111 min. In other words, the particle concentration declines faster than expected. This difference is caused by particle losses to the chamber wall, which acts as an additional particle sink parallel to flushing and reduces the particle lifetime. Nevertheless, statistical analysis of the experimental data results in purely statistical noise centered on the fitting curve used to determine $\tau_{flush}$. This indicates that in terms of mixing no difference between an ideal CSTR and the aerosol chamber used here can be detected with the applied instrumentation.

> When dividing the real particle life time ($\tau_{flush}$) into its individual components, a particle life time upon wall losses ($\tau_{wall-loss}$) of 1600 min can be determined in accordance with first order wall loss kinetic (Crump et al., 1982; Wang et al., 2018). The influence of particle coagulation can be considered negligible due to the low coagulation rate of 100 nm particle at concentrations of maximum 1500 cm$^{-3}$ (Kulkarni et al., 2011).

11. Page 15, line 15: What is the meaning of the last sentence? What does the "otherSS" refer to? Where can readers find the corresponding details? Necessary information is needed.

This refers to scans through different SS. We only discuss the applicability of the CSTR approach in this paper, therefore we refrain from discussing the impact of ozone-aging onto CCN-activity at different SS. This will be done in a separate publication.

Sentence changed from: (old: P16 L14-15)

> The gaps in the curves during steady state are due performing measurements at other SS.

To: (new: P16 L27-28)

The gaps in the curves during steady state are due to changes in the operation of the CCNC form running on a constant SS (1.0% and 1.4%, respectively) to scanning over a range of SS

12. **Figure** 2: Is the "particle age" of x-axis with the same meaning of the "residence time" in Fig.1? If not, please specify accordingly in the corresponding places.

In this context "particle age" and "residence time" are identical. However until this point in the manuscript, this is not explicitly stated.

Therefore we changed the x-axis description from:

> particle age / min

To:

> residence time / min

13. **Figure** 6: Why is the unit of particle concentration in Fig.6(A1) and (B1) different from those in Figure 3? In Fig.6 (B1), why are the data after 800 min missing? As assumed early in this study that all compounds in CSTR have perfectly mixed thus with constant concentrations during steady state, how to explain the increasing trend in observed particle concentration in the duration of 400-600 min, i.e., AF almost reached a stable level around 0.2 at 1.4% SS conditions)? More detailed discussion should be provided in the corresponding data interpretation sections.

The particle number concentrations are not meant to have different units. It is always $\#/cm^3$. The dot in the unit of the y-axises in Fig.6(A1) and (B1) was a graphical error and was removed.

The deviations from the theoretical changes in the particle concentration are due to a slightly changing particle input concentration. From an experimental point of view it is challenging to keep the particle input concentration constant over a period of 12 h. However, this small change affects the outcome only to a small degree.

The section was changed from: (old: P15 L 6-8)

> The particle concentration curves follow the theoretical filling and flushing curves in a CSTR. The slight decline in the concentration observed in the region where steady state is expected in graph A1 is due to a slight but undesired reduction in the feed-in flow that was experienced during the experiment.

To: (new: P15 L4 – 9)

The graphs A1 and B1 in Fig. 6 show the particle concentration (black crosses; left axis), the measured global *AF* (red crosses) and the fitted global *AF* (blue dashed line, both right axis). The particle number concentration curves (black crosses) follow the theoretical filling and flushing curves as expected in a CSTR (Fig. 3). The slight decline in the concentration in steady state in graph A1 is due to a slight reduction in the particle input concentration that was experienced during the experiment. Visa versa the slight increase in the number concentration in graph B1 is due to a slight increase in the particle input concentration over time.

14. Page 19, line 6: The last sentence is a bit confusing. It is better to clarify the "metric" here, e.g. metric of what specific aspects.

We clarified under which circumstances the activation time concept can be benefical for the data analysis

Changed from: (old: P19 L6)

Depending on the parameter of interest we suggest using $t_{act}$ or $P(t_{act})$ as metric.

To: (new: P20 L31-36)

Therefore, we suggest to apply the concept of the activation time $t_{act}$ or the activation time distribution $P(t_{act})$ as metric in addition to calculating average values, such as the global *AF* and OH-exposure if following conditions are met. One, the system or parameter of interest can be described as a binary system and undergoes step-wise / non-gradual transitions such as CCN-activity. Two, the OFR used has a RTD broad enough to influence the outcome. Three, the conditions inside the reactor are either homogeneous or a correction for inhomogeneities (e.g. different oxidants concentrations inside the reactor) is implemented

**Technical corrections:**
1. **Abstract**, Page 1, line 10: *"… the newly introduced metric: activation time"*

done
2. Page 3, line 27: *"… can be calculated as a function of …"*. A similar issue exists in Line 16, Page 6.

done
3. Page 6, line 13: *"… to describe continues continuous changes"*?

done
4. Page 6, line 19: *"… can be considered as a non-gradual change."*

done
5. Page 6, line 20: *"If the all the other parameters stay constant, while a particles undergoes changes that result in a non-gradual transitions…"*

done
6. **Equation 9**: Why do you use different multiplication signs in these equations, e.g., "*" and "·"? It makes more sense to keep consistent within the same manuscript.

done

7. **Table 1**: Why is the layout of this table so different from other two tables in this manuscript? The corresponding details could be better organized.

We harmonized the table layouts

8. Title of **Sect.4.3**: *"Calculation of the total activated fraction"*

done

9. Page 12, line 12: *"While the uniform scenario shows no activity be for reaching $t_{act}$ …"* Do you mean 'before'?

changed to "before"

10. Page 14, line 8-9: *"As there is a significant share of particles activating significantly earlier than the nominal activation time ($\mu$ = 180 ) in the case of a Gaussian distribution a fraction of 1 % of the entire particle population within the CSTR is already activated after 87 min."* A comma is needed to clarify the point.

Changed from:

> As there is a significant share of particles activating significantly earlier than the nominal activation time ($\mu$ = 180 ) in the case of a Gaussian distribution a fraction of 1 % of the entire particle population within the CSTR is already activated after 87 min

To: (P15 L5-7)

> In the case of $t_{act}$-onset, there is a significant share of particles activating significantly earlier than the nominal activation time ($\mu$ = 180) in the case of a Gaussian distribution. Therefore, a fraction of 0.01 of CCN active particles within the entire particle population is already present after 87 min

11. Page 14, line 12: *"The difference in $t_{act}$ of 10 min between the two $P(t_{act})$-approaches is due to the application …"*
It is very common to see that $t_{act}$ was written as $t_{act}$. Similar issues also exist in some other expressions, e.g., $P_{step}$, which should be $P_{step}$. Please check through the manuscript carefully and make necessary updates accordingly.
In the same paragraph, there are many long sentences without proper splits or connections, which might make the readers difficult or even confused to catch the meaning effectively. For instance:

We harmonized the subscripts

12. Page 14, line 15-16: *"As can be seen in Graph C of Fig. 4, 50 % of the particles with a residence time equal to the nominal activation time are activated in the case of a Gaussian distribution corresponding to $t_{act}$0.5."*

Comma added.

13. Page 14, line 23: *"… were diluted with particle-free and VOC-filtered air…"*

Changed from:

    particle free

To:

    particle-free

14. Page 14, line 25: *"The aerosol flow was fed into the aerosol chamber, where a constant  ozone concentration of 200 ppb was …"*

done

15. Page 14, line 27: *"The size distribution data was acquired by a … (SMPS) system from which the  total particle concentration could be derived."*

done

16. Page 15, line 5: The *"(blue solid line)"* is not needed, since there is only one curve in the corresponding subplots.

We thank the reviewer for mentioning this point, but we keep the "(blue solid line)" to avoid any confusion with the curves in the graphs Fig.6 A1 and B1.

17. Page 15, line 18: *"… µ as well as σ  is larger for P(tact) at a 1.0 % SS of compared to the results obtained for 1.4 % SS. The mean activation time being larger for 1.0 % SS indicates that the longer the chemical aging proceeds, the initially inactive soot particles activate  at a lower SS."*

done

18. Page 15, Line 23: The comma between "*P(tact)*" and "requires" is unnecessary.

comma removed

19. Page 17, line 1 and 3: *"Within these types of chambers …"*

done

20. Page 17, line 10: *"secondary organic aerosol (SOA)"*, and the "VOCs" should be defined before when it appeared for the first time.

done

21. Page 17, line 24: *"…to be directly proportional to the AFs…"*

done

22. Page 18, line 3: *"…we  present two scenarios."*

changed

23. Page 18, line 9: *"…other parameters can agree very well."*

done

24. Page 19, line 22: *"…soot particles transitioning  from initial CCN-inactivity to CCN-activity over the course of …*

done

---

## Author Comment (AC3) · 25 Feb 2019

**We thank the reviewer for the comprehensive feedback on our work. With the help of the reviewers' comments we greatly improved the understandability of our work and made it more accessible to a broader audience. Detailed answers to the individual comments are given below. For clarity, the reviewers' comments are written in black, and our response in red. Texts from the old version of the manuscript are typed in green and texts from the revised manuscript in blue.**
* * *
**General comments:**

This manuscript presented an improved experimental approach to perform atmospheric oxidation of soot particles using a Continuous Flow Stirred Tank Reactor (CSTR), which enables extended sampling time within a small-size conventional aerosol chamber. A new metric of activation time ($tact$) was developed to characterize the change of activated fraction (AF) in different regimes (i.e., filling, steady state, and flushing) for soot particles following heterogeneous ozone oxidation. Good agreements between theoretical calculations and parameterized CCN activities using $tact$ were achieved for their experimental data. The $tact$ concept was also applied into some previous studies with continuous flow chambers. Discrepancies in the CCN activity of BES particles can be better explained with considering $tact$ and residence time distribution, in comparison to those initially interpreted by the bulk H/C and O/C ratios, which couldn't fully characterize the detailed change in particle chemical compositions. This work is worth further application in atmospheric sciences, yet some details and interpretations could be clarified, reorganized, and improved accordingly. I would recommend for the final publication in AMT upon major revisions, as detailed below.

**Major comments:**

1. In the motivation section (Page 2, Line 19): The "non-gradual transition" case of CCN activation suddenly appeared, with no prior introduction or definition of this new concept (instead, which was included in Sect.3). This content seemed to be disconnected with the information detailed in the last sentence, and I didn't catch the importance/necessity of developing a mathematical analysis for the non-gradual transitions in the following statements.

We understand that the wording we used does not precisely describe to what kind of changes we refer to. Our approach to improve the understandability is to add synonyms commonly used in the atmospheric science community. Additionally we added examples to illustrate this concepts. For the examples we chose common processes investigated by the atmospheric science community. Nevertheless, these concepts are not limited to atmospheric science and can be applied in different fields as well.

In case of the term "non-gradual" we refer to changes like phase-transitions where a property changes step-wise. This is the opposite of a gradual or continuous change of a property. An example would be the freezing of water. Below or above 0°C the density of liquid water/ice changes gradually with the temperature. At 0°C the density does not change

gradually but changes step-wise by jumping between 0.92 g/cm$^3$ and 1.00 g/cm$^3$ . To clarify what is meant by "non-gradual" we extended the introduction of this phrase and added "step-wise change" and "transition between binary states" as alternative explanations. "Transition between binary states" hereby means that a system/particle can be described by two distinct states. Either a droplet is liquid or frozen. A transition from one state to another can be described as "non-gradual" as well.

We added a list of possible transition that can be described as "non-gradual", "step-wise change" and "transition between binary states" (P2 L26-39)

Such transitions in binary systems are step-wise, also referred to as non-gradual changes in a particle property, such as:

1)      Freezing of a water droplet: Step-wise and therefore non-gradual change in the particle density; the water is either in liquid or solid state.

2)      Deliquescence of soluble aerosol particles: The particles show a step-wise i.e. non-gradual increase in diameter.

Binary particle properties are not necessarily intrinsic particle properties, but can also be defined by the measurement protocol.

3)      CCN-activity: The chemical and physical properties of an aerosol particle can vary, but the particle is either CCN-inactive or CCN-active at a defined super saturation (SS).

4)      Growth beyond a threshold: Condensational growth of an aerosol particle leads to a continuous and gradual increase of the particle diameter. A binary system can be defined by introducing a threshold diameter that can be arbitrarily chosen. The aerosol particle is either smaller or larger than this defined threshold diameter. The same holds true when particles are separated e.g. in aerosol impactors.

Therefore, the concept of non-gradual transitions/transitions within binary systems can be used to describe a multitude of changes in particle properties.

Such transitions between binary states are a necessary requirement for the analysis of data from experiment conducted in a CSTR aerosol chamber. Some of the transitions mentioned above are not triggered by a change in the ambient conditions but by a change of the particle itself e.g. due to aging. Aging processes show typically a time-dependence. We focus on the time needed to modify an aerosol particle to such a degree that it changes its state in a binary system. The example which we present in our manuscript is the CCN-activation of soot particles due to oxidation with ozone. Initially, a single soot particle is CCN-inactive at a given super saturation. After a certain aging time it is able to accumulate water at this super saturation. The chemical modification of the soot particle is a continuous process. The change ig CCN-activity is not a continuous process. A particle is either CCN-inactive or CCN-active. The transformation from CCN-inactive to CCN-active is therefore a step-wise change in the particle properties.

The authors have introduced the concept of CSTR and suggested that *"The steady state in the CSTR is characterized by constant concentration of all compounds and constant reaction rates."*. It is a bit confusing that how the assumed "perfectly internal mixing" is achieved, even if without considering the influences of particle wall loss and coagulation during different experimental regimes.

The steady state in CSTR is indeed characterized by constant concentrations. All processes and chemical reactions proceed, but new compounds are constantly introduced to the CSTR as well as old products are continuously removed from the CSTR. After a certain time span the CSTR reaches a dynamic equilibrium and therefore the concentration does not change anymore. The same is valid for oxidation flow reactors like the Potential Aerosol Mass chamber. These kind of aging chamber also operate in steady state mode. In contrast to that, environmental aging chamber operated in batch mode are not continuously filled with new compounds. These type of chambers have constantly changing conditions.

Particle losses to the chamber wall do occur in CSTR-experiment as well, but due to the constant feed-in of fresh aerosol the particle-wall-loss-rate reaches a constant speed once the CSTR is in its dynamic equilibrium/steady state.

The "perfectly mixed" refers to the active mixing inside the CSTR. We used a fan to mix the freshly introduced aerosol with the aerosol that entered the CSTR before. A real CSTR is theoretically never perfectly mixed. However, a perfect internal mixing can be extremely well approximated by actively stirring the aerosol inside the chamber.

Section changed from: (old P2 L6-8)

The continuous flow stirred tank reactor (CSTR) describes an aerosol chamber, which is continuously filled with an aerosol flow constant in composition over time. The aerosol inside the CSTR is perfectly mixed, therefore a mix of aged and unaged aerosols is continuously extracted from the CSTR for analysis.

To: (new P2 L13-19)

The CSTR approach describes an aerosol chamber, which is continuously filled with an aerosol flow constant in composition over time. The volume of the CSTR is actively stirred in order to achieve a homogenous aerosol mixture. Due to the mixing, the aerosol that is continuously extracted for analysis consists of a well-defined mixture of aerosols at different aging stages.

In addition, we show in statistical analysis of the flushing curves in the two experiments presented in the manuscript reveal that the deviation from perfect mixing is below the detection limit of the instrumentation deployed.

P16 L18-20

Nevertheless, statistical analysis of the experimental data results in purely statistical noise centered on the fitting curve used to determine $\tau_{flush}$. This indicates that in terms of mixing no difference between an ideal CSTR and the aerosol chamber used here can be detected with the applied instrumentation.

How should readers understand the "constant concentration of all compounds" during aging reactions in the CSTR, where the corresponding compositions/concentration of reactants/products are supposed to vary with such processes?

As described above the CSTR can reach a dynamic equilibrium. In this state, particles do react, get lost to the chamber walls, coagulate and so on. Due to the constant feed-in of new aerosol and flush-out old aerosol the overall concentration stays constant over time. We greatly expanded the description of the CSTR and the ongoing processes.

(new: P4 L7 – P5 L15)

**Filling regime**

[revised manuscript text omitted]

Another question is about the configuration of the CSTR in this study: did the authors use a real CSTR device for their experiments or not? what kinds of equipment (and how) were actually coupled with the CSTR, in addition to a CCN counter which enables the CCN activation measurements (i.e., the AF results) of aged soot particles?

The CSTR describes a reactor concept that is continuously filled with educts while products are constantly removed. The compounds inside the CSTR are ideally perfectly mixed, which is realized by actively stirring the compound with e.g. a fan.

In the context of atmospheric science, every environmental aging chamber can be operated as a CSTR. The only technical requirement is the presence of e.g. a fan to stir the aerosol. In our experiments we operated a 2.78m$^3$ steel chamber in CSTR-mode.

The description of the aerosol chamber used was extended from: (old P14 L20 - 21)

In the laboratories at ETH Zurich we performed aging experiments in a 2.78 m$^3$ stainless steel aerosol chamber run in CSTR mode.

To: (new P15 L20-29)

In the laboratories at ETH Zurich we performed aging experiments in a 2.78 m$^3$ stainless steel aerosol chamber operated in CSTR mode. A detailed description of the chamber can be found in Kanji et al., (2013). The chamber was actively mixed with a fan, but had no further features to enhance mixing e.g. baffles. All instruments were connected to the chamber with stainless steel tubing with 4 mm inner diameter. Since the maximal tubing length from the CSTR chamber to the analysis instruments was 3 m the impact on the overall residence time is negligible.

We investigated the change in CCN-activity of soot particles rich in organic carbon due to heterogeneous ozone oxidation. The soot particles were generated with the miniature Combustion Aerosol STandard (miniCAST, Model 4200, Jing Ltd., Zollikofen, Switzerland) which is propelled with propane and operates with a laminar diffusion flame. The miniCAST was operated under fuel-rich conditions (set point 6 according to the manual) in order to generate a soot which was rich in organic compounds (fuel-to-air ration: 1.03).

The aerosol chamber used is a "real CSTR" which is very close to an "ideal CSTR". We added a section where we explicitly point out that in terms of changes in the particle number concentration no deviation between our "real CSTR" and an "ideal CSTR" could be detected. (new P16 L10-20)

In the flushing regime the particle number concentration declines exponentially in both experiments. Eq. (5) describes the ideal/theoretical evolution of the particle number concentration in the flushing regime when taking the hydrodynamic residence time $\tau_{CSTR}$ according to eq. 1 into account. In the ideal case the decay is solely caused by the flushing process. In reality, the decay is a combination of flushing as well as additional particle losses e.g. wall losses or coagulation. Therefore, the real residence time can be obtained by fitting equation 5 to the experimental data after rearrangement for $\tau$, to which we refer to as $\tau_{flush}$ from now on (Kulkarni et al., 2011). In both experiments $\tau_{flush}$ coincides at 104 min, which is lower than the hydrodynamic residence time $\tau_{CSTR}$ of 111 min. In other words, the particle concentration declines faster than expected. This difference is caused by particle losses to the chamber wall, which acts as an additional particle sink parallel to flushing and reduces the particle lifetime. Nevertheless, statistical analysis of the experimental data results in purely statistical noise centered on the fitting curve used to determine $\tau_{flush}$. This indicates that in terms of mixing no difference between an ideal CSTR and the aerosol chamber used here can be detected with the applied instrumentation.

Overall, we conducted a broad range of experiments in which we investigated the impact of different aging conditions on different types of soot particles. We investigate parameters like single particle mass, hygroscopicity, chemical composition, ice nucleation activity as well. Results from these experiments are not presented within this manuscript but will be published soon. Mentioning of these measurements would be beyond the scope of this manuscript. In addition, in view of the rather theoretical nature of our manuscript we aim to streamline the content as much as possible. Thereby, we hope to limit the potential

distraction of the reader to a minimum and allow the reader to focus on the CSTR-approach and on the applicability of this approach.

Corresponding details are suggested to be provided especially
for those who are unfamiliar with such systems. From my perspective, the
organization of this section could be improved for better delivery of the key points.

We expanded and rewrote the introduction section and made a clearer distinction between the application of the CSTR approach for atmospheric experiments, the development of a CSTR-specific mathematical framework, the newly developed $t_{act}$-concept, as well as the application of the $t_{act}$-concept to other continuous flow steady state chambers, namely OFRs. (new: P2 L40- P4 L7)

In the following, we discuss a theoretical basis for the analysis of time-dependent changes in binary systems within well-mixed continuous flow aerosol aging chambers (CSTR-approach). We developed a mathematical framework which allows the retrieval of characteristic parameters from the system of interest (e.g. CCN activity) and which allows for the calculation of the parameter of interest throughout the entire duration. Key element in this framework is the activation time ($t_{act}$) which marks the time after which the individual aerosol particle undergoes a transition within a binary system. We start by introducing an idealized system in which $t_{act}$ can be described by a single number and proceed to a more realistic setting in which we incorporate a distribution of particles with different individual $t_{act}$'s (activation time distribution, P($t_{act}$)). Further, we test the $t_{act}$-concept on real experimental data and finally apply it to other types of continuous flow aging chambers such as OFRs. We show that application of the $t_{act}$-concept is capable of giving new insights to ORF data and further significantly improves the understanding of discrepancies in experimental results obtained in intercomparison studies Lambe et al., (2011) with different reactors such as the Potential Aerosol Mass Chamber (PAM) chamber and the Toronto Photo-Oxidation Tube (TPOT).

Furthermore, we expanded the section where we introduce the batch-mode aerosol chambers, the OFRs and compare both types to the CSTR. (new P3 L17-39)

In an aerosol chamber operated in batch mode, the reaction volume is first filled with the sample aerosol as fast as possible to achieve high homogeneity of the sample. After the desired start concentration is reached further addition of the sample aerosol is stopped and the aging is initiated e.g. by addition of the oxidant. This point in time is generally defined as the start of the experiment and referred to as $t = 0$. Data acquisition of the ageing sample takes place while the reaction volume is flushed with sample-free gas. The composition throughout the chamber is homogeneous but evolving in time, therefore no steady state conditions are ever achieved. This concept is used to operate many large scale environmental chambers (Cocker et al., 2001; Leskinen et al., 2015; Nordin et al., 2013; Paulsen et al., 2005; Platt et al., 2013; Presto et al., 2005; Rohrer et al., 2005).

A PFR is a steady state reactor in which no mixing along the flow path (axial mixing) but perfect mixing perpendicular to the flow (radial mixing) takes place. Further, a continuous feed-in of reactants and withdrawal of sample take place at equal flow rates simultaneously. This results in a constant composition of the output solely depending on the residence time

within the reactor. This ideal system is approximated by many Oxidation Flow Reactors (OFR) e.g. PAM chamber (George et al., 2007), TPOT Chamber (Kang et al., 2007), Micro Smog Chamber (MSC; Keller and Burtscher, 2012), or the TUT Secondary Aerosol Reactor (TSAR ; Simonen et al., 2017). The main difference between an ideal PFR and real OFRs is that in OFRs significant but unintentional mixing of the aerosol along the flow path takes place (Mitroo et al., 2018). Therefore, OFRs show a significant residence distribution.

The CSTR is a steady state reactor with a constant reactant feed in and sample withdrawal as well but opposite to OFRs, the volume is actively stirred to achieve a homogeneous composition throughout the reactor volume. Due to the active mixing, sample stream composition and conditions are the same as within the entire chamber volume. The concept of the CSTR requires perfect internal mixing, which cannot be achieved in real systems. However, due to the good miscibility and low viscosity of gases and the aerosol particles being homogenously dispersed, it is possible to achieve a degree of mixing which is very close to a perfectly mixed system. Especially in the case of mimicking atmospheric processes, residence times of several hours are achieved. Compared to that, the time needed for dissipating all gradients, which is in the order of seconds to minutes, can be considered small.

2. The Section 6, especially the last paragraph of which, is quite confusing. It is good to see the application of the activation time concept ($t_{act}$) into data interpretation of previous chamber studies, with improved agreements among different datasets. Nevertheless, there are several concerns need to be addressed. First of all, the previously used chambers such as PAM, they are actually not CSTR or far from the ideal mixing condition during oxidation. As a result, how can you simply apply the $t_{act}$ or RTD concept for CSTR system into the data interpretation of OFR/PAM reactors? Necessary information is needed to clarify this point.

The PAM and TPOT chambers are indeed no CSTRs, are not internally well-mixed and therefore cannot be described with the CSTR-specific mathematical framework introduced here. However, they show a significant residence time distribution due the mixing along the flow path as shown in multiple publications and discussed in the publication Lambe et al. 2011 which we refer to in our manuscript. The $t_{act}$- is not limited to CSTR chambers but can be applied to resolve the information from any data acquired in a chamber that is characterized by a residence time distribution. A RTD means that the aerosol particles that leave the OFR stayed inside the chamber for different individual durations. If the measured *AF* behind the OFR is 0.3, we raise the question which particles of the whole aerosol particle distribution are the CCN-active particles. Following the general concept in atmospheric science in which it is assumed that an increase in ageing duration is associated with an increase in hydrophilicity/hygroscopicity turning an initially CCN-inactive particle into a CCN-active particle, the $t_{act}$ concept defines that only the oldest 30% of the particles are CCN-active and the youngest 70 % are CCN-inactive. The time that separates the youngest and CCN-inactive 70 % from oldest and CCN-active 30 % is $t_{act}$ (necessary aging time).

In principle it is also possible that young and old particle activate equally well, however this seems to be unlikely for the BES-particles discussed here. Furthermore even this behavior

could be captured by the $t_{act}$-distribution. In this unlikely case the activation time distribution would be a horizontal line ($P(t_{act})$ = 0.3 ) but neither a peak nor a Gaussian shaped distribution.

In this manuscript we define 2 scenarios that combine this $t_{act}$-concept with the RTD reported by Lambe et al. (Figure 7). In the first scenario (High-OH) we calculated the fraction of particles older than 40 s in the PAM and TPOT chamber. In the second scenario we did the same with a time period of 180 s. As can be seen, the fraction of particles older than this threshold time $t_{act}$ varies between both chambers. Since only the oldest particle can be CCN-active this leads to different experimentally determined *AF*-values. In the high-OH scenario, *AF* in the TPOT is higher than in the PAM chamber. In the low-OH scenario, *AF* in the TPOT is lower than in the PAM chamber. The same trend was reported by Lambe et al (2011).

However this a qualitative application of the $t_{act}$-concept. We added following section that mentions what would be needed for a quantitative application of $t_{act}$–concept. (new: P20 L23-36)

Up to now, the discussion did not include many important processes that are relevant in aging chambers e.g. particle wall-interaction, gas-phase-partitioning, fluctuating input concentrations while field measurements, or inhomogeneities inside the OFR. These aspects are important for many processes such as the formation of SOA and can be incorporated to the $t_{act}$-concept by modifying eq. (13). As the actual calculation requires a multidimensional data array and detailed knowledge about the chamber of interest, this subject matter is beyond the scope of this publication and will not be discussed further. Nevertheless, the overall conclusion is that application of the original/non-adjusted $t_{act}$-concept can explain why measurements within different OFR chambers agree in parameters, which dependent on the bulk properties of the aerosol particle population (e.g. average O:C ratio) and at the same time disagree in parameters, which are dependent on the condition/status of the individual particle (e.g. CCN-activity). Therefore, we suggest to apply the concept of the activation time $t_{act}$ or the activation time distribution $P(t_{act})$ as metric in addition to calculating average values, such as the global *AF* and OH-exposure if following conditions are met. One, the system or parameter of interest can be described as a binary system and undergoes step-wise / non-gradual transitions such as CCN-activity. Two, the OFR used has a RTD broad enough to influence the outcome. Three, the conditions inside the reactor are either homogeneous or a correction for inhomogeneities (e.g. different oxidants concentrations inside the reactor) is implemented.

Another issue is that discrepancies in CCN activity of SOA formed from chamber oxidation experiments could be influenced by various factors, such as gas-particle partitioning and particle-phase reactions during SOA production as well as liquidliquid phase separation during activation processes. Additionally, the variability in different operation parameters such as relative humidity, initial concentration of VOC precursors, and acidity in the OFR/PAM chamber can affect the SOA formation process even for a same average OH concentration condition, further influencing the subsequent CCN activation process. In this sense, how to evaluate or exclude the impacts of these factors on the agreement of CCN activity (or AF)

measurements for different types of OFR or PAM experiments? Namely, how can we confirm that the discrepancies are predominantly introduced by the activation time (or RTD) rather than by the other influencing parameters, although the application of tact can better capture the deviation of CCN activity (likely due to change in chemical compositions) than what the bulk H/C and O/C ratios do? Further discussion is needed to clarify the abovementioned points.

We only refer to the aging of BES-particle in the PAM and TPOT-chamber. Therefore, we do not discuss the application of the $t_{act}$-concept onto the formation and aging of SOA. In addition we limit the application of the $t_{act}$-concept to the results obtained by Lambe et Al. (2011) in our manuscript as being qualitative. To identify if the RTD of the two chambers is the only reason for a different measured *AF* a quantitative analysis would be needed. However, this greatly exceeds the scope this manuscript. Nevertheless, the $t_{act}$-concept predicts well the overall trend in *AF* measured downstream both chambers. This is described in the section above.

The formation and subsequent aging of SOA in OFRs is a complex process and cannot by fully described by the $t_{act}$-concept introduced here. The main factor that inhibits a straightforward application is that SOA-particles only form in the chamber. As long as the educts are gaseous and therefore fully miscible, no air parcel can be separated from another air parcel by a measurement of the CCN-activity. After particles formed, each particle can have its individual trajectory inside the OFR. Therefore, particles can have different individual aging times and individual degrees of chemical modification. This can be detected by e.g. a CCN-Counter. For gases in an OFR that's not possible. Furthermore, the $t_{act}$-concept relies on the fact that a single particle is a closed system. If one particle has a high concentration of substance A while another particle has a low concentration of substance A, no exchange processes can dissipate this gradient. In the case of SOA, this can be an invalid assumption. Volatile compounds can evaporate and can condense on other particles and dissipate concentration gradients. An extended framework can potentially capture these processes and the $t_{act}$-concept could be part of this.

Other aspects that should be implemented in such extended framework, would be the internal inhomogeneities of the OFR. For example, the concentration of OH-radicals inside the OFR is inhomogenously distributed. The OH-concentration, for example, increases towards the light sources. Knowledge of the temperature gradient/profile would also be relevant since the speed of chemical reactions is typically temperature dependent. At this point, we want to mention that every additional variable increases the complexity of the math exponentially. This degree of complexity was a major reason why we favored an internally mixed aerosol chamber. This allowed us to assume homogeneous conditions throughout the entire experiment.

Nevertheless, we extended the respective discussion in section 5 to read as follows:

P20 L23-35

Up to now, the discussion did not include many important processes that are relevant in aging chambers e.g. particle wall-interaction, gas-phase-partitioning, fluctuating input concentrations while field measurements, or inhomogeneities inside the OFR. These aspects

are important for many processes such as the formation of SOA and can be incorporated to the $t_{act}$-concept by modifying eq. (13). As the actual calculation requires a multidimensional data array and detailed knowledge about the chamber of interest, this subject matter is beyond the scope of this publication and will not be discussed further. Nevertheless, the overall conclusion is that application of the original/non-adjusted $t_{act}$-concept can explain why measurements within different OFR chambers agree in parameters, which dependent on the bulk properties of the aerosol particle population (e.g. average O:C ratio) and at the same time disagree in parameters, which are dependent on the condition/status of the individual particle (e.g. CCN-activity). Therefore, we suggest to apply the concept of the activation time $t_{act}$ or the activation time distribution $P(t_{act})$ as metric in addition to calculating average values, such as the global $AF$ and OH-exposure if following conditions are met. One, the system or parameter of interest can be described as a binary system and undergoes step-wise / non-gradual transitions such as CCN-activity. Two, the OFR used has a RTD broad enough to influence the outcome. Three, the conditions inside the reactor are either homogeneous or a correction for inhomogeneities (e.g. different oxidants concentrations inside the reactor) is implemented.

**Specific comments:**
1. **Abstract**: What does the "non-gradual transitions" refer to here (Line 12)?

We greatly expanded the explanation of the phrase "non-gradual transition" throughout the entire manuscript (see major comment). In the abstract we changed it from:  (old: P1 L12)

We show that this concept can be applied to other systems investigating non-gradual transitions.

To: (new: P1 L14-15)

This experimental approach and data analysis concept can be applied for the investigation of any transition in aerosol particles properties that can be considered as a binary system.

 In the last sentence, what specific kinds of "discrepancies" are you suggesting? It is better to clarify these concepts precisely, as which are important points to show the significance and applicability of this study.

We show for the specific example of CCN-activation of photochemically aged BES-particles, that the activation time concept is beneficial for the analysis and interpretation of OFR data. However it is not limited to this, since it is a general concept. These points are now intensively discuss in the expanded manuscript.

Changed from: (old: P1 L13-15)

Furthermore we show how $t_{act}$ can be applied for the analysis of data originating from other oxidation flow reactors widely used in atmospheric sciences. This concept allows to explain discrepancies found in intercomparison of different chambers.

To: (new: P1 L15-19)

Furthermore, we show how $t_{act}$ can be applied for the analysis of data originating from other reactor types such as Oxidation Flow Reactors (OFR), which are widely used in atmospheric sciences. The new $t_{act}$ concept significantly supports the understanding of data acquired in OFRs especially these of deviating experimental results in intercomparison campaigns.

*2.* Page 2, line 8: How is the *"perfectly mixed"* defined here? It is unclear especially to readers those are unfamiliar with the CSTR technique.

To avoid the phrase "perfectly mixed" we mention instead that a homogenous aerosol mixture can be achieved by actively stirring the aerosol inside the chamber. This is a essential requirement for a CSTR operation. At this point "perfectly mixed" is accurate since we refer to an ideal system. However, this can be confusing as the reviewer pointed out.

changed from: (old: P2 L7-8)

The aerosol inside the CSTR is perfectly mixed, therefore a mix of aged and unaged aerosols is continuously extracted from the CSTR for analysis.

To: (new: P2 L14-15)

The volume of the CSTR is actively stirred in order to achieve a homogenous aerosol mixture. Due to the mixing, the aerosol that is continuously extracted for analysis consists of a well-defined mixture of aerosols at different aging stages.

 Following which, what do
you mean that *"real processes in the atmosphere where aerosols are constantly emitted, mixed and removed"*? Are you sure of the "constantly" condition in the ambient environment? Which specific atmospheric processes have you included in this statement, any references can be provided to support the idea?

As the reviewer pointed out many readers are not familiar with CSTRs. This reference to atmospheric processes shall give the reader a better understanding what it means to operate an aerosol chamber in CSTR-mode. To the authors' knowledge, all other experimental approaches that use flow tube or batch-chamber aim to generate a uniformly aged aerosol. This stands in a strong contrast to the approach presented here, where we are aiming for a non-uniformly aged aerosol output.

This loosely resembles the conditions in the atmosphere. Due to the persistent emission of aerosol into the atmosphere form various sources, the ongoing modification and the ongoing removal of aerosols from the atmosphere a mixture of young, medium-aged, and old aerosols are present simultaneously in the atmosphere. Similar to that a freshly produced aerosol is fed-in/emitted to the CSTR. Inside the CSTR, the aerosol is constantly chemically modified and mixed with the fresh aerosol. The aerosol that is removed from the chamber is a mixture of young, medium-aged, and old aerosols. We acknowledge that the atmosphere is not a CSTR, however both are comparable in the mixing aspect.

We clarified that we compare the CSTR-approach and the atmosphere in terms of mixing aerosols and measuring a non-uniformly aged aerosol.

Changed from: (old: P2 L8-9)

This approach is close to real processes in the atmosphere where aerosols are constantly emitted, mixed and removed as well.

To: (new: P2 L16-19)

From this perspective, the CSTR approach is closer to atmospheric processes than other reactor types as in the real atmosphere except for individual plume emissions aerosols are rather continuously emitted, mixed, and removed. This results in a mixture of aerosols at different aging stages, but of course, the atmospheric mixture is less well defined compared to an aerosol in a CSTR.

3. **Equation 5**: Why is the exponential part not expressed as "$e^{\left(\frac{t-t_{switch}}{\tau_{CSTR}}\right)}$" for the flushing regime? Please check the conversion carefully.

We thank the reviewer for taking the time and checking the equations in detail. That was an oversight on our side and we implemented a correct version.

Equation changed from

$$[A_{CSTR}(t)]=[A(t=t_{switch})]\cdot e^{\left(\frac{-t}{\tau_{CSTR}}\right)} \tag{4}$$

To

$$[A_{CSTR}(t)]=[A(t=t_{switch})]\cdot e^{\left(\frac{-t-t_{switch}}{\tau_{CSTR}}\right)} \tag{5}$$

4. Page 6, line 20: As a crucial parameter introduced in this study, the activation time ($t_{act}$) for non-gradual transitions was developed. However, what do you mean "If all the other parameters stay constant" during non-gradual transitions, which specific parameters are you referring to?

"If all other parameters stay constant…" refers to external parameters that can trigger non-gradual/step-wise changes in a particle. We added examples for relevant external parameters. We created this $t_{act}$-concept based on this assumption since we operated our experiment in a temperature controlled chamber, at a defined RH, and at an ozone concentration that was actively kept constant.

Changed from: (old: P6 L20-21)

If the all other parameters stay constant, while a particles undergoes changes that result in a non-gradual transitions, this transition can be described as a function of time.

To: (new: P6 L26-30)

We may assume a system in which all external parameters stay constant but the particle itself undergoes a continuous transformation, e.g. due to oxidation. After a certain period of time, this continuous transformation, in this specific case oxidation, can lead to a change in a

binary property, e.g. CCN-activity. Ultimately, the step-wise or non-gradual transition is a function of time. We define the required time span (e.g. necessary aging time) that leads to a change in a specific particle property, resulting in a transition in a binary system in another particle property as the activation time ($t_{act}$).

Is it easy to achieve in practical conditions of laboratory chamber experiments?

This section introduces a theoretical and idealized concept for which constant background concentrations are assumed, therefore, we try to limit the discussion of experimental short-comings in this section. Nevertheless, it is experimentally feasible to keep certain parameters like temperature, relative humidity, and ozone concentration constant in the aerosol chamber at ETH Zurich and any chamber designed similarly. The fan inside the chamber creates a homogenous atmosphere. The temperature can be actively controlled with a heater/chiller. The ozone concentration was constantly measured and kept stable with a feedback loop that regulated the ozone source. This is an experimental advantage towards many OFRs. Results from additional experiments not mentioned within this manuscript will be published within the next months.

5. Equation 7: I think it should be "$e^{\frac{-t_{switch}}{\tau_{CSTR}}}$" for the flushing regime? Please check the conversion carefully

We thank the reviewer for raising this point and have reevaluated the equation. The version suggested by the reviewer would cause $t_{switch}$ to have an influence on the filling regime. This is not possible since $t_{switch}$ defines the end of it. Therefore, we refrain from changing the equation.

6. **Equation 8**: Why is the simplified equation not expressed as '$t_{act}$ = $-\ln(AF(t))\cdot \tau_{CSTR}$'? I'm wondering how will the value of $AF(t\rightarrow\infty)$ be, could it be 0 as suggested by the exponentially decreased curve in Fig.2, or probably approaching 1 like what AF responds when switching to the flushing regime as shown in Fig.3? How should the readers understand the corresponding physical meaning of $AF(t\rightarrow\infty)$ in this steady state condition? Corresponding details are necessary.

The equation 6a. 6b and 7 describes how $AF$ inside the CSTR evolves over time while filling the CSTR.

$$t \leq t_{act}: \quad AF(t)= 0 \tag{6a}$$

$$t > t_{act}: \quad AF(t)= \frac{\text{activated particles}}{\text{all particles}} = \frac{\int_{t=t_{act}}^{t} RTD(t)\,dt}{\int_0^t RTD(t)\,dt} = \frac{RTD_{sum}(t)-RTD_{sum}(t=t_{act})}{RTD_{sum}(t)} \tag{6b}$$

$$t_{act}= \ln\left(1- \left((1-AF(t))\cdot\left(1-e^{\frac{-t}{\tau_{CSTR}}}\right)\right)\right)\cdot(-\tau_{CSTR}) \tag{7}$$

During this time the particle concentration inside the CSTR is increasing continuously. Parallel to that $AF$ is also changing. After a certain time the changes in the particle concentration become negligible. The CSTR is then in a dynamic equilibrium called "steady state". In this dynamic equilibrium, aerosol particles still become CCN-active due to aging but already CCN-active particle are also constantly removed from the CSTR by flushing. This state can be maintained for an in theory infinite time span. This means that the experimental duration $t$ grows continuously, but the conditions inside the CSTR are constant. Therefore, the distribution of young, medium-aged, and old particle inside the CSTR is constant and decoupled from the experimental duration $t$. This also means that $AF$ and the activation time $t_{act}$ calculated from it do not change anymore. The same applies to OFRs which are operated as steady state reactor as well.

In the old version of the manuscript the x-axis in Fig. 2 was labeled with "particle age / min" which was now changed to "residence time / min". The individual residence time is equal to the particle age. However, this is only mentioned at a later point in the manuscript, which is indeed misleading. The curve in Fig. 2 does therefore not show how particle number concentration that declines with increasing experimental duration. Instead it shows the abundance of particles with a defined residence time (=individual particle age) in the total aerosol particle mixture.

Figure 2 shows the RTD during steady state. The area under the curve represents the total particle population. The colored area is the fraction of particles that is older than the defined $t_{act}$. This "oldest fraction" is equal the CCN-active fraction in our model. Since this curve refers to the steady state it does not change over time, but stays constant. Therefore $AF$ does neither decline exponentially nor does it approach 1, but stays constant with increasing experimental duration. Also, the correlation between the individual necessary aging time $t_{act}$ with the measured $AF$ is not linear but exponential as can be deduced from the different $AF$-values in Fig.2.

Equation 8 does not contain any dependency on the experimental duration $t$. In fact, equation 8 is derived from eq 7 for ($t \rightarrow \infty$). This is now stated in the manuscript as: (new: P7 L20 –P7 L1)

[revised manuscript text omitted]

7. Page 8, line 7: It sounds a bit strange of "global" AF? Is the "global" trying to represent the specific exponentially increased AF inside CSTR or just to show a different AF case with other non-CSTR chamber experiments?

A consequence of the CSTR-approach is that aerosol particles with different individual aging times are present in the chamber at the same time. Each of these aerosol fractions is CCN-active to a different degree. The term "global $AF$" refers to the combination of all $AF$-values from different individual aerosol particle fractions combined (=global). This global $AF$ is the $AF$ that is measured downstream the chamber

The explanation of the global $AF$ in the manuscript was changed and postponed to section 3.2 "Introducing the activation time distribution $P(t_{act})$". The text now reads as follows

To: (new: P11 L9-22)

The approach discussed so far is based on the assumption that all aerosol particles are identical and therefore a specific property of the whole aerosol population can be described with a single parameter. In other words, all particles have the same tact in case of CCN activity being the specific property. However, this is not the case for many parameters. In case of the particle diameter, for example, every aerosol particle has its individual diameter and the total population can be described by a distribution of particle diameters around a mean diameter. An eventual size-selection does impact the mean diameter and the width of the distribution. Still, the size selected particles will not have the identical diameter. Furthermore the aerosol population might be mono-modal and narrowly-distributed with respect to one parameter such as the aerosol particles electrical mobility diameter, but it can be multi-modal or broader distributed with respect to another parameter e.g. the aerodynamic diameter.. Therefore, it has to be expected that the activation time ($t_{\mathrm{act}}$) is also characterized by a distribution. For this we introduce the activation time distribution $P(t_{\mathrm{act}})$ and discuss its theoretical impact on transitions within binary systems in CSTR-experiments. In contrast to a uniform $t_{\mathrm{act}}$ valid for all particles the activation time distribution $P(t_{\mathrm{act}})$ is more realistic as it takes the individual $t_{\mathrm{act}}$ of the individual particles within the population into account. Nevertheless, only one value for $AF$ can be determined experimentally. This single value, from now on referred to as global $AF$, represents the average $AF$ over all $AF$s of the individual sub-fractions within the population as will be explained in more detail in the upcoming sections.

Line 10: *"… and therefore the global AF only if tact = tswitch."* Some information was missed in this sentence.

We thank the reviewer for pointing out the limited text to explain the concept. The concept present here assumes that $AF$ inside the CSTR is equal to the fraction of particles older than the threshold time $t_{\mathrm{act}}$. Therefore, $AF$ can be calculated by calculating the fraction of particles older than this time. However, this comes with some difficulties in the flushing regime.

In Figure 1 it can be seen how the RTD changes after the particle feed-in is stopped. The whole RTD-curve from steady state is shifted towards longer residence times. As a result of this, the fraction of particles older than a threshold time increases exponentially. This increase of the "old particle fraction" is captured in equation 9.

$$AF(t)_{\mathrm{flushing}} = \int_{t_{\mathrm{switch}}/\tau_{CSTR}}^{t/\tau_{CSTR}} e^{\left(\frac{t - 2 \cdot t_{\mathrm{switch}}}{\tau_{CSTR}}\right)} \, \mathrm{d}\left(\frac{t}{\tau_{CSTR}}\right) \tag{9}$$

This equation 9 is the result of centering of the RTD during steady state at $t = t_{\mathrm{switch}}$. Therefore, equation 9 describes only this fraction of particles which is older than $t_{\mathrm{switch}}$. Since the fraction of particles older than a defined threshold time is equal $AF$, this equation would describe the change in $AF$ for the specific case that $t_{\mathrm{switch}}$ and $t_{\mathrm{act}}$ are equal. Implementing all other cases requires a shift of the starting point of equation 9. This is done by introducing the parameter $t_{\mathrm{offset}}$.

We rewrote the entire section, in order to increase the understandability (new: P9 L7 – P10 L14)

[revised manuscript text omitted]

**8. Title of Sect.4**: What does the "first experiments" mean? Try to update the message into a more informative one.

The caption "Application in first experiments" was removed. The following captions were rephrased.

9. Page 12, line 10: What does the "uniform" mean: "activate uniformly" here and "a uniform aerosol population" in the caption of Fig.5? Are you trying to say the initial particles with the same particle size and chemical composition? If so, how to understand the Gaussian distribution scenario (i.e., *"This is because there are some particles in the population, that activate earlier than the mean activation time."*), as all the uniform particles are supposed to activate at a same activation time? More straightforward/concise descriptions would be useful to explain the scenario clearly.

As we aim to introduce a new metric we tried to increase the complexity of the topic stepwise by starting with in idealized scenario, which we do not consider scenario as the most likey/realistic scenario. It is more realistic that some particles activate after a shorter/longer aging time than other particles. A reason for this can be that even in a size-selected aerosol particle flow some particles are slightly small/larger than the average diameter. Considering the case of salt particles, the CCN-activity shows a size-dependence which can lead to a non-uniform activation of the aerosol particles.

In the sections 3.2 "Introducing the activation time distribution $P(t_{act})$" and 3.3 "Impact of the activation time distribution on the individual $AF$" we move on to implement the more realistic scenario by implementing a scenario where particles do not activate uniformly, but

show a distribution of different $t_{act}$'s. We use a Gaussian distribution to capture the fraction of particles that activate after a certain necessary aging time $t_{act}$.

In section 3.4 "Calculation of the total activated fraction (global *AF*)" implement the activation dime distribution/$P(t_{act})$-concept and calculate the change in global *AF* throughout a full CSTR-experiment. This is done by integration of the contribution of individual aerosol fractions over the whole range of possible $t_{act}$'s (equation 13)

$$AF(t) = \int_{t_{act}=0}^{t_{act}=t} AF(t_{act}, t) \cdot P(t_{act}) \; dt_{act} \tag{13}$$

We further revised the text to highlight the difference between $t_{act}$ of individual particle sub-populations leading to individual *AF*s and the global *AF*, which can be determined experimentally. The respective text reads now as follows:

P12 L1-22

For simplicity we discuss the impact of the activation time distribution ($P(t_{act})$) in steady state first but the concept is the same throughout the entire experiment including the filling as well as the flushing regime. In a CSTR particles with different individual residence times are present at the same time due to the continuous feed in of fresh particles and the active mixing. For a better understanding the "individual residence time of a particle" will be referred to "particle age" from here on. With an increasing particle age the number of particles (=area under the curve) declines in a CSTR during steady state (Graph A, Fig. 4). Nevertheless, the activation time distribution $P(t_{act})$ is the same for all particles (Graph B, C, D in Fig. 4), regardless of their age. The fraction of activated particles inside the CSTR (global *AF*) therefore has to be described as an overlap of the residence time distribution (RTD; black curve in Fig. 4.A) and the activation time distribution ($P(t_{act})$; red curves in Fig. 4.B, C, and D).

In Fig 4.A, three individual sub-populations are indicated by red vertical bars. The first sub-population at $t$ = 60 min = 0.5 $\tau$ consists of rather young and fresh particles. Their corresponding activation time distribution $P(t_{act})$ is shown in sub-panel B. While there are a lot of particles (indicated by the large area under the red curve) only a small fraction of these particles is CCN-active. The active fraction is indicated by the red colored area and corresponds to the particles with a very low individual $t_{act}$. The contribution of this sub-population to the global *AF* is therefore small. In sub-panel D, a sub-population at $t$ = 360 min = 2.0 $\tau$ of old and well-aged particles is shown. Due to their high age, litterally all particle in this sub-population are CCN-active as their individual particle age is more than 6 sigma beyond the mean value of the exemplarily discussed Gaussian activation time distribution $P(t_{act})$. Since the overall fraction of these old particles is low as indicated by the significantly reduced area underneath the red curve compared to the first sub-population (panel B), they contribute only little to the global *AF*. In sub-panel C, a medium aged sub-population at $t$ = 180 min = 1.5 $\tau$ is shown. On the one hand, there are significantly less particles than in the first sub-population (panel B), which is indicated by the reduced area underneath the red curve. On the other hand, the fraction of CCN active particles within this

sub-population is significantly larger than in the first one. In fact, the fraction is 0.5 as this sub-population has an individual residence time that is equal to the mean ($\mu$) of $P(t_{act})$.

10. Page 15, line 10: How was the particle wall loss rate of $k$ = 0.000625 min-1 estimated? Where can the readers find the corresponding clues/data for calculation?

Within this manuscript the wall loss rate was obtained from the difference between theoretical and expected particle loss during flushing under the assumption of a first order loss kinetic.

We extended the explanation of how we calculated the particle wall loss numbers and refer to publications were the same calculation regarding particle losses was applied.

The section was changed from: (old: P15 L8-11)

The measured particle flush rate $\tau_{flush}$ obtained during the flushing regime is 104 min in both experiments, which differs slightly from the theoretical flush rate $\tau_{CSTR}$ = 111 min. This difference is caused by particle losses to the chamber wall. From this difference the particle wall loss rate of $k$ = 0.000625 min-1 and a mean particle life time upon wall loss of 1600 min was determined assuming first order loss rates

To: (new: P16 L10 –L24)

In the flushing regime the particle number concentration declines exponentially in both experiments. Eq. (5) describes the ideal/theoretical evolution of the particle number concentration in the flushing regime when taking the hydrodynamic residence time $\tau_{CSTR}$according to eq. 1 into account. In the ideal case the decay is solely caused by the flushing process. In reality, the decay is a combination of flushing as well as additional particle losses e.g. wall losses or coagulation. Therefore, the real residence time can be obtained by fitting equation 5 to the experimental data after rearrangement for $\tau$, to which we refer to as $\tau_{flush}$ from now on (Kulkarni et al., 2011). In both experiments $\tau_{flush}$ coincides at 104 min, which is lower than the hydrodynamic residence time $\tau_{CSTR}$ of 111 min. In other words, the particle concentration declines faster than expected. This difference is caused by particle losses to the chamber wall, which acts as an additional particle sink parallel to flushing and reduces the particle lifetime. Nevertheless, statistical analysis of the experimental data results in purely statistical noise centered on the fitting curve used to determine $\tau_{flush}$. This indicates that in terms of mixing no difference between an ideal CSTR and the aerosol chamber used here can be detected with the applied instrumentation.

When dividing the real particle life time ($\tau_{flush}$) into its individual components, a particle life time upon wall losses ($\tau_{wall-loss}$) of 1600 min can be determined in accordance with first order wall loss kinetic (Crump et al., 1982; Wang et al., 2018). The influence of particle coagulation can be considered negligible due to the low coagulation rate of 100 nm particle at concentrations of maximum 1500 cm$^{-3}$ (Kulkarni et al., 2011).

11. Page 15, line 15: What is the meaning of the last sentence? What does the "otherSS"

refer to? Where can readers find the corresponding details? Necessary information is needed.

We apologize for being not clear in our wording. Very often CCN measurements are performed by scanning through different supersaturations while e.g. the size of the particles is kept constant. Within the experiments presented herein, the CCNC was set to a constant supersaturation throughout most of the experimental duration to achieve a high time resolution in data acquisition. Nevertheless, after achieving steady state, the operation conditions of the CCNC were modified to allow for data acquisition at a range of supersaturations. Still, within figure 6, we present data at one supersaturation only. Therefore, in during the time, when data at a differing supersaturation was acquired, no data points are presented leading to the rather coarse time resolution in steady state.

Sentence changed from: (old: P16 L14-15)

The gaps in the curves during steady state are due performing measurements at other SS.

To: (new: P16 L27-28)

The gaps in the curves during steady state are due to changes in the operation of the CCNC form running on a constant SS (1.0% and 1.4%, respectively) to scanning over a range of SS.

12. **Figure** 2: Is the "particle age" of x-axis with the same meaning of the "residence time" in Fig.1? If not, please specify accordingly in the corresponding places.

We thank the reviewer for highlighting the fact that until this point in the manuscript we have not defined a potential difference in these two notations.

Therefore we changed the x-axis description from:

particle age / min

To:

residence time / min

13. **Figure** 6: Why is the unit of particle concentration in Fig.6(A1) and (B1) different from those in Figure 3? In Fig.6 (B1), why are the data after 800 min missing? As assumed early in this study that all compounds in CSTR have perfectly mixed thus with constant concentrations during steady state, how to explain the increasing trend in observed particle concentration in the duration of 400-600 min, i.e., AF almost reached a stable level around 0.2 at 1.4% SS conditions)? More detailed discussion should be provided in the corresponding data interpretation sections.

We apologize for the dot in the unit of the y-axises in Fig.6(A1) and (B1). This is a graphical error and was removed. The particle number concentrations are not meant to have different units, but are meant to be $\#/cm^3$.

The deviations from the theoretical changes in the particle concentration are due to a slightly changing particle input concentration. Despite all efforts, it was challenging to keep

the particle input concentration absolutely constant over a period of 12 h. However, this small change affects the outcome only to a small degree.

The section was changed from: (old: P15 L 6-8)

The particle concentration curves follow the theoretical filling and flushing curves in a CSTR. The slight decline in the concentration observed in the region where steady state is expected in graph A1 is due to a slight but undesired reduction in the feed-in flow that was experienced during the experiment.

To: (new: P15 L4 – 9)

The graphs A1 and B1 in Fig. 6 show the particle concentration (black crosses; left axis), the measured global *AF* (red crosses) and the fitted global *AF* (blue dashed line, both right axis). The particle number concentration curves (black crosses) follow the theoretical filling and flushing curves as expected in a CSTR (Fig. 3). The slight decline in the concentration in steady state in graph A1 is due to a slight reduction in the particle input concentration that was experienced during the experiment. Visa versa the slight increase in the number concentration in graph B1 is due to a slight increase in the particle input concentration over time.

14. Page 19, line 6: The last sentence is a bit confusing. It is better to clarify the "metric" here, e.g. metric of what specific aspects.

We extended the text appropriately to clarify under which circumstances the activation time concept can be beneficial for the data analysis

Changed from: (old: P19 L6)

Depending on the parameter of interest we suggest using $t_{act}$ or $P(t_{act})$ as metric.

To:  (new: P20 L31-36)

Therefore, we suggest to apply the concept of the activation time $t_{act}$ or the activation time distribution $P(t_{act})$ as metric in addition to calculating average values, such as the global *AF* and OH-exposure if following conditions are met. One, the system or parameter of interest can be described as a binary system and undergoes step-wise / non-gradual transitions such as CCN-activity. Two, the OFR used has a RTD broad enough to influence the outcome. Three, the conditions inside the reactor are either homogeneous or a correction for inhomogeneities (e.g. different oxidants concentrations inside the reactor) is implemented

**Technical corrections:**
1. **Abstract**, Page 1, line 10: *"… the newly introduced metric: activation time"*

done
2. Page 3, line 27: *"… can be calculated as a function of …"*. A similar issue exists in Line 16, Page 6.

done
3. Page 6, line 13: *"… to describe continues continuous changes"*?

done

4. Page 6, line 19: *"… can be considered as a non-gradual change."*

done

5. Page 6, line 20: *"If the all the other parameters stay constant, while a particles undergoes changes that result in a non-gradual transitions…"*

done

6. **Equation 9**: Why do you use different multiplication signs in these equations, e.g., "*" and "·"? It makes more sense to keep consistent within the same manuscript.

done

7. **Table 1**: Why is the layout of this table so different from other two tables in this manuscript? The corresponding details could be better organized.

We harmonized the table layouts

8. Title of **Sect.4.3**: *"Calculation of the total activated fraction"*

done

9. Page 12, line 12: *"While the uniform scenario shows no activity be for reaching tact …"* Do you mean 'before'?

changed to "before"

10. Page 14, line 8-9: *"As there is a significant share of particles activating significantly earlier than the nominal activation time ($\mu$ = 180 ) in the case of a Gaussian distribution a fraction of 1 % of the entire particle population within the CSTR is already activated after 87 min."* A comma is needed to clarify the point.

Changed from:

As there is a significant share of particles activating significantly earlier than the nominal activation time ($\mu$ = 180 ) in the case of a Gaussian distribution a fraction of 1 % of the entire particle population within the CSTR is already activated after 87 min

To: (P15 L5-7)

In the case of $t_{act}$-onset, there is a significant share of particles activating significantly earlier than the nominal activation time ($\mu$ = 180) in the case of a Gaussian distribution. Therefore, a fraction of 0.01 of CCN active particles within the entire particle population is already present after 87 min

11. Page 14, line 12: *"The difference in tact of 10 min between the two P(tact)-approaches is due to the application …"*
It is very common to see that tact was written as tact. Similar issues also exist in some other expressions, e.g., Pstep, which should be Pstep. Please check through the manuscript carefully and make necessary updates accordingly.
In the same paragraph, there are many long sentences without proper splits or

connections, which might make the readers difficult or even confused to catch the meaning effectively. For instance:

We apologize for the inconsistency and harmonized the subscripts.

12. Page 14, line 15-16: *"As can be seen in Graph C of Fig. 4, 50 % of the particles with a residence time equal to the nominal activation time are activated in the case of a Gaussian distribution corresponding to tact0.5."*

Comma added.

13. Page 14, line 23: *"… were diluted with particle-free and VOC-filtered air…"*

Changed from:

particle free

To:

particle-free

14. Page 14, line 25: *"The aerosol flow was fed into the aerosol chamber, where a constant  ozone concentration of 200 ppb was …"*

done

15. Page 14, line 27: *"The size distribution data was acquired by a … (SMPS) system from which the  total particle concentration could be derived."*

done

16. Page 15, line 5: The *"(blue solid line)"* is not needed, since there is only one curve in the corresponding subplots.

We thank the reviewer for mentioning this point. We have substantially revised our graphical presentations in the aim to keep the same type of curve the same layout throughout all figures. Therefore, we decided to keep the expression"(blue solid line)" to avoid any confusion with the curves in the graphs Fig.6 A1 and B1.

17. Page 15, line 18: *"… μ as well as σ  is larger for P(tact) at a 1.0 % SS of compared to the results obtained for 1.4 % SS. The mean activation time being larger for 1.0 % SS indicates that the longer the chemical aging proceeds, the initially inactive soot particles activate  at a lower SS."*

done

18. Page 15, Line 23: The comma between "*P(tact)*" and "requires" is unnecessary.

comma removed

19. Page 17, line 1 and 3: *"Within these types of chambers …"*

done

20. Page 17, line 10: *"secondary organic aerosol (SOA)"*, and the "VOCs" should be defined before when it appeared for the first time.

done

21. Page 17, line 24: *"…to be directly proportional to the AFs…"*

done

22. Page 18, line 3: *"…we  present two scenarios."*

changed

23. Page 18, line 9: *"…other parameters can agree very well."*

done

24. Page 19, line 22: *"…soot particles transitioning  from initial CCN-inactivity to CCN-activity over the course of …*

done

---

## Author Response (AR2)

**The reviewer's comments are written in black and our responses in red. Texts from the old version of the manuscript are given in blue and texts from the revised manuscript in green.**

Overall the manuscript has a few technical and stylistics quirks that are arguable (and some less so, e.g., P1 L17 "... (OFR)..." should be plural) yet I do not feel fit to criticize or amend, and leave to the discretion of the Editor. I thank the authors for educating me on the difference between an aerosol and an aerosol particle, as well as taking my previous suggestions into consideration.

We thank the reviewer once again for taking the effort in revising our manuscript. Supported and motivated by the two very detailed reviewer comments on the first manuscript we were eager to improve the understandability and quality of our manuscript. We are very pleased to hear that our efforts were successful and thank the reviewer for the positive feedback.

**OFR changed to OFRs throughout the entire manuscript**

P1 L10: I will insist here as I did throughout my first review. I recommend changing "...aerosol chamber as a Continuous flow Stirred Tank Reactor (CSTR)." To "...aerosol chamber in continuous fashion, aiming to mimic the characteristics of a Continuous flow Stirred Tank Reactor (CSTR)." The authors are free to choose what they see fit, as this is a small detail, but I will put my view forth once more. Same applies for: P2 L4-5.

We thank the reviewer for expressing his concerns. We agree upon the fact that the actively stirred aerosol chamber here used is not an ideal system and that discussing the difference between "real" and "ideal" reactors is crucial.

The acronym "CSTR" is widely used, but the definition of what a CSTR is can vary slightly. One approach is to describe an ideal model system with the acronym "CSTR". A real reactor system therefore could never be described as "CSTR". The second approach is to define the CSTR as real reactor system which mimics the operational conditions of an ideal CSTR. In these cases the phrases "ideal CSTR" (Mo and Jensen, 2016) or "Continuous Ideally Stirred-Tank Reactor" (Patil et al., 2007) are used to refer to the ideal reactor concept.

In accordance with the literature, we prefer to follow the second approach. Therefore, throughout the manuscript we refer to the "CSTR" as a real reactor system that approximates the model system of an "ideal CSTR". We discuss the difference between the "real CSTR" and the "ideal CSTR" on P3 L35-38. To ensure consistency in the wording throughout the whole manuscript we prefer keep the initial wording.

P2 L6-7: Unless I'm mistaken, the authors don't mean what they're saying when referring to Simonen et al.'s TSAR. When I read "This greatly exceeds the typical aging times of several minutes that can be reach within Oxidation Flow Reactors (OFRs; (Simonen et al., 2017)..." I automatically think the TSAR only ages compounds for a couple of minutes, but that's not the case. It just has a low residence time. May I ask the authors to rephrase, likely (if I understand correctly) from a point of view of V/Q?

We understand the objections of the reviewer. Within the atmospheric science community, the phrase "particle age" is frequently used synonymous with the "atmospheric age" or "photo-chemical age". The "atmospheric age" or "photo-chemical age" refers to an equivalent atmospheric aging time that can be simulated in e.g. OFRs in this context. For the calculation of the equivalent atmospheric aging time the residence time is multiplied with the reactant concentration inside the OFR normalized by its average atmospheric concentration.

We define the time an aerosol particle resides inside the chamber and is exposed to the oxidant as the actual time an aerosol particle is aged (=particle aging time). This definition is valid throughout the entire manuscript. We do not refer to an "equivalent atmospheric aging time". Therefore, we want to point out that the phrases "atmospheric age" or "photo-chemical age" are not the same as "particle age".

To avoid any potential confusion we changed the wording from:

This greatly exceeds the typical aging times of several minutes that can be reached within Oxidation Flow Reactors

To:

This greatly exceeds the typical exposure times of several minutes that can be reached within Oxidation Flow Reactors

P2 L13: consider a hyphen e.g. "CSTR-approach", as used in P2 L41.

We followed the reviewer suggestion and put hyphenated "CSTR-approach".

P2 L14: Again, I insist that phrasing like "The volume of the CSTR is actively stirred..." be changed. May I offer, for example, "The vessel is actively stirred...".

We implemented the reviewer suggestions but preferred to use a slightly different wording to avoid the use of additional synonyms that describe the same.

**Changed from:**

The volume of the CSTR is actively stirred in order to achieve a homogenous aerosol mixture.

To:

The volume of the aerosol chamber is actively stirred in order to achieve a homogenous aerosol mixture.

**P3 L5: "ORF"**

**changed to OFR**

P3 L7: Please cite Kang et al. after the PAM and George et al. after the TPOT.

**Citations added**

P3 L16: From a technical perspective, a flow tube reactor is not a PFR. Unless I'm mistaken, it's a term arising from atmospheric chemists. I'd ask the authors to remove "...or flow tube reactor...".

We thank the reviewer for pointing this out and removed "...or flow tube reactor...".

P3 L27: Citations for PAM and TPOT are reversed.

We adjusted the citations.

P3 L33: To the authors' choice, they can consider changing "volume" to "vessel".

**To keep the wording consistent we choose to keep the initial wording (see comment to line P2 L4).**

P3 L34-37: Yes! I thank the authors for taking my previous suggestions on v1 of the manuscript into consideration.

P6 in general: Thank you again to the authors for having a much clearer description of what 'nongradual transitions' and 'external parameters' are. I think now the readers, by enlarge, should be able to understand this at first pass.

P15 L20: Didn't the authors already introduce the SAPHIR earlier? I think they could just mention it so the reader is more aware they have prior knowledge of the SAPHIR and its modes of operation (continuous vs. batch).

The SAPHIR chamber at Research Center (FZ) Julich as well as the difference between continuous and batch mode operation were introduced in section "1 Motivation" and section "2 Introduction of the CSTR". The section starting at P15 L20 "4 Application of the new  $t_{act}$ -concept to experimental data from CSTR-aging experiments" focuses on chambers operated in continuous mode in general and specifically to the aerosol chamber deployed at ETH Zurich.

P15 L20-24: What was the flowrate? Depending on that the reader can estimate V/Q in the tubing and then decide if it really is negligible.

We added that the flow rate in the sample line was between 2 to 5 lpm. The residence time in the stainless steel tubes of 3 m length and 4 mm inner diameter ranged from 0.45 to 1.13 s. This is negligible compared to an average residence time of approximately 2 hours within the chamber.

**We changed the sentence from:**

Since the maximal tubing length from the CSTR chamber to the analysis instruments was 3 m the impact on the overall residence time is negligible.

**To:**

Since the maximal tubing length from the aerosol chamber to the analysis instruments was 3 m at flow rates were between 2 and 5 lpm the impact on the overall residence time (0.45 to 1.13 s) is negligible compared to an average residence time on the order of hours within the chamber.

**P16 L17: Visa versa?**

**Changed to: Vice versa**

P18 L9: I have never head of the puls-function. Perhaps pulse-function? But even then, I strongly believe in the technical difference between a pulse (finite peak height) vs. the PFR E-Curve (infinite peak height), so I would argue "often referred to as puls-function" be removed. In the same vein, it's not "due to technical restrictions" that any reactor (OFR or otherwise) is not on either end of the CSTR to PFR spectrum, it's simply reality. I appreciate these are small details but I think they're important.

We agree with the reviewer comment and apologize for the spelling mistake. The Dirac-Deltafunction is often approximated by a pulse-function. Therefore, both terms are used in the same context. However, they are not the same. The correct synonym for the Dirac-Delta-function is "impulse-function". Furthermore we agree that "technical restrictions" are not the only reasons why OFRs are neither ideal PFRs nor ideal CSTRs.

The section was change from:

The ideal OFR would be an ideal plug flow reactor (PFR) with a RTD being a Dirac-Delta-function, often referred to as puls-function. However, due to technical restrictions all OFRs have a RTD that lies between an ideal CSTR and an ideal PFR and is further dependent on the individual design of the OFR (George et al., 2007; Huang et al., 2017; Kang et al., 2007; Simonen et al., 2017).

To:

The ideal OFR would be an ideal plug flow reactor (PFR) with a RTD being a Dirac-Delta-function, often referred to as impulse-function. However, all OFRs have a RTD that lies between an ideal CSTR and an ideal PFR and is further dependent of the individual design of the OFR (George et al., 2007; Huang et al., 2017; Ihalainen et al., 2019; Kang et al., 2007; Simonen et al., 2017).

P20 L21: agree in 'some' parameters and disagree in 'other' parameters? This sentence, though, is a very good highlight of the work the authors have done and why it is important.

Once again, we would like to thank the reviewer for the positive feedback to our revision efforts.

[revised manuscript text omitted]